# The Case for Cleaner Biosignals: High-fidelity Neural Compressor Enables Transfer from Cleaner iEEG to Noisier EEG

**Francesco S. Carzaniga**[1,2]**, Gary Hoppeler**[3]**, Michael Hersche**[1]**,**
**Kaspar A. Schindler**[2]**, Abbas Rahimi**[1]
[1]IBM Research – Zurich, Rüschlikon, Switzerland
[2]Department of Neurology, Inselspital, Sleep-Wake-Epilepsy-Center,
Bern University Hospital, Bern University, Bern, Switzerland
[3] ETH Zürich, Zürich, Switzerland
frc@zurich.ibm.com

## Abstract

All data modalities are not created equal, even when the signal they measure comes from the same source. In the case of the brain, two of the most important data modalities are the scalp electroencephalogram (EEG), and the intracranial electroencephalogram (iEEG). iEEG benefits from a higher signal-to-noise ratio (SNR), as it measures the electrical activity directly in the brain, while EEG is noisier and has lower spatial and temporal resolutions. Nonetheless, both EEG and iEEG are important sources of data for human neurology, from healthcare to brain–machine interfaces. They are used by human experts, supported by deep learning (DL) models, to accomplish a variety of tasks, such as seizure detection and motor imagery classification. Although the differences between EEG and iEEG are well understood by human experts, the performance of DL models across these two modalities remains under-explored. To help characterize the importance of clean data on the performance of DL models, we propose BrainCodec, a high-fidelity EEG and iEEG neural compressor. We find that training BrainCodec on iEEG and then transferring to EEG yields higher reconstruction quality than training on EEG directly. In addition, we also find that training BrainCodec on both EEG and iEEG improves fidelity when reconstructing EEG. Our work indicates that data sources with higher SNR, such as iEEG, provide better performance across the board also in the medical time-series domain. This finding is consistent with reports coming from natural language processing, where clean data sources appear to have an outsized effect on the performance of the DL model overall. BrainCodec also achieves up to a $64\times$ compression on iEEG and EEG without a notable decrease in quality. BrainCodec markedly surpasses current state-of-the-art compression models both in final compression ratio and in reconstruction fidelity. We also evaluate the fidelity of the compressed signals objectively on a seizure detection and a motor imagery task performed by standard DL models. Here, we find that BrainCodec achieves a reconstruction fidelity high enough to ensure no performance degradation on the downstream tasks. Finally, we collect the subjective assessment of an expert neurologist, that confirms the high reconstruction quality of BrainCodec in a realistic scenario. The code is available at `https://github.com/IBM/eeg-ieeg-brain-compressor`.

## 1 Introduction

Collecting high signal-to-noise ratio (SNR) data can prove to be a challenging endeavor in many situations, especially when considering human data. However, noisier signals are sometimes adequate to perform the task at hand. Following this principle, different data modalities can be collected from the same source with varying levels of quality. For instance, the electroencephalogram (EEG)

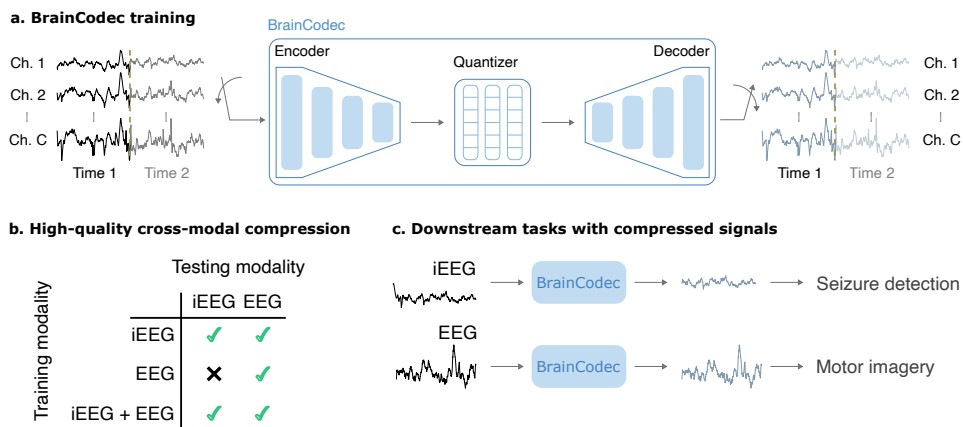

Figure 1: **BrainCodec training and usage**. **a.** BrainCodec can be trained on EEG or iEEG data. **b.** BrainCodec trained on iEEG can compress both iEEG and EEG data, while BrainCodec trained on EEG can only compress other EEG data. **c.** BrainCodec's high-fidelity compressed signals can be used to perform downstream classification on iEEG and EEG data.

is a multi-variate time-series recording of the electrical activity of the brain, where the quality and SNR vary considerably based on the specific recording setup. On the one hand, non-invasive scalp EEG can be easily collected in many environments through the placement of extracranial electrodes on the scalp, but suffers from low spatial and temporal resolutions. The use of ultra long-term non-invasive EEG systems is expected to help improve personalized patient care, for example, by objectively assessing seizure rate, and also macro- and microstructure of sleep (Yilmaz et al., 2024; Casson et al., 2010), which are both treatable risk factors for dementia (Hanke et al., 2022). On the other hand, intracranial EEG (iEEG) collected through invasive intracranial electrodes benefits from a higher SNR and a more direct physical connection to the brain. In particular, iEEG is used by neurologists to delineate the seizure onset zone in patients suffering from pharmacoresistant epilepsy, and deep learning (DL) models have been developed to support them (Kuhlmann et al., 2018; Craik et al., 2019). As such, both EEG and iEEG are essential data sources available to physicians and researchers for the study and treatment of a variety of neurological diseases. However, the storage and transmission of EEG signals is often very costly in these highly critical and sensitive environments.

Data compression is a viable solution to reduce the costs associated with storing and transmitting the large quantities of (i)EEG data that are collected every day. Data compression significantly predates the current rise of DL; therefore, the most successful algorithms in this field do not belong to the class of DL models, i.e., they are not neural compressors. However, recent developments (Dani & Srinivasan, 2021; Webex, 2022; Défossez et al., 2023) have shown that DL models may outperform more traditional approaches. Nonetheless, no compressor has yet found widespread use for EEG and iEEG data.

Data compressors can be divided into two main classes: lossless and lossy. Lossless compression, e.g., lz4, guarantees full signal integrity, at the cost of lower compression ratios. Lossy compression, on the contrary, achieves high compression ratios; however, the fidelity of the reconstruction cannot be guaranteed. As such, no lossy compression algorithm can be used for manual or automated seizure detection without thorough evaluation in real-world scenarios. For time-series, it is usually preferred to use lossy compression, as they often contain information that is noisy and/or not relevant for use by humans. For example, lossy audio compressors are ubiquitous and achieve remarkable size reductions by discarding all content that is mostly unnoticeable by humans. We can do the same with EEG and iEEG. However, the spectral content and the frequencies of interest in audio and EEG are significantly different. Moreover, lossy compression is strongly affected by noise. For these reasons, we expect the behavior of lossy compression on EEG and iEEG to be notably different between the two modalities, and also from audio.

The state-of-the-art neural model (EnCodec; Défossez et al. (2023)) currently available for high-fidelity lossy audio signal compression cannot be used directly on EEG signals, since its design is inadequate in preserving the information content relevant for further analysis. Moreover, the state-

of-the-art neural compressors have not been validated as viable or effective on either EEG or iEEG signals. As we have seen previously, it is well-known that EEG is overall a noisier data source than iEEG, so a model that is successful on one modality is not guaranteed to work on the other. At the same time, it has been reported in other fields, such as natural language processing (Lee et al., 2022; Muennighoff et al., 2023; Gunasekar et al., 2023), that training DL models on cleaner and less noisy data can provide increased performance across the board. Therefore, we also aim to provide concrete guidelines on the performances of neural compressors when transferring between the two data modalities.

To guide our evaluation throughout this work, we propose a set of criteria to define the high-fidelity reconstruction of EEG signals: 1. percentage root-mean-square difference (PRD) lower than 30 as suggested by Higgins et al. (2010); and 2. less than 1% drop in classification performance of downstream tasks such as seizure detection or motor imagery classification for brain–machine interface; and 3. high reconstruction quality as rated by an expert neurologist. Any EEG compressor that fulfills all the criteria outlined above is considered a high-fidelity compressor.

In this work, we introduce BrainCodec (see Figure 1), a high-fidelity quantized autoencoder compressor for EEG and iEEG. BrainCodec has the following features:

- universal compression of both iEEG and EEG using the same model;
- favorable transfer from cleaner iEEG to noisier EEG;
- high-fidelity compression of EEG signals up to a compression ratio as high as 64;
- variable compression ratio, depending on the task requirements.

Remarkably, a BrainCodec model trained on iEEG signals and used to compress EEG signals consistently achieves better performance at high compression ratios compared to a BrainCodec trained on the same EEG modality. This indicates that training with higher quality, higher SNR data generalizes better even across modalities. This is also consistent with previous body of work on natural language, and highlights the advantage of clean signals with high SNR for the pretraining stage. In fact, training DL models on such high-quality signals can improve performance even on their noisier counterparts.

## 2 RELATED WORK

**Neural audio compression.** Neural network models have recently started gaining popularity in the audio compression domain due to their high compression ratios and design flexibility. Most neural compression architectures consist of the encoder-decoder pair of an autoencoder, together with a quantizer to generate discrete representations. VQ-VAE (Van Den Oord et al., 2017) introduced this method—unrelated to the compression objective—by combining variational autoencoders (VAE) with vector quantization (VQ). VQ uses a learnable codebook containing a discrete set of vectors to represent a larger set of input vectors.

GANs have been shown (Kumar et al., 2019; Yamamoto et al., 2020; Kong et al., 2020) to be an effective solution to drive the overall neural compressor towards better representations. MelGAN (Kumar et al., 2019) introduced a multi-scale discriminator that consists of three convolutional discriminators that operate on different scales of the waveform. This architecture restricts the discriminators to specific frequency bands so that they learn features of different scales. The learned features can be used to train a generator by minimizing the distance between features of real and synthetic data.

Combining autoencoder, quantizer, and GAN, SoundStream (Zeghidour et al., 2022) and EnCodec (Défossez et al., 2023) represent the state-of-the-art audio compression models. They are based on a fully convolutional encoder-decoder network with a residual vector quantizer (RVQ) and a convolutional GAN discriminator, operating on the frequencies of interest of the signal. All components are jointly trained end-to-end by minimizing reconstruction, quantization, as well as perceptual adversarial losses.

**Lossless EEG compression.** Standard lossless compression algorithms such as gzip, zstd, and lz4 are used routinely to reduce the storage requirements of large EEG collections. Typical compression ratios for these algorithms on EEG are $1.2\times$ to $1.5\times$. Lossless compression models developed

specifically for EEG are more scarce (Alsenwi et al., 2018; Hadi et al., 2021; Al-Nassrawy et al., 2022) and have not found use in practice.

**Lossy EEG compression.** Lossy compression has not been adopted for either clinical or research use due to the uncertainty about the fidelity of the reconstructed signal. In particular, standard lossy time-series compressed formats such as mp3 have not been developed for EEG and thus have poor performance. Recently, wavelet transform-based techniques using NLSPIHT (Xu et al., 2015), arithmetic coding (AAC; Nguyen et al. (2017)), and artificial neural networks (ANN; Hejrati et al. (2017)), have shown impressive results on EEG, with compression ratios up to $8\times$ and high reconstruction fidelity. Given the importance of EEG signals in a variety of clinical tasks, seizure detection (Nguyen et al., 2018) has been used to validate the reconstruction fidelity of lossy compression as well. Finally, DL models equipped with compressed sensing techniques (CS; Du et al. (2024)) have achieved state-of-the-art compression performance in terms of reconstruction fidelity. Our BrainCodec surpasses existing work by consistently achieving higher compression ratios and higher reconstruction fidelity.

## 3 BRAINCODEC: QUANTIZED AUTOENCODER NEURAL COMPRESSOR

This section presents the main contribution of this work, the neural compressor BrainCodec.

First, we outline a typical use case for BrainCodec. Consider an EEG signal $X \in \mathbb{R}^{C \times T}$ having $C$ channels and a duration $T = d \times f_s$ of $d$ seconds at a sampling frequency of $f_s$. First, each channel of the signal is fed separately to the BrainCodec encoder, which outputs a compressed representation for the given channel. The compressed signal can now be transmitted and stored at a fraction of the cost of the original signal. The BrainCodec decoder reconstructs the original EEG signal from the compressed representation, preserving the relevant information content and producing a high-fidelity result.

We now focus on the design of our neural compressor model, specifically adapted to EEG signals. We adopt the basic quantized autoencoder design of SoundStream (Zeghidour et al., 2022) and EnCodec (Défossez et al., 2023), and tailor it to the EEG use case by modifying the loss function and the parameters of the architecture. The compressor consists of three components: an encoder, a quantizer, and a decoder. The encoder maps the EEG signal to a latent representation. The quantizer compresses this latent representation to a quantized representation using residual vector quantization (RVQ). Finally, the decoder reconstructs the signal from the RVQ output. We design the compressor to achieve high compression ratios while preserving the information content needed to perform classification on the signal. The model is trained end-to-end together with a discriminator that learns multi-scale features of the input data. We apply multiple losses over both the time and frequency domain to capture different properties of the signal. This allows our compressor to be used in end-to-end classification pipelines as we show in our seizure detection and motor imagery results, providing significant storage savings.

**Encoder and decoder.** First, we divide the EEG signal ($X$) channel-wise into short patches $\boldsymbol{x}_{i,j} \in \mathbb{R}^W$, $i \in 1, \ldots, C$, $j \in 1, \ldots, \frac{T}{W} = T_W$, with the patch size $W$ in the order of a few seconds. In particular, we choose $W$ to be 4 seconds long at the signal sampling frequency. The patches serve as the input to the encoder. The encoder is composed of a 2D convolutional layer with 1 input channel, $F$ output channels, and a kernel size of $(3, 1)$. This ensures that each channel is treated separately, and allows the encoder to work on signals with varying number of channels. The initial layer is followed by $N$ encoder blocks, where $N$ depends on the compression ratio. Each encoder block comprises a residual block as well as a 2D convolutional layer with a kernel size of $(K, 1)$ and a stride of $(S, 1)$ for down-sampling, with $K$ twice the size of $S$. The number of channels is doubled with each down-sampling layer until there are 256. The encoder blocks are followed by a final 2D convolutional layer with $D$ output channels and a kernel size of $(3, 1)$. We choose $F = 16, S = 2, D = 64$, and ELU as the activation function. The encoder finally outputs a latent representation $\boldsymbol{z}_{i,j}$ for each input patch. The decoder mirrors the encoder, replacing strided convolutions with transposed convolutions.

**Quantizer.** We quantize the latent representation ($\boldsymbol{z}$) to a compressed representation ($\boldsymbol{z}_q$) through RVQ. A codebook stores a finite set of learnable prototype vectors that are used to represent a larger set of input vectors. When compressing $\boldsymbol{z}$, the quantizer maps the input vector to the closest pro-

totype vector in the codebook. RVQ (Zeghidour et al., 2022) extends this principle to an iterative process. After mapping an input vector onto a prototype vector, RVQ computes the residual and maps it to another prototype vector from a second codebook. By repeating this process, the sum of prototype vectors converges to the original vector. As suggested in previous literature (Zeghidour et al., 2022; Dhariwal et al., 2020), the selected prototype vectors are updated using an exponential moving average with a decay of 0.99, whereby the entries that have not been assigned to an input vector are replaced by a randomly sampled input vector. To improve the initialization of the codebooks, we apply k-means clustering to the first training batch and use the centroids as prototype vectors. During training, we use a straight-through estimator (Bengio et al., 2013) to pass the gradients from the decoder to the encoder. At the same time, we compute the MSE between $z$ and $z_q$ and add it to the overall loss. For all our models, we use $4$ codebooks each of size $256$ (i.e., the storage of an index that refers to a codebook entry requires $8$ bits).

**Discriminator (GAN component).** During training, we use a multi-scale STFT-based (MS-STFT) discriminator (Défossez et al., 2023) to improve the reconstruction of high frequencies. The MS-STFT discriminator is composed of 5 convolutional discriminators operating on different scales of the complex-valued spectrogram. Each discriminator is composed of an initial 2D convolutional layer with 64 output channels and a kernel size of $(3, 3)$. The initial layer is followed by 3 convolutional layers with increasing dilation in the time dimension of 1, 2, and 4, a kernel size of $(3, 3)$, and a stride of $(1, 2)$. Another 2D convolutional layer with a kernel size of $(3, 3)$ is followed by the final 2D convolutional layer with 1 output channel and a kernel size of $(3, 3)$. We use $(2048, 1024, 512, 256, 128)$ as STFT window lengths and LeakyReLU as the activation function.

# 4 TRAINING SETUP

We train BrainCodec following the schema of SoundStream (Zeghidour et al., 2022). To help guide BrainCodec towards reconstructed signals apt for downstream classification, we also add a new loss based on the line length, which is widely considered to be a useful feature for EEG classification (Schindler et al., 2001; Guo et al., 2010; Burrello et al., 2021). We observe that the GAN has a significant effect on the reconstruction fidelity of the signal (see App. C.3); therefore, we train one model with GAN and one without. To avoid the risk of cross contamination between the subjects, we always train the model on one single subject, and test the reconstruction on the remaining subjects. To provide a direct baseline for the performance of BrainCodec in the seizure detection task, we also train a standard EEGWaveNet on the original signal, and then test it using the reconstructed signal.

**Validation.** We also provide objective metrics of the reconstruction performance of our BrainCodec in two downstream tasks: the iEEG seizure detection task, and the EEG motor imagery classification task. In particular, for iEEG we train a standard EEGWaveNet (Thuwajit et al., 2022) classifier on the original signal, and evaluate its performance on the reconstructed signal. For EEG, we train an MI-BMInet (Wang et al., 2024b) classifier on the original signal, and evaluate its performance on the reconstructed signal. For more information on the detailed training regime of BrainCodec refer to App. A.

**Optimizer and setup.** We apply the 1-cycle learning rate policy, with a learning rate varying from $10^{-5}$ to $10^{-4}$ for the generator and from $10^{-7}$ to $10^{-6}$ for the discriminator. We further use the weights $\lambda_t = 1$ and $\lambda_q = 1$ for the base model. For the GAN model, we choose $\lambda_t = 0.1$, $\lambda_s = 1$, $\lambda_l = 0.1$, $\lambda_f = 3$, $\lambda_g = 3$, and $\lambda_q = 1$. The model is trained with full fp32 precision.

## 4.1 DATASETS

**SWEC iEEG (Burrello et al., 2019).** This short-term iEEG dataset contains 15 subjects, 14 hours of recording, and 104 ictal events. The iEEG signals were recorded intracranially with a sampling rate of either $512\,\text{Hz}$ or $1024\,\text{Hz}$. The signals were median-referenced and band-pass filtered between 0.5 and $120\,\text{Hz}$ using a fourth-order Butterworth filter, both in a forward and backward pass. All the recordings were inspected by an expert neurologist for identification of seizure onsets and offsets, and to remove channels corrupted by artifacts.

**Multi-center (MC) iEEG (Li et al., 2021).** This iEEG dataset contains iEEG signals around ictal events for 91 subjects for a total of 462 events, with a sampling rate up to 1000 Hz. The onset and offset times of the seizures are included in the dataset.

**Brain Treebank iEEG (Wang et al., 2024a).** This iEEG dataset contains 10 subjects for a total of 43 hours. The subjects have an average of 168 electrodes with a sampling rate of 2048 Hz. As this is not an ictal dataset, there are no labels for seizures.

**CHB-MIT (Shoeb, 2010; 2009).** This EEG dataset contains a total of 24 subjects, 983 hours of recording, and 198 seizures. All signals have a sampling rate of 256 Hz. The onset and offset times of the seizures are included in the dataset.

**BONN (Andrzejak et al., 2001).** This EEG dataset is composed of five sets (Z, O, N, F, S), each containing approximately 40 minutes of single-channel EEG. All signals have a sampling rate of 173.61 Hz, and they were band-pass filtered between 0.5 Hz and 40 Hz. We use all sets for evaluating the compression only.

**BCI Competition IV-2a (Tangermann et al., 2012).** This EEG dataset contains a total of 9 subjects. The data was collected for the purpose of 4-class motor imagery classification in brain-machine interfaces, so we have useful labels for downstream classification. All signals have a sampling rate of 250 Hz, and they were band-pass filtered between 0.5 Hz and 120 Hz with an additional notch filter at 50 Hz to remove line noise.

## 4.2 BASELINES

We compare the performance of BrainCodec with multiple state-of-the-art lossy compression algorithms that have been developed for EEG. We benchmark against the following methods: NL-SPIHT (Xu et al., 2015), a classical algorithm; ANN (Hejrati et al., 2017), a deep-learning model; AAC (Nguyen et al., 2017), a wavelet-based model with adaptive arithmetic coding; and CS (Du et al., 2024), a compressed-sensing based deep learning model.

To showcase the adaptability of BrainCodec, we test it on three iEEG datasets (SWEC, MC, Treebank), which have a high SNR, and also on multiple EEG datasets (CHB-MIT, BONN, and BCI IV-2a), which are noisier. Finally, we evaluate the generalization capabilities of BrainCodec across datasets and modalities. The full comparison with all the baselines is shown in App. D.

## 5 RESULTS

To evaluate the compression performance in terms of reconstruction fidelity, we report the percentage root-mean-square distortion (PRD):

$$\text{PRD} = \frac{\|\boldsymbol{x} - \hat{\boldsymbol{x}}\|_2}{\|\boldsymbol{x}\|_2} \cdot 100, \tag{1}$$

which represents the relative $L_2$-distance between the original and the reconstructed signal. In order to have a comprehensive outlook, we train multiple models at varying compression ratios.

## 5.1 BRAINCODEC CROSS-MODAL COMPRESSION

Reports from other fields, especially natural language processing, have shown (Lee et al., 2022; Muennighoff et al., 2023; Gunasekar et al., 2023) that training with higher-quality data often yields better performance than simply training with more, and more similar, data. We aim to characterise this phenomenon for human iEEG and EEG, where, due to its higher SNR, we consider iEEG to be of higher quality than EEG in the signal processing domain.

To evaluate the role of high-SNR data in human EEG signals, we train an instance of BrainCodec on the SWEC iEEG dataset and use it to compress EEG signals. Figure 2a shows the median PRD across all tested EEG datasets when training BrainCodec on iEEG or on EEG. The EEG-trained compressor performs slightly better at lower compression ratios, while the iEEG-trained model becomes competitive and even achieves higher performance at higher compression ratios. Aside from the pure compression advantage, variational autoencoders have been shown to produce better representations when the information bottleneck becomes more restrictive and learning pressure

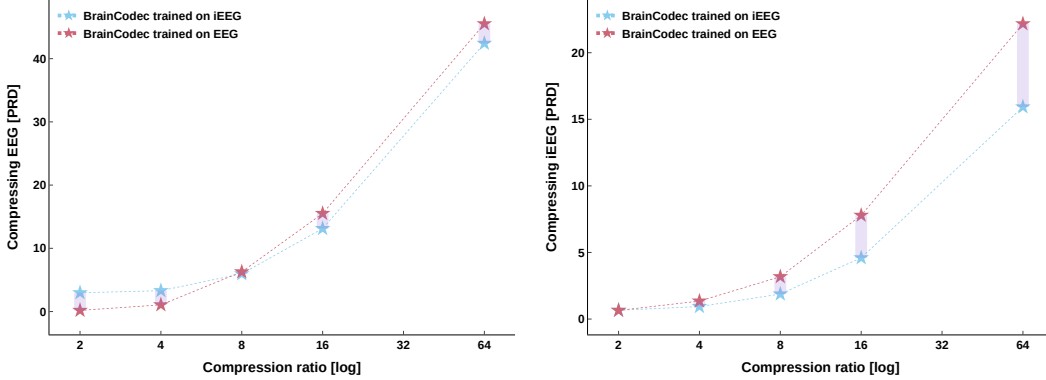

(a) Testing cross-modal BrainCodec on scalp EEG.

(b) Testing cross-modal BrainCodec on intracranial EEG (iEEG).

Figure 2: **Cross-modality signal reconstruction fidelity of BrainCodec.** BrainCodec trained on iEEG (higher SNR) always performs better at moderate and higher compression ratios compared to BrainCodec trained on scalp EEG (lower SNR), even when compressing EEG.

increases (Burgess et al., 2018). In this region of interest, high-SNR iEEG also becomes a better data source than low-SNR EEG. We hypothesize that the increased performance of the cross-modal BrainCodec (i.e., from iEEG to EEG) at above moderate compression ratios can be traced back to the effect of the noise content of the lower SNR EEG with respect to the higher SNR iEEG (see App. C.1 for more details).

Conversely, we train an instance of BrainCodec on the EEG CHB-MIT dataset and use it to compress the SWEC iEEG dataset. As expected, Figure 2b shows that the EEG to iEEG cross-modal BrainCodec performs worse than the within-modality model, indicating that a lower-quality signal is less effective at generalizing than a higher-quality one.

In summary, by characterizing the behavior of BrainCodec when trained across modality, we have corroborated previous reports about data quality from natural language processing. In particular, we have shown that training with higher quality data sources yields improved performance even when generalizing to lower quality ones. The converse is, expectedly, not true. App. C.1 provides more details.

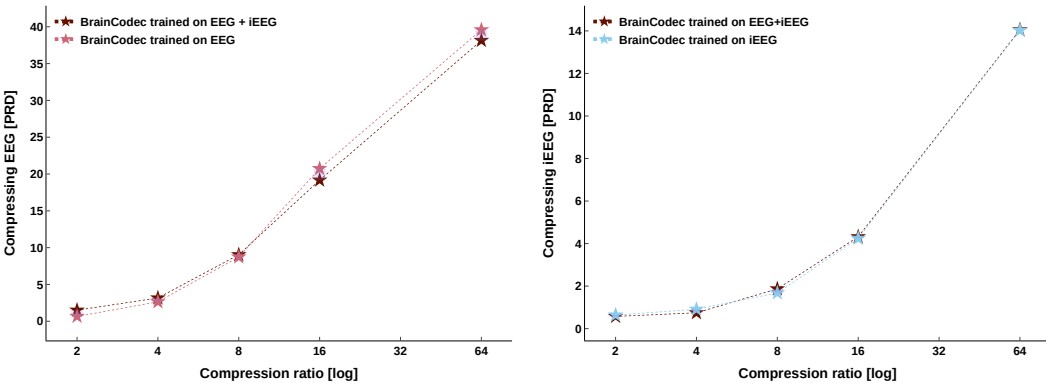

(a) Testing mixed-modal BrainCodec on scalp EEG.

(b) Testing mixed-modal BrainCodec on intracranial EEG (iEEG).

Figure 3: **Mixed-modality signal reconstruction fidelity of BrainCodec.** BrainCodec trained on both intracranial EEG and scalp EEG maintains the reconstruction fidelity of an iEEG-model when compressing iEEG. At the same time, it improves performance at high compression ratios with respect to a scalp EEG-trained model compressing scalp EEG.

## 5.2 BRAINCODEC MIXED-MODAL COMPRESSION

Given the promising results shown by BrainCodec when transferring across modalities, we now investigate the performance of our neural compressor when trained with both modalities — scalp EEG and intracranial EEG — at the same time.

To evaluate the effect of mixed-modal compression, we train BrainCodec on both the SWEC iEEG dataset and the CHB EEG dataset, for the same overall amount of data as the previous models to keep the evaluation balanced. Figure 3a shows that the median PRD of our mixed model is notably superior to the EEG-only model when compressing EEG signals, indicating that the performance benefits of iEEG training have transferred successfully. In line with the previous cross-modal results, this improvement is more marked at high compression ratios. At the same time, the mixed model also performs on par with the EEG-only model at lower compression ratios, mitigating the drawback we had reported in the previous cross-modal results.

On the other hand, we also test mixed BrainCodec on iEEG recordings. In this case, we do not observe any benefit of mixed-modal training in reconstruction fidelity. However, we also do not observe any notable reduction in performance.

## 5.3 BRAINCODEC COMPRESSION PERFORMANCE

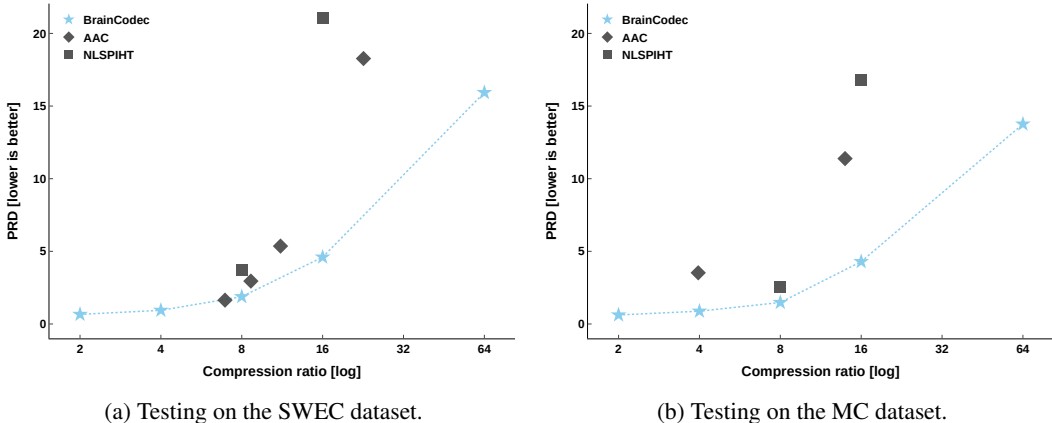

(a) Testing on the SWEC dataset.      (b) Testing on the MC dataset.

Figure 4: **Within-modality signal reconstruction fidelity of BrainCodec on intracranial EEG (iEEG).** BrainCodec trained only on the SWEC dataset shows increased performance across the board both on the SWEC and MC dataset, and also reaches higher compression ratios while maintaining a moderate PRD.

Next, we train BrainCodec exclusively on iEEG data to compress other iEEG datasets. Likewise, we train BrainCodec on EEG data to compress other EEG datasets, ensuring compression is performed within the same modality. We then compare the results with baseline methods.

We compress iEEG and EEG signals with a varying compression ratios from $2\times$ to $64\times$. We compare BrainCodec with multiple state-of-the-art methods found in the literature, both neural network-based and classical. The full set of results can be found in App. D.

We choose two iEEG datasets to test the performance of BrainCodec with iEEG compression: SWEC and MC. We train all models that require training on a subset of the respective dataset and test on the remaining part. The results of the best performing BrainCodec model on the SWEC dataset are shown in Figure 4a. At lower compression ratios, BrainCodec improves on the PRD compared to both AAC and NLSPIHT, but the baselines remain competitive. At higher compression ratios, however, BrainCodec is notably better, with a lower PRD at almost twice the compression. In particular, BrainCodec remains in the high-fidelity regime even with a $64\times$ compression ratio. Figure 4b paints a similar picture for the iEEG MC dataset. BrainCodec surpasses all baselines at lower compression ratios and reaches much higher ratios overall. Performance is high when trained with the iEEG SWEC dataset, indicating that BrainCodec generalizes well across iEEG datasets within the same modality.

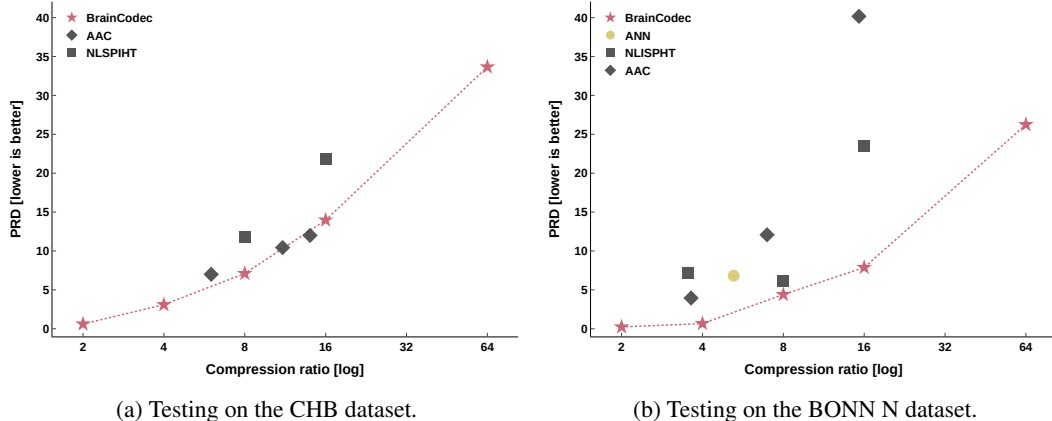

(a) Testing on the CHB dataset.            (b) Testing on the BONN N dataset.

Figure 5: **Within-modality signal reconstruction fidelity of BrainCodec on scalp EEG.** Brain-Codec trained only on the CHB dataset, shows increased performance across the board both on the CHB and the BONN dataset, and also reaches higher compression ratios while maintaining a moderate PRD.

We test BrainCodec's performance on EEG data with three EEG datasets, CHB-MIT, BONN, and BCI IV-2a. Figure 5a shows that BrainCodec improves on the PRD compared to all other methods on the EEG CHB-MIT dataset. Moreover, it can also achieve much higher compression ratios. The model BrainCodec Base is trained on a subset of the CHB-MIT dataset and tested on the remaining part, to ensure no contamination between the subjects. The same trend holds for the EEG BONN N dataset (Figure 5b). The BONN dataset is too small to both train and test the two neural network-based approaches, BrainCodec and ANN. Therefore, we first train them on other EEG datasets and then test them on BONN. As seen previously with iEEG, the performance of BrainCodec is notably better than the baselines even when trained across EEG datasets within the same modality. The results of BrainCodec on BCI can be found in Figure A6.

Finally, we also provide a subjective evaluation by an expert neurologist of the SWEC iEEG and the CHB-MIT EEG datasets as compressed by BrainCodec. The evaluation subjectively confirms that BrainCodec achieves a high-fidelity reconstruction both on iEEG and EEG. More details can be found in App. E.

We find that BrainCodec shows superior performance compared to all baselines across our suite of benchmarks both on iEEG and EEG, as can be seen in App. D. Overall, these results indicate that BrainCodec is an efficient iEEG and EEG compressor, both within and across datasets.

## 5.4 DOWNSTREAM CLASSIFICATION TASKS

As another objective measurement of reconstruction quality, we validate the reconstruction fidelity on two downstream classification tasks: the iEEG seizure detection task, and the EEG motor imagery task.

First, we evaluate BrainCodec on the iEEG seizure detection task, by testing a subject-specific EEG-WaveNet (Thuwajit et al., 2022) seizure classifier on the reconstructed signal. We test the EEG-WaveNet across all subjects with a leave-one-out cross-validation scheme, training for each subject on all seizures but one and testing on the remaining seizure. Table 1 provides the compression ratio, PRD, and resulting F1-score for the signals reconstructed by BrainCodec. Even at a $64\times$ compression, there is no loss of performance in the seizure detection task. Moreover, the F1-score of the BrainCodec GAN model degrades only by 8% at $256\times$ compression, likely due to its better reconstruction of the higher frequencies with respect to the Base model (see App. C.3). Thus, the relevant information content is preserved by BrainCodec while providing significant storage and transmission savings.

Second, we evaluate BrainCodec on the EEG motor imagery task using the MI-BMInet (Wang et al., 2024b) classifier. The training and testing setup is analogous to the seizure detection task. Specif-

| | Original iEEG | BrainCodec Base | | | | BrainCodec GAN | | | |
|---|---|---|---|---|---|---|---|---|---|
| CR ↑ | n.a. | 8 | 64 | 128 | 256 | 8 | 64 | 128 | 256 |
| F1 ↑ | 0.79 | 0.79 | 0.78 | 0.72 | 0.62 | 0.78 | 0.78 | 0.72 | 0.72 |
| PRD ↓ | n.a | 1.88 | 15.7 | 26.3 | 40.7 | 2.37 | 17.6 | 30.0 | 47.4 |

Table 1: **Performance of the BrainCodec compressor on the iEEG seizure detection task.** The PRD remains low ($<30$) even at $64\times$ compression, and the F1-score remains high when a standard subject-dependent EEGWaveNet makes inferences with the reconstructed data instead of the original iEEG ($<1\%$ drop).

ically, we train the compressor on the first subject of the BCI dataset, and then report the average test accuracy across all remaining 8 subjects. BrainCodec shows high-fidelity reconstructions up to a $4\times$ compression ratio, and maintains a useful classification performance up to $16\times$. This rate of compression is expectedly lower than iEEG, as EEG has intrinsically lower SNR and is thus less amenable to compression.

| | Original EEG | BrainCodec Base | | | | BrainCodec GAN | | | |
|---|---|---|---|---|---|---|---|---|---|
| CR ↑ | n.a. | 4 | 8 | 16 | 64 | 4 | 8 | 16 | 64 |
| Acc. ↑ | 77% | 76% | 73% | 72% | 63% | 74% | 73% | 73% | 57% |
| PRD ↓ | n.a | 3.44 | 10.4 | 29.0 | 45.5 | 8.04 | 39.2 | 45.3 | 73.8 |

Table 2: **Performance of the BrainCodec compressor on the EEG motor imagery task (BCI IV-2a).** Subject 1 has been excluded from the evaluation, as it has been used to train BrainCodec. The PRD remains moderate even at high compression ratios, and the accuracy remains high when a standard subject-dependent MI-BMInet makes inferences with the reconstructed data instead of the original EEG.

Overall, we confirm that BrainCodec can achieve high compression ratios both on iEEG and EEG signals without notably impacting downstream task performance. Therefore, we have sufficiently characterized and validated BrainCodec as an efficient iEEG and EEG compressor.

## 6 DISCUSSION

In this work, we present BrainCodec, a high-fidelity neural compressor for EEG and iEEG signals. BrainCodec is effective across both modalities and a variety of datasets, indicating that it can successfully replace existing methods and be introduced in any EEG processing pipeline. Compression by BrainCodec up to $64\times$ does not affect downstream seizure detection performance as evaluated both by human experts and deep learning models. We also observe that BrainCodec performs better when trained with high SNR iEEG. This performance increase is maintained when compressing the noisier EEG signal, compared to the same model trained on the very same EEG modality. We therefore highlight the importance of training deep learning models on high-quality signals also in the medical domain.

Overall, BrainCodec is an immediate compression replacement for many EEG and iEEG applications, enabling transmission and storage cost savings in critical clinical environments. We expect the adoption of BrainCodec to increase the feasibility of long-term recordings and wearable devices.

Further work is necessary to assess whether the intermediate representations of BrainCodec can be directly used by other deep learning models, to increase performance and provide an additional speed-up. Architectural changes to BrainCodec could be made, for example by utilising decoder models specifically developed for biosignals (Zhang et al., 2023). Moreover, the RVQ quantization schema of BrainCodec is known to be not necessarily codebook efficient, and improvements are already being developed in the field (Kumar et al., 2024). On this front, more venues are being explored to replace RVQ with another quantization schema, such as Finite Scalar Quantization (Mentzer et al., 2024).

ACKNOWLEDGMENTS

This work is supported by the Swiss National Science foundation (SNF), grant no. 200800.

REPRODUCIBILITY

The training setup, losses, and optimizers are described in detail in Sec. 4 and App. A. All datasets used are listed and are publicly available for download, and the data selection for the generation of the results is also explained in detail. Finally, the code is available at `https://github.com/IBM/eeg-ieeg-brain-compressor`.

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

# A  NEURAL COMPRESSOR DETAILS

## A.1  TRAINING OBJECTIVE

We apply multiple losses to capture properties of both the time and frequency domain. The reconstruction loss over the time domain $\ell_t$ measures the $L_1$-distance between the original signal ($\boldsymbol{x}$) and the reconstructed signal ($\hat{\boldsymbol{x}}$):

$$\ell_t(\boldsymbol{x}, \hat{\boldsymbol{x}}) = \|\boldsymbol{x} - \hat{\boldsymbol{x}}\|_1. \tag{2}$$

The reconstruction loss over the frequency domain $\ell_s$ consists of a linear combination between $L_1$- and $L_2$-distance computed over different scales of the spectrogram. In particular,

$$\ell_s(\boldsymbol{x}, \hat{\boldsymbol{x}}) = \sum_{i \in I} \frac{1}{|S_i(\boldsymbol{x})|} (\|S_i(\boldsymbol{x}) - S_i(\hat{\boldsymbol{x}})\|_1 + \alpha \|S_i(\boldsymbol{x}) - S_i(\hat{\boldsymbol{x}})\|_2), \tag{3}$$

where $S_i(\cdot)$ is the output of an STFT with a window size of $2^i$ and a hop length of $2^{i-2}$. We choose $I = \{5, 6, ..., 11\}$ and $\alpha = 1$.

We further introduce a new relative line length loss that accounts for large differences in amplitude. Often the amplitude of a seizure sample is much larger than the amplitude of a non-seizure sample. When reconstructing such a seizure sample, a relatively small deviation can cause a large $L_1$- or $L_2$-distance. In these cases, the weight updates are mainly driven by seizure samples, since they contribute much more to the overall loss than non-seizure samples. To address this imbalance, we compute the relative difference in line length of the original and reconstructed signal, which is independent of the amplitude. Formally,

$$\ell_l(\boldsymbol{x}, \hat{\boldsymbol{x}}) = \sum_{w=0}^{W-1} \sum_{t=1}^{T} \frac{||\boldsymbol{x}_{t+wS} - \boldsymbol{x}_{t-1+wS}| - |\hat{\boldsymbol{x}}_{t+wS} - \hat{\boldsymbol{x}}_{t-1+wS}||}{|\boldsymbol{x}_{t+wS} - \boldsymbol{x}_{t-1+wS}|}, \tag{4}$$

with $T_W$ windows of size $W$ and stride $S$. We choose $T = 128$ and $S = 64$.

To improve the reconstruction of high frequencies, we apply perceptual losses that are based on the MS-STFT discriminator. The discriminator network is trained by minimizing the adversarial loss of the generator $\ell_g$ as well as the adversarial loss of the discriminator $\ell_d$. The two losses are defined as follows:

$$\ell_g(\hat{\boldsymbol{x}}) = \sum_{k=1}^{K} \text{mean}(\max(0, 1 - D_k(\hat{\boldsymbol{x}}))), \tag{5}$$

$$\ell_d(\boldsymbol{x}, \hat{\boldsymbol{x}}) = \sum_{k=1}^{K} \text{mean}(\max(0, 1 - D_k(\boldsymbol{x}))) + \text{mean}(\max(0, 1 + D_k(\hat{\boldsymbol{x}}))), \tag{6}$$

where $K$ is the number of discriminators and $D_k$ is the output of the respective discriminator.

At the same time, we exploit the multi-scale features learned by the discriminators to compute a feature loss,

$$\ell_f(\boldsymbol{x}, \hat{\boldsymbol{x}}) = \sum_{k=1}^{K} \sum_{l=1}^{L} \frac{\|D_k^l(\boldsymbol{x}) - D_k^l(\hat{\boldsymbol{x}})\|_1}{\|D_k^l(\boldsymbol{x})\|_1}, \tag{7}$$

where $K$ is the number of discriminators, $L$ is the number of layers of a discriminator, and $D_k^l$ is the output of the respective layer.

We further add a quantization loss $\ell_q$ that computes the MSE between the latent representation ($\mathbf{z}$) and its quantized version ($\mathbf{z}_q$):

$$\ell_q(\mathbf{z}, \mathbf{z}_q) = \sum_{c=1}^{C} \|\mathbf{z}_c - (\mathbf{z}_q)_c\|_2^2, \tag{8}$$

where $C$ is the number of codebooks. Since the codebook entries are updated using an exponential moving average, we compute the gradients only with respect to the latent representation ($\mathbf{z}$).

With the exception of $\ell_d$, all losses are added up to the overall generator loss,

$$L_G = \lambda_t \cdot \ell_t + \lambda_s \cdot \ell_s + \lambda_l \cdot \ell_l + \lambda_f \cdot \ell_f + \lambda_g \cdot \ell_g + \lambda_q \cdot \ell_q, \tag{9}$$

where $\lambda_t$, $\lambda_s$, $\lambda_l$, $\lambda_f$, $\lambda_g$, and $\lambda_q$ are the weights to balance between the loss terms. We further use a loss balancer introduced by Défossez et al. (2023) to deal with the varying scale of the gradients and to stabilize training between generator and discriminator.

## A.2 TRAINING SCHEMA

For the iEEG compression task, we use the recordings from subject ID1, which have 47 channels for a total duration of 5604 seconds, and contain 13 seizures. We randomly sample 80% of that data for training and use the remaining 20% for validation. For the seizure detection task, we exclude ID1 from the dataset and use the recordings from the remaining 15 subjects of the database.

For the leave-one-out cross-validation, we train $N$ classifiers per subject, where $N$ is the number of seizures of the respective subject. We use the original recordings of $N-1$ seizures for training, and the reconstructed recording of the remaining seizure for validation. The performance for a subject is the average across $N$ trials. The final performance is the average across all subjects, weighted by the number of seizures. All classifiers are trained for 50 epochs with a learning rate of $3 \cdot 10^{-4}$. We use samples of 5 seconds and a batch size of 128.

## B ABLATIONS

### B.1 REFERENCE SCHEMA

The reference schema is an often underappreciated aspect of data collection. To evaluate how BrainCodec's performance varies with different references, we use the unprocessed Brain Treebank dataset and reference it in post-processing. Laplacian refers to referencing each electrode with the mean within its electrode group.

| **Reference** | CR ↑ | PRD ↓ | PRD-spec ↓ | RMSE ↓ | SNR ↑ | PSNR ↑ |
|---|---|---|---|---|---|---|
| None | 64 | 15.21 | 10.27 | 10400 | 18.49 | 27.81 |
| Median | 64 | 14.67 | 9.89 | 8440 | 18.36 | 27.95 |
| Bipolar | 64 | 14.36 | 9.19 | 8439 | 18.89 | 28.38 |
| Laplacian | 64 | 16.56 | 11.75 | 1160 | 18.96 | 27.53 |

Table A1: Ablation of the reference schema.

Table A1 shows that BrainCodec is robust to the reference schema. Moreover, variations can most likely be explained with the changes in SNR of the dataset due to the reference itself.

### B.2 ENCODER

Out of the three components of BrainCodec the Encoder is directly in contact with the raw data, such that its parameters have a noticeable effect on all downstream components. In particular, we adopt the same choices as SoundStream (Zeghidour et al., 2022) except for the initial kernel size, which needs to be modulated based on the specific characteristics of EEG and iEEG.

We evaluate the effect of the initial kernel size of the Encoder component on the compression performance in Table A2. We use the Base model at 64× compression trained on the iEEG SWEC dataset and tested on the same dataset.

| Kernel size | PRD ↓ | PRD-spec ↓ | RMSE ↓ | SNR ↑ | PSNR ↑ |
|---|---|---|---|---|---|
| 3 | 15.74 | 10.93 | 16.14 | 17.96 | 36.73 |
| 5 | 15.94 | 11 | 16.37 | 17.94 | 33.86 |
| 7 | 15.91 | 10.91 | 16.27 | 17.83 | 33.87 |

Table A2: Ablation of the Encoder's initial kernel size.

A value of 7 was chosen in the original architecture; however, in accordance with the decreased sample rate of our data, our choice of a kernel size of 3 yields increased performance with respect to all alternatives.

## B.3 QUANTIZER

We evaluate the effects of the parameters of the Quantizer component on the compression performance. In particular, we vary the number of residuals in Table A3, and doing so we also change the compression ratio. In contrast, for our models we choose to change the Encoder's framerate to adjust the compression ratio, as it yields superior performance.

| Residuals | CR ↑ | PRD ↓ | PRD-spec ↓ | RMSE ↓ | SNR ↑ | PSNR ↑ |
|---|---|---|---|---|---|---|
| 1 | 256 | 38.36 | 31.75 | 36.03 | 9.67 | 25.72 |
| 4 | 64 | 15.74 | 10.93 | 16.14 | 17.96 | 36.73 |
| 8 | 32 | 10.26 | 6.08 | 10.95 | 22.04 | 37.94 |
| 16 | 16 | 8.74 | 4.32 | 9.44 | 23.76 | 39.57 |

Table A3: Ablation of the Quantizer's number of residuals.

The compression ratio is also affected, albeit in a minor way, by the size of the codebook. Therefore, we also perform an ablation on the codebook size, as shown in Table A4.

| Codebook size | CR ↑ | PRD ↓ | PRD-spec ↓ | RMSE ↓ | SNR ↑ | PSNR ↑ |
|---|---|---|---|---|---|---|
| 64 | 85.33 | 21.09 | 17.82 | 21.58 | 15.47 | 31.26 |
| 128 | 73.14 | 18.16 | 14.40 | 18.70 | 16.79 | 32.63 |
| 256 | 64 | 15.74 | 10.93 | 16.14 | 17.96 | 36.73 |
| 512 | 56.89 | 13.95 | 9.02 | 14.34 | 19.02 | 35.02 |
| 1024 | 51.20 | 12.74 | 7.85 | 13.09 | 19.77 | 35.83 |

Table A4: Ablation of the Quantizer's codebook size.

Our choice of 256 represents an effective compromise between reconstruction fidelity and compression, requiring only $\log_2 256 = 8$ bits to store.

## B.4 LINE LENGTH LOSS

One of the major improvements we adopt over the original SoundStream to better suit our data's characteristics is an additional line length loss term. This term is informed by the existing consensus that line length can be a useful indicator of seizures and other pathological states. Considering that most collected iEEG data is pathological by nature, considering its line length can yield effective improvement in compression and in downstream classification tasks. Therefore, we evaluate the effect of our additional line length loss on the reconstruction fidelity. Table A5 shows the performance of a Base model trained both with and without line length loss. Results show that there is a net improvement both in PRD and SNR.

| Line length | CR ↑ | PRD ↓ | PRD-spec ↓ | RMSE ↓ | SNR ↑ | PSNR ↑ |
|---|---|---|---|---|---|---|
| Yes | 64 | 15.74 | 10.93 | 16.14 | 17.96 | 36.73 |
| No | 64 | 15.95 | 10.87 | 16.26 | 17.93 | 33.84 |

Table A5: Ablation of the line length loss.

However, a larger effect can be appreciated in the spectrogram of signals reconstructed by the GAN model.

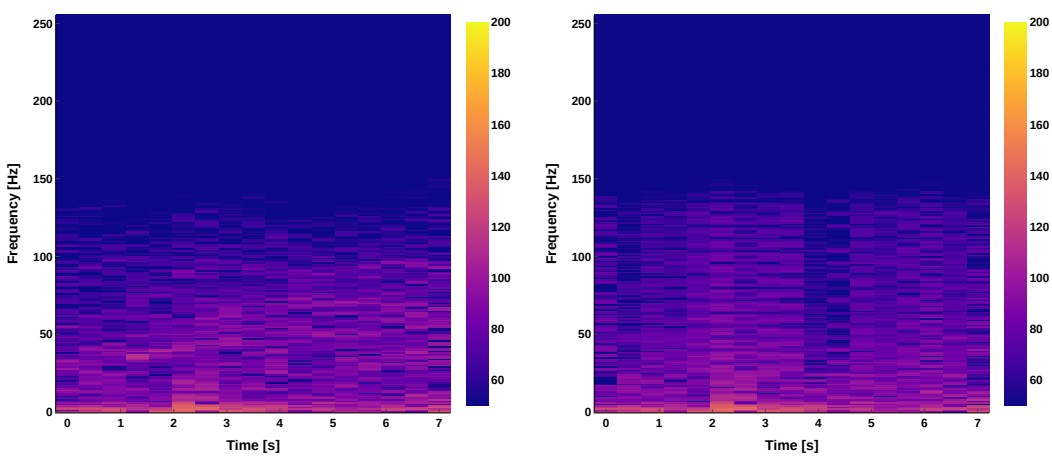

(a) Ictal sample reconstructed with line length loss.  (b) Ictal sample reconstructed without line length loss.

Figure A1: Difference in reconstruction fidelity of GAN BrainCodec trained with and without line length loss. The performance on high frequencies notably improves with the line length loss.

Figure A1 shows that the line length loss improves the reconstruction fidelity of high frequencies, especially on ictal samples. In particular, the model without line length loss shows clear artifacting above 40 Hz.

## B.5 NUMBER OF TRAINING SUBJECTS

As detailed in Section A.2 we only use one subject to train each compression model, which showcases our model cross-subject generalisability. In Table A6 we report that, while the performance of BrainCodec increases as the number of training subjects increases, there are diminishing returns. In particular, we train two additional models on subjects ID01, ID03, and ID01, ID03, ID04, ID05, and always test on all subjects except ID01, ID03, ID04, ID05. Therefore, our model presents the best compromise between training data, training complexity, and reconstruction fidelity.

| Training patients | CR ↑ | PRD ↓ | PRD-spec ↓ | RMSE ↓ | SNR ↑ | PSNR ↑ |
|---|---|---|---|---|---|---|
| 1 | 64 | 15.56 | 10.73 | 16.83 | 18.26 | 34.41 |
| 2 | 64 | 14.31 | 7.36 | 13.56 | 18.58 | 34.97 |
| 4 | 64 | 13.96 | 7.37 | 13.45 | 18.79 | 35.2 |

Table A6: Ablation of the number of training subjects.

## C  ADDITIONAL RESULTS

### C.1  CROSS-DATASET AND CROSS-MODALITY COMPRESSION

We test the performance of BrainCodec both in a cross-dataset and a cross-modality scenario. We evaluate the iEEG to EEG and EEG to iEEG generalization by training BrainCodec on the SWEC, CHB-MIT, and BCI IV-2a datasets and using those same models to compress the SWEC, MC, CHB-MIT, BCI IV-2a, and finally all the subsets of the BONN dataset.

On the iEEG SWEC dataset (see Figure A2) the neural compressor trained on the same SWEC dataset performs notably better for compression ratios above 16, but the CHB model is still competitive with the other baselines even across modalities. For lower compression ratios, the CHB-MIT and the SWEC models both perform on par. Finally, the BCI model is never competitive, likely because the BCI dataset does not contain any seizures and the model is thus unable to compress them well.

On the iEEG MC dataset (see Figure A3) the neural compressor trained on SWEC surpasses all baselines that are trained on the same MC dataset.

On the iEEG Brain Treebank dataset (see Figure A4) the neural compressor trained on SWEC performs at a comparable level to testing on SWEC itself. Due to the challenging nature of training GAN models, some of the results obtained by training on the combined SWEC and TreeBank datasets are not favourable.

On the CHB-MIT dataset (see Figure A5), the three models paint a different pictures. While the CHB model performs the best, being trained on the same dataset as it is tested on, the SWEC models is very competitive across the board, even beating the native model at the $64\times$ compression ratio. On the other hand, the BCI model still performs the worst, but the gap is smaller than on the SWEC dataset.

Next, we test on the BCI IV-2a dataset. Figure A6 indicates that, like previous results, cross-testing with different datasets and modalities generally lowers performance. However, the distance between the two models train on EEG datasets is less significant here, presumably due to the fact that the testing dataset is large enough to lessen the effects of overfitting. In fact, when training on a different modality, the lower compression scenario is most affected, where the specific information content of the signal is more relevant to a better compression. On the contrary, with a higher compression ratio fidelity suffers as a consequence, and hence the differences between EEG and iEEG also become less impactful.

Finally, we test the cross-dataset and cross-modalities generalization performance on the BONN dataset. Figure A7 shows that the CHB (same modality) and SWEC (different modalities) models are competitive. In particular, the model trained on CHB-MIT tends to perform better at lower compression ratios. The opposite is true for the model trained on SWEC, which scales more gracefully with the compression ratio.

Across all the different datasets we observe similar trends: the EEG models perform better at lower compression while the iEEG model performs better at higher compression. One possible interpretation is that EEG models are exposed to a sufficient amount of noise and are able to compress it at low compression ratios, while they tend to overfit on that same noise and are less effective in a higher compression scenario. On the contrary, iEEG models are less exposed to noise and thus do not compress it as efficiently, but also do not lose performance because of it with high compression ratios.

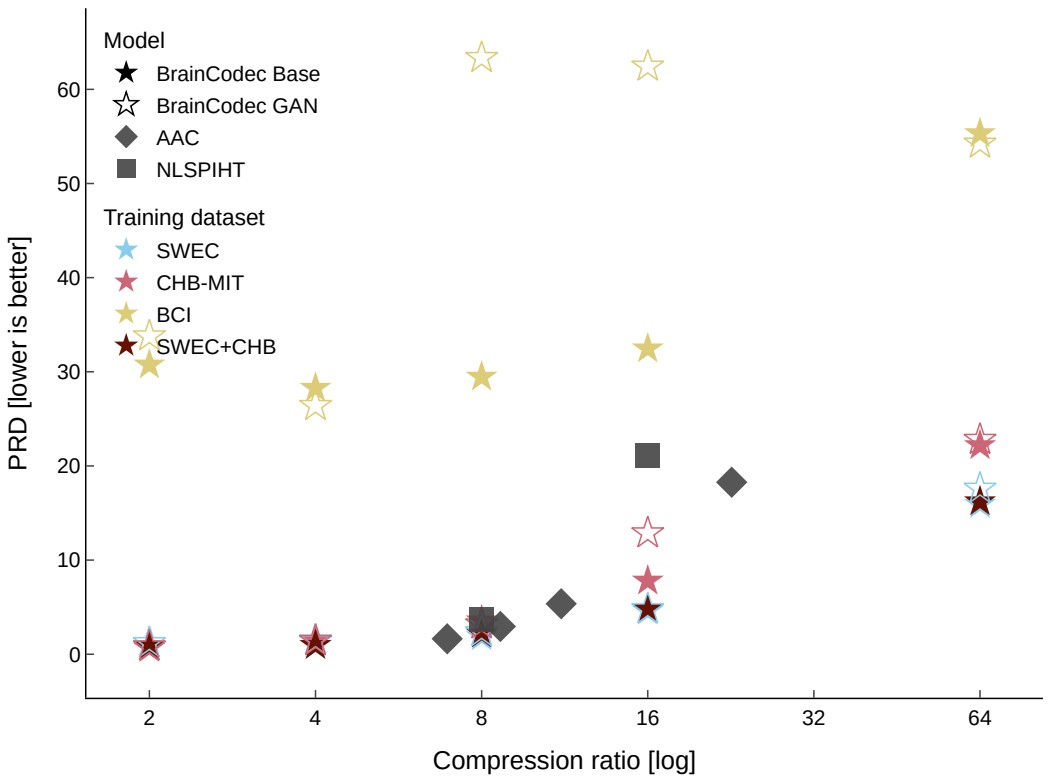

Figure A2: Reconstruction fidelity of BrainCodec on the iEEG SWEC dataset.

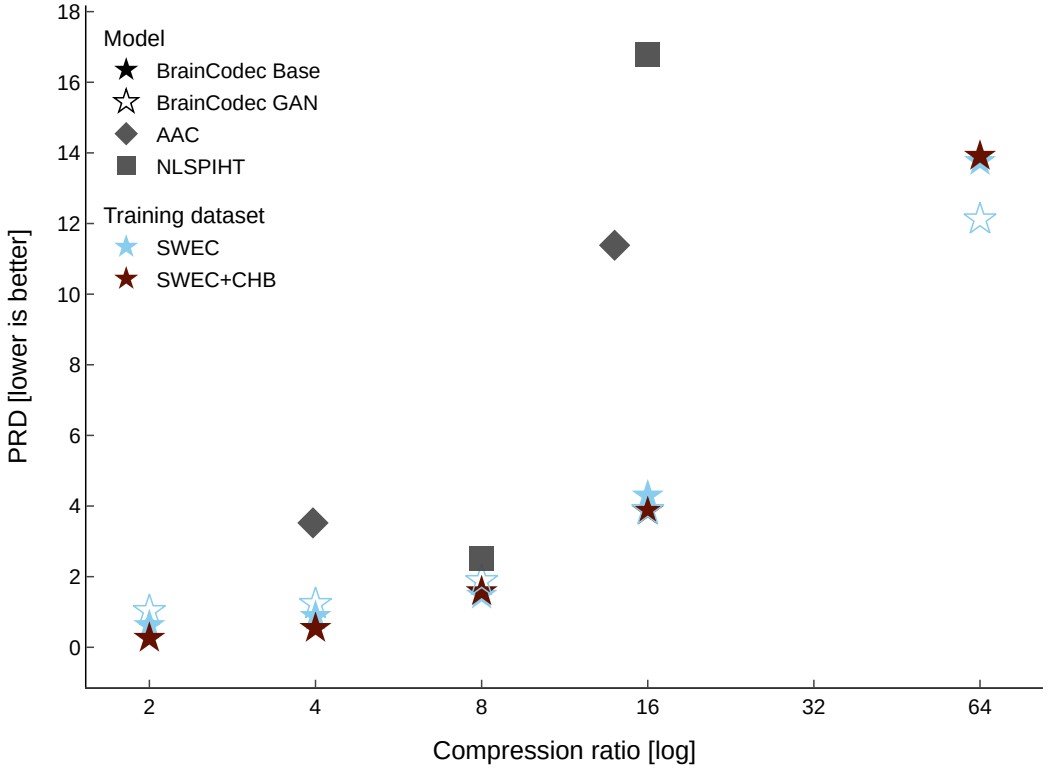

Figure A3: Reconstruction fidelity of BrainCodec on the iEEG MC dataset.

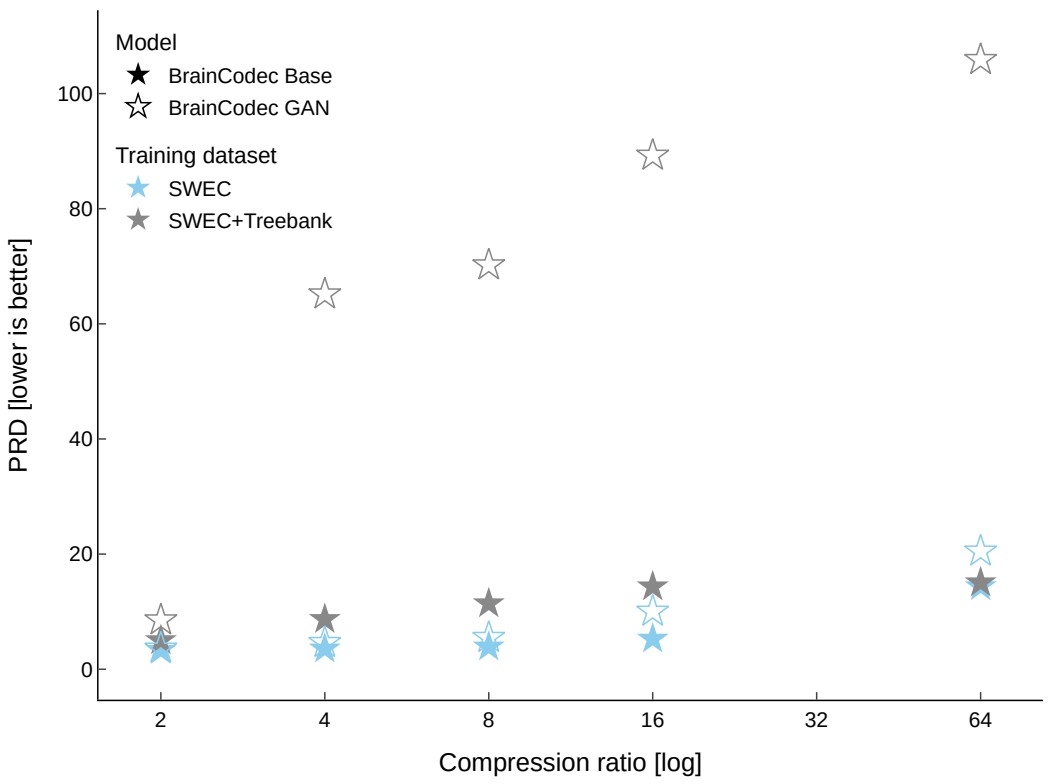

Figure A4: Reconstruction fidelity of BrainCodec on the iEEG Brain Treebank dataset.

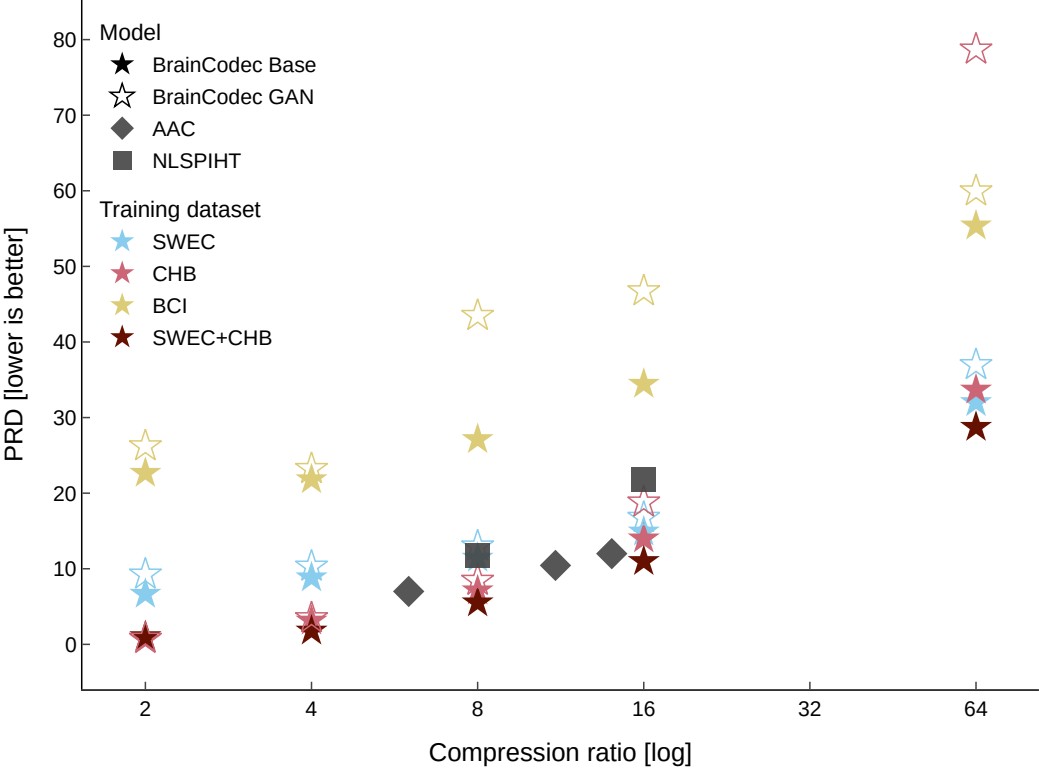

Figure A5: Reconstruction fidelity of BrainCodec on the EEG CHB dataset.

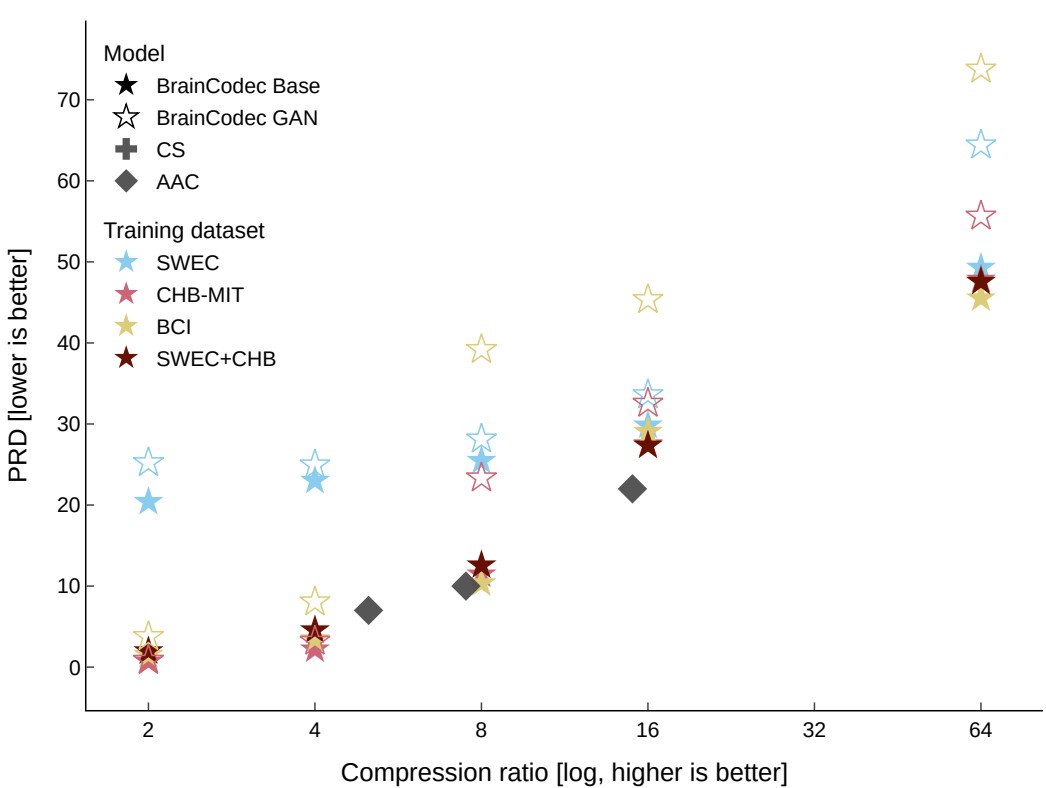

Figure A6: Reconstruction fidelity of BrainCodec on the EEG BCI IV-2a dataset.

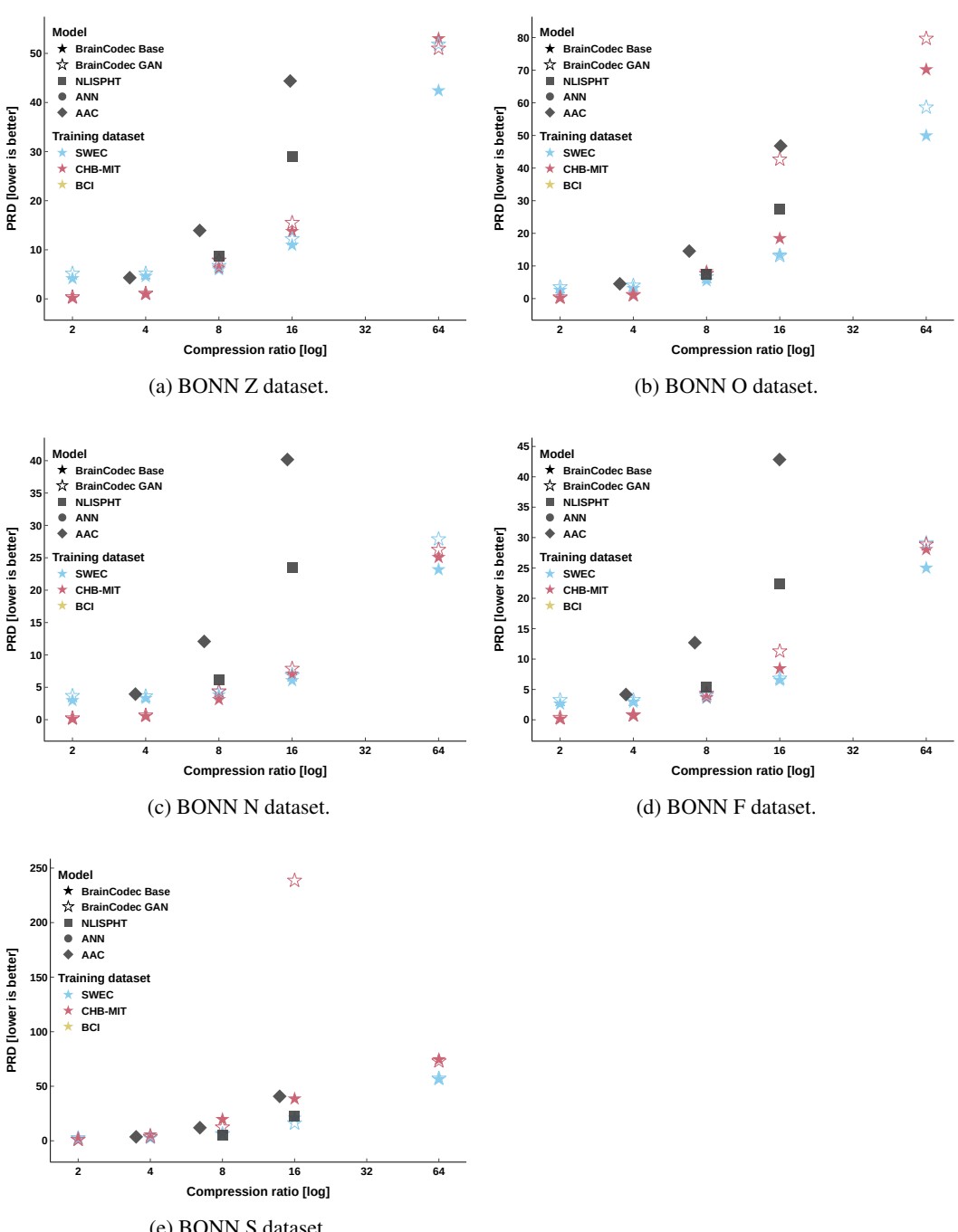

(a) BONN Z dataset.

(b) BONN O dataset.

(c) BONN N dataset.

(d) BONN F dataset.

(e) BONN S dataset.

Figure A7: Reconstruction fidelity of BrainCodec on the EEG BONN dataset.

## C.2 SEIZURE DETECTION AT EXTREME COMPRESSION RATIOS

We analyze the performance of BrainCodec in retaining the information content of the iEEG signal necessary to perform seizure detection.

Figure A8a shows the same scenario as in the main Results, i.e., when an EEGWaveNet model trained on the original signal is tested on the reconstructed signal. Both F1-score and accuracy are stable up to a $64\times$ ratio, and then start to decrease significantly. Interestingly, the GAN model loses performance more gracefully than the base model, indicating that it retains more of the information used by the automated seizure detection model. This is in contrast to human preference, as the expert evaluates the Base model as better than the GAN model for seizure detection (see App. E).

The opposite scenario, i.e., when an EEGWaveNet model trained on the reconstructed signal is tested on the original signal, shows similar results in Figure A8b. Here, the GAN model keeps the same performance up to a $128\times$ compression ratio, but then also decreases rapidly. The Base model still reaches a $64\times$ compression at the same F1-score level.

Finally, Figure A8c shows a significantly different behavior. When the models are both trained and tested on the reconstructed signal, only minor degradation in performance can be observed even at a $512\times$ compression ratio. This result suggests that the neural compressor is preserving some of the necessary information to perform seizure detection even at such high compression, but the EEGWaveNet model relies on other, not maintained, information when trained on the original signal. Only when trained with this new implicit set of features is the classification model able to fully make use of it.

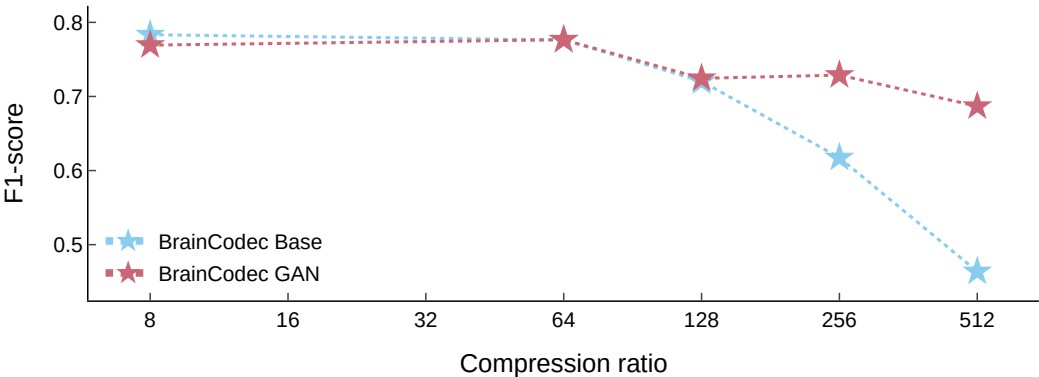

(a) EEGWaveNet is trained on the original iEEG signal, and then the compressed signal is used to perform classification.

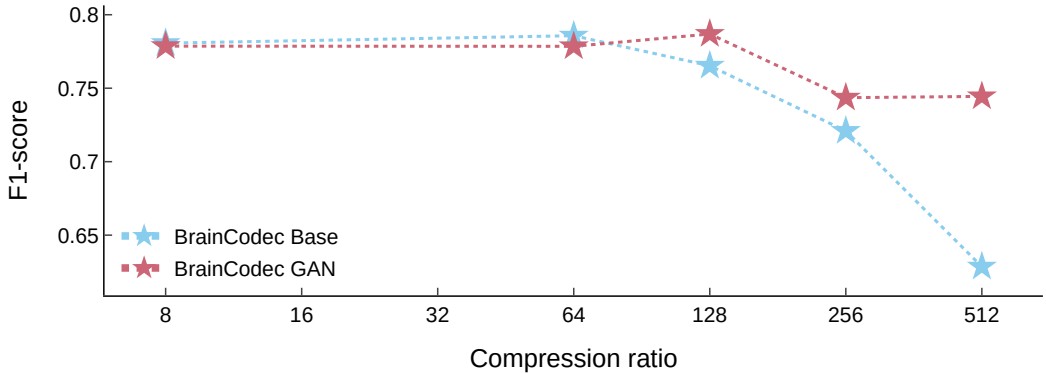

(b) Seizure classification performance of BrainCodec with EEGWaveNet. EEGWaveNet is trained on the original iEEG signal, and then the compressed signal is used to perform classification.

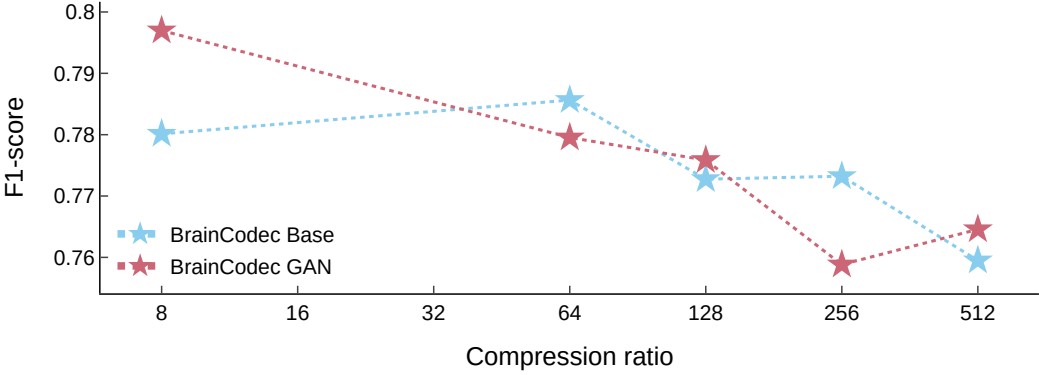

(c) Seizure classification performance of BrainCodec with EEGWaveNet. EEGWaveNet is trained on the compressed iEEG signal, and then the same compressed signal is used to perform classification.

Figure A8: Seizure classification performance of BrainCodec with EEGWaveNet.

### C.3 BASE VS GAN NEURAL COMPRESSOR

The architecture of BrainCodec allows us to introduce an adversarial training routine using a GAN, which has empirically shown better performance in compressing audio (Défossez et al., 2023). We observe that the effect of GAN on EEG is more mixed, improving results in some aspects and worsening in others. On the one hand, as seen in the main Results and in App. C.2, the GAN model better compresses the information content required by the EEGWaveNet model to perform seizure detection. Moreover, Figure A9 shows that the reconstruction of the higher frequencies is significantly better for the GAN model. In fact, with an increasing compression ratio, the Base model suppresses frequencies above $\sim 40$Hz. The GAN, on the other hand, recovers the entire range of frequencies even at high compression. Figure A9 further illustrates the trade-off between the time and frequency domain. Comparing Figure A9c and Figure A9d, it is clear that by increasing the weight $\lambda_t$, the output of the GAN converges to the output of the Base model.

On the other hand, the Base model obtains a higher PRD overall, indicating that the EEG signal contains predominantly lower frequency information. Given that the Base model does not hinder seizure detection even at $64\times$ compression, this indicates that EEGWaveNet mostly focuses on the lower frequency components to make its prediction even when the full signal is available. However, the GAN model with a fuller spectrum of frequency reconstruction still outperforms the Base model on the downstream task. This observation might extend to human experts as well, as our neurologist subjectively evaluated the signal compressed by the Base model as higher fidelity than the GAN model and more useful for seizure classification (see App. E).

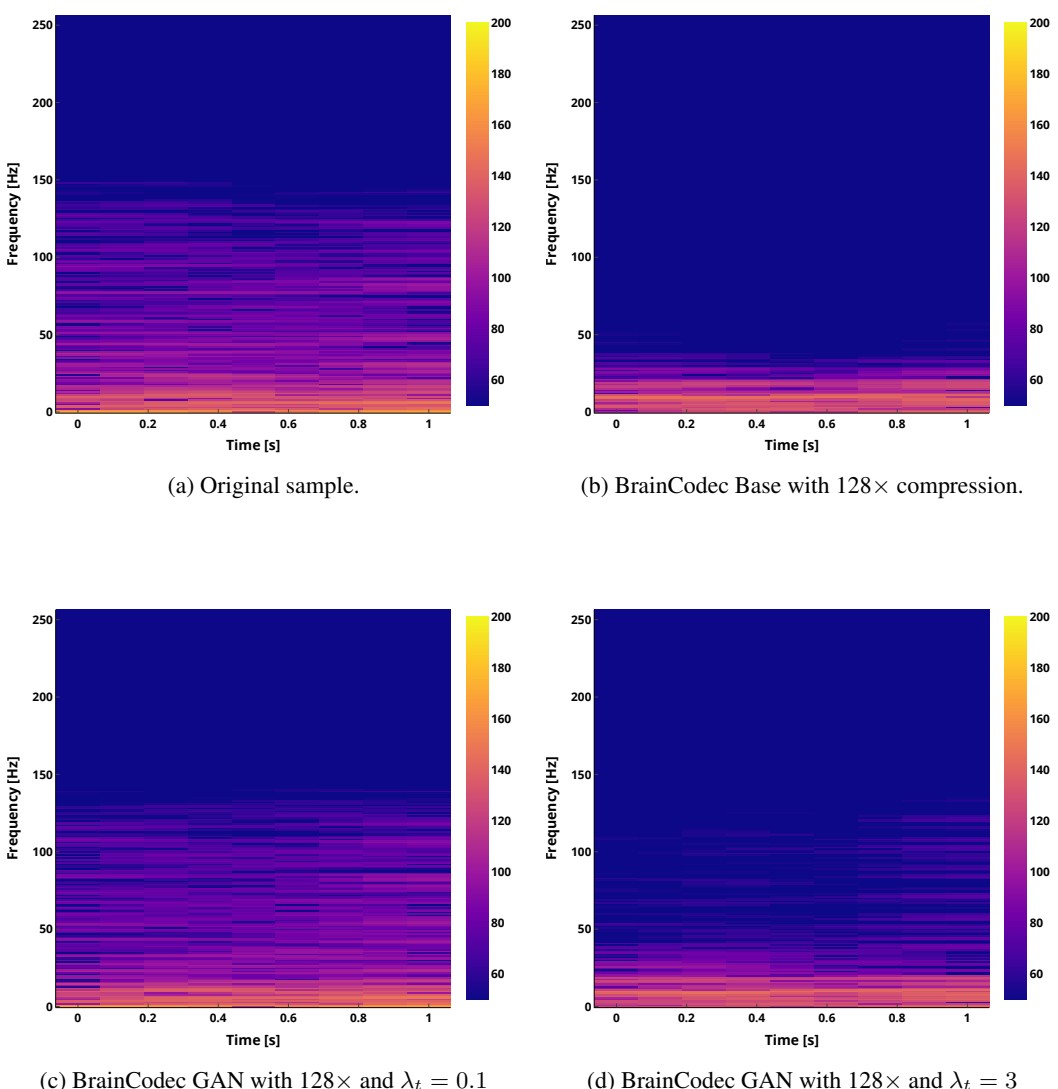

(a) Original sample.

(b) BrainCodec Base with $128\times$ compression.

(c) BrainCodec GAN with $128\times$ and $\lambda_t = 0.1$

(d) BrainCodec GAN with $128\times$ and $\lambda_t = 3$

Figure A9: Comparison among the spectrograms generated by different BrainCodec models.

# D FULL COMPRESSION RESULTS

| SWEC | CR ↑ | PRD ↓ | PRD-spec ↓ | RMSE ↓ | SNR ↑ | PSNR ↑ | Source |
|---|---|---|---|---|---|---|---|
| | 2 | 0.66 | 0.81 | 1.67 | 57.60 | 70.14 | |
| | 4 | 0.94 | 1.05 | 2.12 | 50.95 | 64.89 | |
| Base | 8 | 1.88 | 1.46 | 3.11 | 39.86 | 55.12 | Ours |
| | 16 | 4.60 | 3.39 | 6.33 | 31.08 | 46.00 | |
| | 64 | 15.74 | 10.93 | 16.14 | 17.96 | 36.73 | |
| | 2 | 0.65 | 0.88 | 1.78 | 61.76 | 73.64 | |
| | 4 | 1.35 | 1.20 | 3.04 | 51.11 | 63.26 | |
| Base [CHB] | 8 | 3.56 | 3.02 | 5.67 | 36.60 | 49.93 | Ours |
| | 16 | 7.79 | 7.57 | 10.90 | 28.63 | 42.01 | |
| | 64 | 22.17 | 22.54 | 25.08 | 16.23 | 31.19 | |
| | 2 | 30.74 | 51.56 | 36.53 | 26.35 | 29.58 | |
| | 4 | 28.27 | 47.78 | 33.39 | 22.74 | 29.90 | |
| Base [BCI] | 8 | 29.45 | 47.94 | 34.44 | 18.34 | 29.08 | Ours |
| | 16 | 32.45 | 49.87 | 36.49 | 14.99 | 27.61 | |
| | 64 | 55.31 | 76.93 | 55.54 | 7.24 | 22.23 | |
| | 2 | 0.89 | 0.86 | 1.83 | 54.63 | 63.58 | |
| | 4 | 0.95 | 0.85 | 1.91 | 47.78 | 62.53 | |
| Base [SWEC+CHB] | 8 | 2.16 | 1.69 | 3.64 | 39.29 | 53.88 | Ours |
| | 16 | 4.78 | 3.51 | 6.33 | 30.35 | 45.36 | |
| | 64 | 16.21 | 13.00 | 16.98 | 17.63 | 33.62 | |
| | 2 | 1.20 | 1.13 | 2.41 | 49.92 | 60.72 | |
| | 4 | 1.49 | 1.15 | 2.68 | 44.51 | 57.67 | |
| GAN | 8 | 2.37 | 1.57 | 3.49 | 37.20 | 52.23 | Ours |
| | 16 | 4.79 | 2.15 | 5.83 | 29.15 | 44.98 | |
| | 64 | 17.60 | 7.82 | 16.52 | 16.31 | 32.79 | |
| | 2 | 0.84 | 1.04 | 2.14 | 54.33 | 68.50 | |
| | 4 | 1.42 | 2.32 | 3.17 | 47.13 | 61.30 | |
| GAN [CHB] | 8 | 3.18 | 1.57 | 4.34 | 34.23 | 49.25 | Ours |
| | 16 | 12.86 | 30.72 | 17.73 | 27.71 | 41.40 | |
| | 64 | 22.79 | 71.43 | 26.09 | 15.88 | 31.41 | |
| | 2 | 33.78 | 56.68 | 38.63 | 20.16 | 28.02 | |
| | 4 | 26.39 | 46.92 | 30.14 | 18.80 | 29.64 | |
| GAN [BCI] | 8 | 63.35 | 61.97 | 70.76 | 12.33 | 22.34 | Ours |
| | 16 | 62.43 | 63.04 | 69.80 | 11.51 | 22.46 | |
| | 64 | 54.25 | 63.29 | 52.04 | 6.54 | 22.30 | |
| | 2 | 2.78 | 1.91 | 3.99 | 36.76 | 53.00 | |
| | 4 | 1.25 | 1.00 | 2.30 | 45.19 | 59.32 | |
| GAN [SWEC+CHB] | 8 | 15.80 | 3.79 | 14.58 | 17.80 | 36.31 | Ours |
| | 16 | 7.67 | 2.91 | 7.87 | 24.51 | 43.11 | |
| | 64 | 16.97 | 7.08 | 16.09 | 16.56 | 33.12 | |
| | 6.93 | 1.64 | / | / | / | / | |
| AAC | 8.65 | 2.95 | / | / | / | / | Reproduced (Nguyen et al., 2017) |
| | 11.15 | 5.36 | / | / | / | / | |
| | 22.71 | 18.27 | / | / | / | / | |
| NLSPIHT | 8 | 3.69 | / | / | / | / | Reproduced (Xu et al., 2015) |
| | 16 | 21.09 | / | / | / | / | |

Table A7: Full compression results on the SWEC iEEG dataset.

| MC | CR ↑ | PRD ↓ | PRD-spec ↓ | RMSE ↓ | SNR ↑ | PSNR ↑ | Source |
|---|---|---|---|---|---|---|---|
| | 2 | 0.62 | 0.11 | 1.07 | 47.49 | 61.83 | |
| | 4 | 0.88 | 0.37 | 1.77 | 44.60 | 58.28 | |
| Base [**SWEC**] | 8 | 1.48 | 0.64 | 3.07 | 39.43 | 53.45 | Ours |
| | 16 | 4.29 | 5.01 | 11.80 | 32.59 | 44.80 | |
| | 64 | 13.76 | 19.31 | 37.43 | 21.03 | 34.12 | |
| | 2 | 0.26 | 0.22 | 0.64 | 56.44 | 67.82 | |
| | 4 | 0.54 | 0.36 | 1.39 | 48.54 | 61.87 | |
| Base [**SWEC+CHB**] | 8 | 1.58 | 1.14 | 3.95 | 39.36 | 52.70 | Ours |
| | 16 | 3.86 | 3.81 | 9.93 | 32.09 | 45.14 | |
| | 64 | 13.91 | 19.38 | 37.25 | 20.31 | 33.83 | |
| | 2 | 1.03 | 0.54 | 2.14 | 43.87 | 56.84 | |
| | 4 | 1.23 | 0.63 | 2.69 | 41.85 | 55.20 | |
| GAN [**SWEC**] | 8 | 1.88 | 1.23 | 4.09 | 37.75 | 51.43 | Ours |
| | 16 | 3.89 | 3.08 | 9.68 | 31.81 | 45.02 | |
| | 64 | 12.12 | 10.38 | 29.19 | 20.70 | 34.73 | |
| | 2 | 1.85 | 1.91 | 4.75 | 38.86 | 52.65 | |
| | 4 | 0.74 | 0.32 | 1.85 | 45.61 | 58.92 | |
| GAN [**SWEC+CHB**] | 8 | 9.21 | 4.64 | 19.98 | 22.97 | 38.37 | Ours |
| | 16 | 5.14 | 3.40 | 11.07 | 28.50 | 43.84 | |
| | 64 | 11.89 | 8.49 | 28.06 | 20.70 | 34.89 | |
| AAC | 3.96 | 3.52 | | | | | Reproduced (Nguyen et al., 2017) |
| | 13.93 | 11.38 | | | | | |
| NLSPIHT | 8 | 2.52 | | | | | Reproduced (Xu et al., 2015) |
| | 16 | 16.78 | | | | | |

Table A8: Full compression results on the MC iEEG dataset.

| Treebank | CR ↑ | PRD ↓ | PRD-spec ↓ | RMSE ↓ | SNR ↑ | PSNR ↑ | Source |
|---|---|---|---|---|---|---|---|
| | 2 | 3.32 | 1.51 | 8418.56 | 34.58 | 44.10 | |
| | 4 | 3.54 | 1.63 | 8417.44 | 33.70 | 43.23 | |
| Base [**SWEC**] | 8 | 3.89 | 1.80 | 8418.83 | 32.56 | 42.10 | Ours |
| | 16 | 5.26 | 3.16 | 8422.62 | 28.92 | 38.45 | |
| | 64 | 14.36 | 9.19 | 8438.63 | 18.89 | 28.38 | |
| | 2 | 2.78 | 1.91 | 3.99 | 36.76 | 53.00 | |
| | 4 | 1.25 | 1.00 | 2.30 | 45.19 | 59.32 | |
| Base [**SWEC+Treebank**] | 8 | 15.80 | 3.79 | 14.58 | 17.80 | 36.31 | Ours |
| | 16 | 7.67 | 2.91 | 7.87 | 24.51 | 43.11 | |
| | 64 | 16.97 | 7.08 | 16.09 | 16.56 | 33.12 | |
| | 2 | 1.20 | 1.13 | 2.41 | 49.92 | 60.72 | |
| | 4 | 1.49 | 1.15 | 2.68 | 44.51 | 57.67 | |
| GAN [**SWEC**] | 8 | 2.37 | 1.57 | 3.49 | 37.20 | 52.23 | Ours |
| | 16 | 4.79 | 2.15 | 5.83 | 29.15 | 44.98 | |
| | 64 | 17.60 | 7.82 | 16.52 | 16.31 | 32.79 | |
| | 2 | 8.52 | 3.74 | 6312.93 | 23.08 | 32.41 | |
| | 4 | 65.14 | 65.24 | 7145.75 | 4.62 | 13.81 | |
| GAN [**SWEC+Treebank**] | 8 | 70.19 | 66.90 | 7166.63 | 3.53 | 12.73 | Ours |
| | 16 | 89.24 | 76.40 | 7634.31 | 1.01 | 10.19 | |
| | 64 | 105.87 | 105.57 | 10410.84 | -0.41 | 8.76 | |

Table A9: Full compression results on the Brain Treebank iEEG dataset.

| CHB | CR ↑ | PRD ↓ | PRD-spec ↓ | RMSE ↓ | SNR ↑ | PSNR ↑ | Source |
|---|---|---|---|---|---|---|---|
| | 2 | 0.61 | 0.44 | 0.90 | 55.30 | 70.21 | |
| | 4 | 3.10 | 2.73 | 4.90 | 42.22 | 57.80 | |
| Base | 8 | 7.10 | 4.11 | 6.82 | 28.32 | 44.88 | Ours |
| | 16 | 13.97 | 7.99 | 12.07 | 21.01 | 37.74 | |
| | 64 | 33.66 | 20.21 | 24.69 | 10.96 | 28.20 | |
| | 2 | 6.65 | 2.67 | 5.25 | 30.66 | 46.53 | |
| | 4 | 8.82 | 3.92 | 7.27 | 27.36 | 43.96 | |
| Base [**SWEC**] | 8 | 11.42 | 5.14 | 9.12 | 23.70 | 40.75 | Ours |
| | 16 | 14.88 | 6.74 | 11.64 | 19.99 | 37.21 | |
| | 64 | 32.00 | 16.41 | 22.47 | 11.17 | 28.69 | |
| | 2 | 22.65 | 38.81 | 20.08 | 34.25 | 35.14 | |
| | 4 | 21.80 | 35.30 | 18.98 | 26.35 | 34.12 | |
| Base [**BCI**] | 8 | 27.11 | 38.01 | 22.51 | 18.22 | 30.94 | Ours |
| | 16 | 34.43 | 44.01 | 26.82 | 13.86 | 28.02 | |
| | 64 | 55.37 | 66.13 | 39.41 | 6.66 | 22.99 | |
| | 2 | 1.04 | 0.34 | 0.72 | 46.81 | 58.77 | |
| | 4 | 1.78 | 0.55 | 1.28 | 37.49 | 54.62 | |
| Base [**SWEC+CHB**] | 8 | 5.51 | 1.91 | 4.15 | 28.13 | 45.36 | Ours |
| | 16 | 10.98 | 4.09 | 7.94 | 21.54 | 38.84 | |
| | 64 | 28.73 | 14.79 | 19.60 | 11.71 | 29.35 | |
| | 2 | 0.77 | 0.52 | 1.02 | 49.61 | 65.74 | |
| | 4 | 3.48 | 35.17 | 3.89 | 39.77 | 55.94 | |
| GAN | 8 | 8.43 | 3.24 | 6.86 | 25.63 | 42.70 | Ours |
| | 16 | 18.78 | 74.91 | 16.00 | 19.01 | 35.95 | |
| | 64 | 78.70 | 10214.30 | 64.54 | 9.76 | 27.07 | |
| | 2 | 9.25 | 4.02 | 7.40 | 27.03 | 42.91 | |
| | 4 | 10.30 | 4.40 | 8.32 | 25.11 | 41.77 | |
| GAN [**SWEC**] | 8 | 12.97 | 5.12 | 10.01 | 22.23 | 39.35 | Ours |
| | 16 | 16.76 | 6.56 | 12.36 | 18.67 | 36.11 | |
| | 64 | 36.96 | 13.44 | 24.91 | 9.42 | 27.39 | |
| | 2 | 26.22 | 43.68 | 22.09 | 24.60 | 32.26 | |
| | 4 | 23.22 | 36.67 | 19.39 | 19.99 | 32.13 | |
| GAN [**BCI**] | 8 | 43.47 | 47.88 | 34.92 | 12.73 | 26.24 | Ours |
| | 16 | 46.73 | 50.64 | 36.91 | 11.20 | 25.47 | |
| | 64 | 59.94 | 55.57 | 39.82 | 4.74 | 22.11 | |
| | 2 | 6.79 | 1.85 | 4.85 | 26.42 | 43.65 | |
| | 4 | 2.28 | 0.59 | 1.59 | 34.89 | 52.03 | |
| GAN [**SWEC+CHB**] | 8 | 31.50 | 9.82 | 21.50 | 11.02 | 29.07 | Ours |
| | 16 | 21.01 | 4.90 | 14.33 | 15.61 | 33.55 | |
| | 64 | 36.80 | 9.99 | 25.47 | 9.63 | 27.58 | |
| | 6 | 7 | | | | | |
| AAC | 11.07 | 10.44 | | | | | Reported (Nguyen et al., 2017) |
| | 14 | 12 | | | | | |
| NLSPIHT | 8 | 11.76 | | | | | Reported (Xu et al., 2015) |
| | 16 | 21.83 | | | | | |

Table A10: Full compression results on the CHB EEG dataset.

| BCI | CR ↑ | PRD ↓ | PRD-spec ↓ | RMSE ↓ | SNR ↑ | PSNR ↑ | Source |
|---|---|---|---|---|---|---|---|
| | 2 | 1.50 | 0.88 | 0.23 | 40.73 | 51.20 | |
| | 4 | 3.44 | 1.79 | 0.52 | 31.83 | 44.29 | |
| Base | 8 | 10.36 | 5.98 | 1.78 | 19.81 | 34.86 | Ours |
| | 16 | 29.05 | 11.73 | 4.41 | 12.38 | 25.66 | |
| | 64 | 45.51 | 25.37 | 7.34 | 6.65 | 22.31 | |
| | 2 | 20.40 | 5.74 | 3.04 | 15.65 | 28.93 | |
| | 4 | 23.00 | 6.14 | 3.44 | 14.10 | 27.83 | |
| Base [**SWEC**] | 8 | 25.46 | 7.18 | 4.05 | 13.01 | 26.83 | Ours |
| | 16 | 29.82 | 9.99 | 6.36 | 11.05 | 25.21 | |
| | 64 | 49.29 | 24.52 | 10.53 | 5.42 | 21.48 | |
| | 2 | 0.71 | 0.71 | 1.00 | 46.29 | 60.51 | |
| | 4 | 2.19 | 0.91 | 1.13 | 33.07 | 48.19 | |
| Base [**CHB**] | 8 | 11.44 | 5.08 | 3.11 | 17.63 | 33.41 | Ours |
| | 16 | 27.50 | 11.36 | 7.85 | 11.26 | 25.95 | |
| | 64 | 47.81 | 23.80 | 11.56 | 5.93 | 21.81 | |
| | 2 | 1.97 | 0.92 | 0.53 | 35.12 | 48.41 | |
| | 4 | 4.55 | 1.74 | 0.94 | 25.68 | 41.10 | |
| Base [**SWEC+CHB**] | 8 | 12.56 | 5.56 | 2.45 | 16.60 | 32.41 | Ours |
| | 16 | 27.35 | 10.08 | 5.09 | 10.81 | 25.84 | |
| | 64 | 47.55 | 25.33 | 9.88 | 5.44 | 21.79 | |
| | 2 | 3.77 | 2.12 | 0.55 | 29.23 | 43.14 | |
| | 4 | 8.04 | 6.24 | 1.17 | 22.13 | 36.70 | |
| GAN | 8 | 39.18 | 10.84 | 21.53 | 9.72 | 23.46 | Ours |
| | 16 | 45.35 | 11.62 | 22.24 | 8.27 | 21.95 | |
| | 64 | 73.78 | 34.63 | 11.22 | 2.05 | 17.88 | |
| | 2 | 25.22 | 7.90 | 3.79 | 13.60 | 26.99 | |
| | 4 | 24.94 | 6.87 | 3.73 | 13.20 | 27.08 | |
| GAN [**SWEC**] | 8 | 28.14 | 7.30 | 4.38 | 12.10 | 25.93 | Ours |
| | 16 | 33.60 | 10.16 | 6.29 | 9.63 | 24.11 | |
| | 64 | 64.42 | 25.23 | 11.18 | 2.39 | 19.06 | |
| | 2 | 0.84 | 0.49 | 0.42 | 41.08 | 56.14 | |
| | 4 | 3.19 | 12.46 | 0.93 | 29.63 | 44.97 | |
| GAN [**CHB**] | 8 | 23.34 | 6.27 | 3.74 | 13.35 | 27.53 | Ours |
| | 16 | 32.55 | 9.53 | 7.17 | 10.21 | 24.52 | |
| | 64 | 55.64 | 28.73 | 17.46 | 4.36 | 20.46 | |
| | 2 | 19.31 | 6.99 | 3.52 | 14.55 | 30.14 | |
| | 4 | 5.53 | 1.84 | 1.04 | 23.54 | 39.31 | |
| GAN [**SWEC+CHB**] | 8 | 46.38 | 14.64 | 8.04 | 6.69 | 22.06 | Ours |
| | 16 | 44.57 | 13.24 | 7.35 | 7.01 | 22.81 | |
| | 64 | 58.81 | 23.49 | 10.16 | 3.32 | 19.88 | |
| | 5 | 7 | | | | | |
| AAC | 7.5 | 10 | | | | | Reported (Nguyen et al., 2017) |
| | 15 | 22 | | | | | |
| CS | 5 | 20.19 | | | | | Reported (Du et al., 2024) |
| | 10 | 30.23 | | | | | |

Table A11: Full compression results on the BCI EEG dataset.

| BONN Z | CR ↑ | PRD ↓ | PRD-spec ↓ | RMSE ↓ | SNR ↓ | PSNR ↓ | Source |
|---|---|---|---|---|---|---|---|
| | 2 | 4.15 | 0.51 | 1.91 | 28.63 | 39.67 | |
| | 4 | 4.60 | 0.69 | 2.11 | 27.55 | 38.75 | |
| Base [**SWEC**] | 8 | 5.97 | 1.75 | 2.74 | 25.20 | 36.26 | Ours |
| | 16 | 10.97 | 5.14 | 5.05 | 20.07 | 30.85 | |
| | 64 | 42.42 | 20.82 | 19.41 | 8.21 | 19.00 | |
| | 2 | 0.18 | 0.12 | 0.08 | 57.12 | 66.63 | |
| | 4 | 0.98 | 0.38 | 0.45 | 42.52 | 51.95 | |
| Base [**CHB**] | 8 | 6.28 | 2.68 | 2.90 | 25.56 | 35.72 | Ours |
| | 16 | 13.75 | 6.15 | 6.33 | 18.17 | 28.87 | |
| | 64 | 53.02 | 24.89 | 24.61 | 6.49 | 17.07 | |
| | 2 | 2.64 | 2.75 | 1.36 | 43.37 | 44.69 | |
| | 4 | 6.20 | 3.99 | 2.98 | 30.30 | 35.90 | |
| Base [**BCI**] | 8 | 21.71 | 17.09 | 10.21 | 17.08 | 24.99 | Ours |
| | 16 | 32.19 | 23.89 | 14.98 | 11.80 | 21.44 | |
| | 64 | 58.33 | 46.19 | 27.08 | 5.25 | 16.07 | |
| | 2 | 5.16 | 1.17 | 2.37 | 27.58 | 37.75 | |
| | 4 | 5.19 | 0.94 | 2.38 | 26.54 | 37.61 | |
| GAN [**SWEC**] | 8 | 6.73 | 1.38 | 3.07 | 23.86 | 35.12 | Ours |
| | 16 | 12.21 | 3.62 | 5.61 | 18.84 | 29.89 | |
| | 64 | 51.84 | 19.67 | 23.67 | 5.90 | 17.23 | |
| | 2 | 0.37 | 0.26 | 0.17 | 50.14 | 60.28 | |
| | 4 | 1.09 | 0.57 | 0.50 | 40.84 | 50.95 | |
| GAN [**CHB**] | 8 | 7.84 | 1.57 | 3.59 | 22.84 | 33.85 | Ours |
| | 16 | 15.51 | 4.34 | 7.10 | 16.88 | 27.87 | |
| | 64 | 51.00 | 23.91 | 23.36 | 6.09 | 17.38 | |
| | 2 | 6.79 | 6.00 | 3.33 | 29.15 | 35.09 | |
| | 4 | 12.79 | 11.02 | 6.09 | 21.25 | 29.38 | |
| GAN [**BCI**] | 8 | 27.93 | 21.00 | 13.08 | 12.88 | 22.70 | Ours |
| | 16 | 35.83 | 22.84 | 16.69 | 10.26 | 20.51 | |
| | 64 | 69.00 | 37.75 | 31.51 | 3.39 | 14.67 | |
| | 3.44 | 4.31 | | | | | |
| AAC | 6.67 | 13.93 | | | | | Reproduced (Nguyen et al., 2017) |
| | 15.71 | 44.38 | | | | | |
| NLSPIHT | 8 | 8.64 | | | | | Reproduced (Xu et al., 2015) |
| | 16 | 29 | | | | | |
| ANN [**BCI**] | 3.96 | 6.64 | | | | | Reported (Hejrati et al., 2017) |

Table A12: Full compression results on the Z subset of the BONN EEG dataset.

| **BONN O** | CR ↑ | PRD ↓ | PRD-spec ↓ | RMSE ↓ | SNR ↓ | PSNR ↓ | Source |
|---|---|---|---|---|---|---|---|
| | 2 | 2.57 | 0.49 | 1.70 | 33.20 | 42.19 | |
| | 4 | 3.13 | 0.78 | 2.07 | 31.45 | 40.57 | |
| Base [**SWEC**] | 8 | 5.48 | 2.28 | 3.65 | 26.48 | 35.48 | Ours |
| | 16 | 13.44 | 8.84 | 9.23 | 18.92 | 27.37 | |
| | 64 | 49.93 | 37.88 | 33.93 | 6.93 | 15.88 | |
| | 2 | 0.18 | 0.12 | 0.12 | 57.50 | 64.88 | |
| | 4 | 1.06 | 0.45 | 0.74 | 42.48 | 49.41 | |
| Base [**CHB**] | 8 | 8.32 | 5.33 | 5.85 | 24.20 | 31.67 | Ours |
| | 16 | 18.43 | 14.43 | 13.07 | 16.68 | 24.71 | |
| | 64 | 70.19 | 53.87 | 48.72 | 3.93 | 12.92 | |
| | 2 | 8.52 | 10.22 | 6.77 | 39.43 | 33.01 | |
| | 4 | 12.02 | 12.10 | 9.01 | 26.12 | 28.61 | |
| Base [**BCI**] | 8 | 34.94 | 39.01 | 24.89 | 12.87 | 19.04 | Ours |
| | 16 | 46.80 | 48.75 | 32.80 | 8.52 | 16.42 | |
| | 64 | 76.50 | 117.39 | 53.85 | 3.13 | 12.09 | |
| | 2 | 3.52 | 1.07 | 2.33 | 31.26 | 39.49 | |
| | 4 | 4.02 | 1.11 | 2.67 | 29.31 | 38.26 | |
| GAN [**SWEC**] | 8 | 6.61 | 1.75 | 4.36 | 24.63 | 34.07 | Ours |
| | 16 | 13.13 | 5.07 | 8.85 | 18.45 | 27.56 | |
| | 64 | 58.62 | 37.89 | 39.84 | 5.02 | 14.48 | |
| | 2 | 0.40 | 0.26 | 0.28 | 49.93 | 57.72 | |
| | 4 | 1.14 | 0.56 | 0.77 | 41.24 | 48.77 | |
| GAN [**CHB**] | 8 | 7.61 | 2.22 | 5.15 | 23.61 | 32.42 | Ours |
| | 16 | 42.66 | 315.81 | 36.42 | 16.47 | 23.75 | |
| | 64 | 79.69 | 450.99 | 57.10 | 3.76 | 12.81 | |
| | 2 | 13.42 | 17.12 | 10.27 | 25.60 | 27.80 | |
| | 4 | 19.14 | 24.02 | 13.92 | 18.49 | 24.27 | |
| GAN [**BCI**] | 8 | 40.33 | 46.75 | 28.54 | 9.95 | 17.71 | Ours |
| | 16 | 53.15 | 56.85 | 37.60 | 7.33 | 15.33 | |
| | 64 | 81.31 | 67.09 | 55.69 | 1.99 | 11.55 | |
| | 3.52 | 4.53 | | | | | |
| AAC | 6.79 | 14.56 | | | | | Reproduced (Nguyen et al., 2017) |
| | 16.13 | 46.79 | | | | | |
| NLSPIHT | 8 | 7.58 | | | | | Reproduced (Xu et al., 2015) |
| | 16 | 27.3 | | | | | |
| ANN [**BCI**] | 2.84 | 2 | | | | | Reported (Hejrati et al., 2017) |

Table A13: Full compression results on the O subset of the BONN EEG dataset.

| BONN N | CR ↑ | PRD ↓ | PRD-spec ↓ | RMSE ↓ | SNR ↓ | PSNR ↓ | Source |
|---|---|---|---|---|---|---|---|
| | 2 | 2.97 | 0.24 | 1.70 | 31.34 | 43.06 | |
| | 4 | 3.30 | 0.34 | 1.89 | 30.25 | 42.14 | |
| Base [**SWEC**] | 8 | 3.83 | 0.81 | 2.20 | 28.84 | 40.77 | Ours |
| | 16 | 6.05 | 2.33 | 3.55 | 25.01 | 36.74 | |
| | 64 | 23.18 | 12.17 | 13.58 | 13.24 | 24.83 | |
| | 2 | 0.11 | 0.09 | 0.06 | 59.97 | 70.89 | |
| | 4 | 0.53 | 0.19 | 0.32 | 47.21 | 57.96 | |
| Base [**CHB**] | 8 | 3.12 | 1.07 | 1.88 | 31.40 | 42.70 | Ours |
| | 16 | 7.07 | 2.58 | 4.22 | 23.84 | 35.45 | |
| | 64 | 25.07 | 13.91 | 14.89 | 12.77 | 24.24 | |
| | 2 | 6.74 | 9.41 | 4.85 | 46.02 | 40.34 | |
| | 4 | 7.80 | 9.52 | 5.36 | 35.87 | 36.49 | |
| Base [**BCI**] | 8 | 13.20 | 12.78 | 8.68 | 24.30 | 30.74 | Ours |
| | 16 | 20.84 | 19.55 | 13.05 | 17.77 | 26.21 | |
| | 64 | 48.97 | 52.07 | 29.16 | 7.47 | 18.13 | |
| | 2 | 3.68 | 0.63 | 2.10 | 30.33 | 41.20 | |
| | 4 | 3.63 | 0.43 | 2.07 | 29.47 | 41.28 | |
| GAN [**SWEC**] | 8 | 4.31 | 0.72 | 2.44 | 27.64 | 39.65 | Ours |
| | 16 | 6.85 | 1.32 | 3.95 | 23.66 | 35.55 | |
| | 64 | 27.88 | 8.74 | 16.00 | 11.16 | 23.16 | |
| | 2 | 0.23 | 0.17 | 0.13 | 53.61 | 64.86 | |
| | 4 | 0.67 | 0.40 | 0.38 | 44.30 | 55.68 | |
| GAN [**CHB**] | 8 | 4.40 | 0.64 | 2.56 | 27.68 | 39.53 | Ours |
| | 16 | 7.88 | 1.62 | 4.60 | 22.45 | 34.37 | |
| | 64 | 26.25 | 9.50 | 15.19 | 11.76 | 23.71 | |
| | 2 | 10.38 | 13.99 | 7.13 | 33.77 | 33.97 | |
| | 4 | 11.51 | 13.76 | 7.50 | 26.42 | 31.67 | |
| GAN [**BCI**] | 8 | 22.17 | 23.59 | 14.18 | 17.60 | 25.73 | Ours |
| | 16 | 26.17 | 24.44 | 16.55 | 15.31 | 24.20 | |
| | 64 | 55.56 | 42.30 | 31.92 | 5.40 | 16.93 | |
| | 3.63 | 3.95 | | | | | |
| AAC | 6.96 | 12.08 | | | | | Reproduced (Nguyen et al., 2017) |
| | 15.3 | 40.17 | | | | | |
| NLSPIHT | 8 | 6.13 | | | | | Reproduced (Xu et al., 2015) |
| | 16 | 23.55 | | | | | |
| ANN [**BCI**] | 5.23 | 6.82 | | | | | Reported (Hejrati et al., 2017) |

Table A14: Full compression results on the N subset of the BONN EEG dataset.

| BONN F | CR ↑ | PRD ↓ | PRD-spec ↓ | RMSE ↓ | SNR ↓ | PSNR ↓ | Source |
|---|---|---|---|---|---|---|---|
| | 2 | 2.63 | 0.25 | 1.61 | 33.16 | 42.27 | |
| | 4 | 2.92 | 0.37 | 1.80 | 32.10 | 41.31 | |
| Base [**SWEC**] | 8 | 3.57 | 0.91 | 2.40 | 30.32 | 39.24 | Ours |
| | 16 | 6.53 | 2.83 | 5.36 | 25.78 | 33.88 | |
| | 64 | 25.00 | 14.66 | 18.73 | 13.43 | 21.98 | |
| | 2 | 0.14 | 0.11 | 0.19 | 60.50 | 68.00 | |
| | 4 | 0.75 | 0.38 | 1.16 | 48.11 | 54.46 | |
| Base [**CHB**] | 8 | 3.69 | 1.58 | 3.79 | 32.33 | 39.41 | Ours |
| | 16 | 8.43 | 4.14 | 7.99 | 24.51 | 32.01 | |
| | 64 | 28.05 | 17.80 | 23.06 | 12.71 | 21.07 | |
| | 2 | 12.82 | 15.81 | 17.68 | 44.92 | 35.97 | |
| | 4 | 12.85 | 14.89 | 16.79 | 35.08 | 32.39 | |
| Base [**BCI**] | 8 | 19.13 | 19.26 | 20.88 | 23.88 | 26.30 | Ours |
| | 16 | 26.91 | 27.10 | 25.90 | 17.42 | 22.23 | |
| | 64 | 53.77 | 54.26 | 43.01 | 6.90 | 15.27 | |
| | 2 | 3.27 | 0.64 | 2.04 | 32.10 | 40.28 | |
| | 4 | 3.28 | 0.47 | 2.07 | 31.14 | 40.11 | |
| GAN [**SWEC**] | 8 | 4.04 | 0.76 | 2.67 | 28.95 | 38.09 | Ours |
| | 16 | 6.81 | 1.54 | 4.81 | 24.41 | 33.34 | |
| | 64 | 29.10 | 10.84 | 20.41 | 11.30 | 20.61 | |
| | 2 | 0.34 | 0.31 | 0.63 | 54.15 | 61.99 | |
| | 4 | 0.78 | 0.52 | 0.88 | 45.14 | 52.78 | |
| GAN [**CHB**] | 8 | 4.30 | 0.88 | 3.35 | 28.97 | 37.55 | Ours |
| | 16 | 11.29 | 8.53 | 16.70 | 23.15 | 31.33 | |
| | 64 | 28.87 | 14.65 | 23.85 | 11.74 | 20.76 | |
| | 2 | 16.01 | 19.79 | 19.89 | 33.02 | 30.22 | |
| | 4 | 15.62 | 18.81 | 17.52 | 25.99 | 27.97 | |
| GAN [**BCI**] | 8 | 27.74 | 28.25 | 27.50 | 17.24 | 22.35 | Ours |
| | 16 | 32.62 | 30.71 | 31.15 | 14.86 | 20.58 | |
| | 64 | 60.25 | 48.45 | 44.95 | 4.98 | 14.12 | |
| | 3.73 | 4.17 | | | | | |
| AAC | 7.16 | 12.7 | | | | | Reproduced (Nguyen et al., 2017) |
| | 15.96 | 42.84 | | | | | |
| NLSPIHT | 8 | 5.41 | | | | | Reproduced (Xu et al., 2015) |
| | 16 | 22.42 | | | | | |
| ANN [**BCI**] | 6.57 | 6.96 | | | | | Reported (Hejrati et al., 2017) |

Table A15: Full compression results on the F subset of the BONN EEG dataset.

| **BONN S** | CR ↑ | PRD ↓ | PRD-spec ↓ | RMSE ↓ | SNR ↓ | PSNR ↓ | Source |
|---|---|---|---|---|---|---|---|
| Base [**SWEC**] | 2 | 1.71 | 0.40 | 5.54 | 41.16 | 45.89 | |
| | 4 | 2.40 | 1.07 | 8.56 | 38.94 | 43.42 | |
| | 8 | 6.72 | 4.63 | 26.51 | 30.91 | 35.10 | Ours |
| | 16 | 20.91 | 18.55 | 77.48 | 18.39 | 24.14 | |
| | 64 | 56.48 | 50.38 | 188.22 | 7.17 | 14.59 | |
| Base [**CHB**] | 2 | 0.79 | 0.57 | 3.47 | 58.20 | 56.62 | |
| | 4 | 5.56 | 2.72 | 24.41 | 40.40 | 39.47 | |
| | 8 | 19.70 | 17.29 | 76.47 | 22.01 | 25.24 | Ours |
| | 16 | 38.51 | 35.99 | 138.30 | 12.83 | 18.44 | |
| | 64 | 74.53 | 77.03 | 246.71 | 4.25 | 12.14 | |
| Base [**BCI**] | 2 | 53.88 | 66.61 | 179.43 | 18.66 | 15.08 | |
| | 4 | 48.06 | 58.86 | 161.23 | 14.29 | 16.06 | |
| | 8 | 58.85 | 66.20 | 191.14 | 8.46 | 14.13 | Ours |
| | 16 | 67.96 | 76.13 | 223.31 | 5.30 | 12.86 | |
| | 64 | 90.94 | 95.46 | 289.01 | 1.35 | 10.23 | |
| GAN [**SWEC**] | 2 | 2.56 | 1.06 | 8.63 | 38.64 | 42.38 | |
| | 4 | 3.32 | 1.34 | 11.78 | 35.35 | 40.06 | |
| | 8 | 6.14 | 3.64 | 23.34 | 30.22 | 35.25 | Ours |
| | 16 | 16.14 | 11.32 | 60.02 | 19.46 | 26.44 | |
| | 64 | 57.16 | 39.31 | 187.63 | 6.45 | 14.47 | |
| GAN [**CHB**] | 2 | 1.18 | 0.88 | 4.96 | 51.26 | 52.11 | |
| | 4 | 3.84 | 3.03 | 16.08 | 41.12 | 41.97 | |
| | 8 | 12.38 | 8.79 | 48.49 | 24.42 | 29.51 | Ours |
| | 16 | 238.57 | 2555.31 | 896.74 | 14.64 | 12.48 | |
| | 64 | 72.99 | 83.32 | 240.56 | 4.55 | 12.35 | |
| GAN [**BCI**] | 2 | 55.59 | 71.96 | 180.73 | 13.06 | 14.69 | |
| | 4 | 44.07 | 58.14 | 147.30 | 11.98 | 16.73 | |
| | 8 | 73.77 | 79.87 | 243.40 | 5.88 | 12.21 | Ours |
| | 16 | 80.76 | 88.15 | 261.84 | 4.33 | 11.33 | |
| | 64 | 90.58 | 92.99 | 286.55 | 1.30 | 10.27 | |
| AAC | 3.49 | 3.65 | | | | | |
| | 6.44 | 12.01 | | | | | Reproduced (Nguyen et al., 2017) |
| | 13.86 | 40.75 | | | | | |
| NLSPIHT | 8 | 4.93 | | | | | Reproduced (Xu et al., 2015) |
| | 16 | 23.03 | | | | | |
| ANN [**BCI**] | 4.45 | 7.22 | | | | | Reported (Hejrati et al., 2017) |

Table A16: Full compression results on the S subset of the BONN EEG dataset.

# E  SUBJECTIVE EVALUATION OF THE COMPRESSION

We collect the subjective evaluations of an expert epileptologist on the quality of the signal reconstruction for seizure detection. Below we present the evaluations together with the evaluated samples, both for iEEG and EEG.

## E.1  IEEG

The quality of the reconstruction of iEEG signals allows us to push the compression ratio up to $64\times$ without loss of F1 performance, as shown previously. For this reason, we also subjectively evaluate the performance in the same regime.

**Base model.** Both original and reconstruction are plausibly biological signals, with no meaningful way to distinguish between the two without knowing of the comparison. Nonetheless, BrainCodec Base model can be seen acting as a low pass filter. This has been judged as not sufficiently influential, and would not impede seizure classification by a human expert.

**GAN model.** BrainCodec GAN is more successful in automatic seizure detection but is rated as having lower quality by the expert. It provides a better reconstruction of higher frequencies but also introduces some spurious high-frequency oscillations which pollute the signal (especially noticeable in Figure A19 and A20).

## E.2  EEG

EEG signals are noisier than their intracranial counterpart, making their compression more difficult as well.

In particular, $64\times$ compression on EEG creates a significant low pass effect. Moreover, the signal obtains a more stylized sinusoidal effect. Overall, $64\times$ compression is rated as having a significant quality drop with respect to the original signal by the expert.

For this reason, we also evaluate an $8\times$ compression. This achieves a higher fidelity level, comparable to the effects of the $64\times$ compression base model on iEEG. The expert evaluates the low-pass filtering effect as not influential for seizure classification.

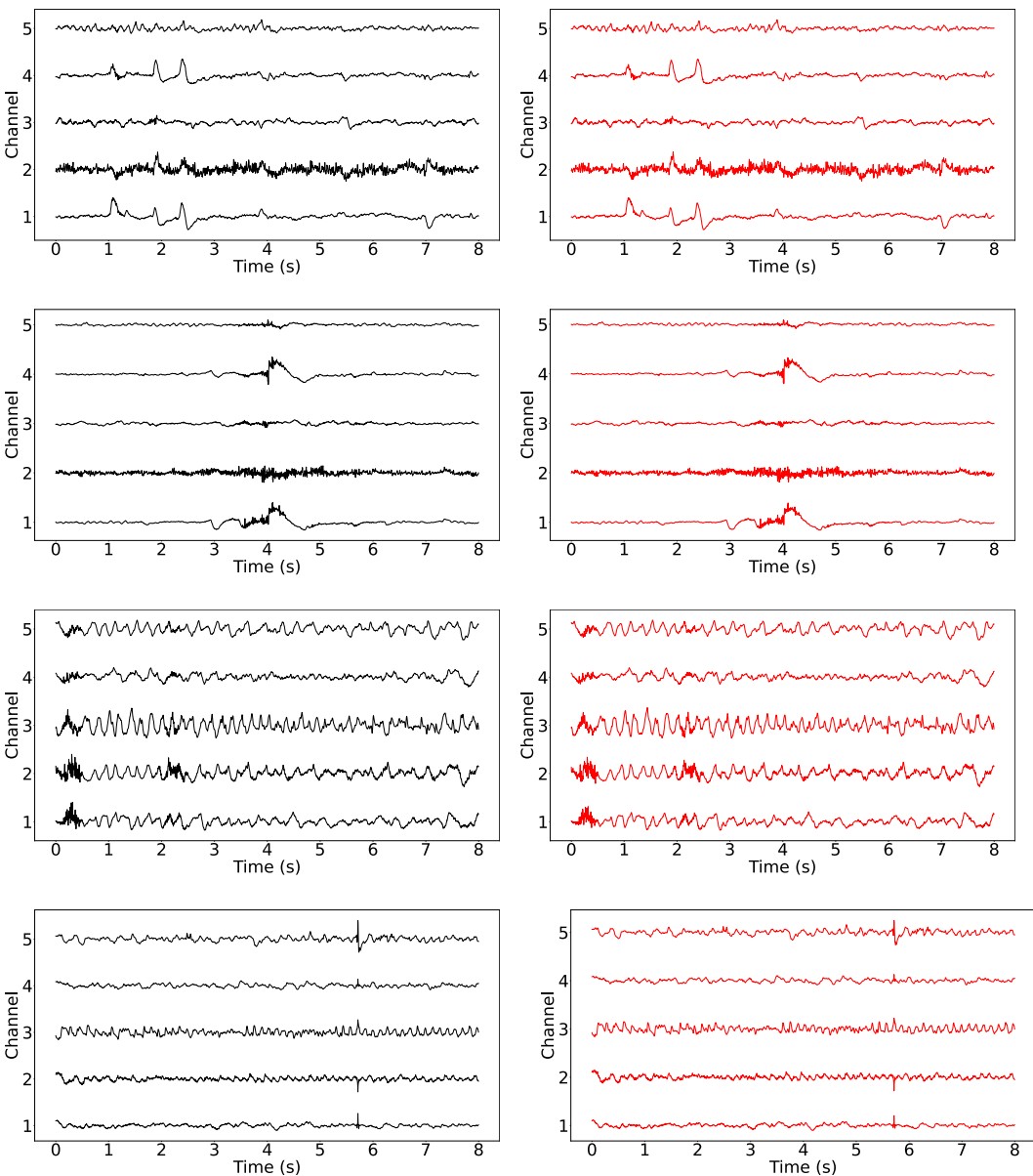

Figure A10: Reconstruction from samples of subject 2 of the CHB-MIT dataset by BrainCodec Base with $8\times$ compression ratio.

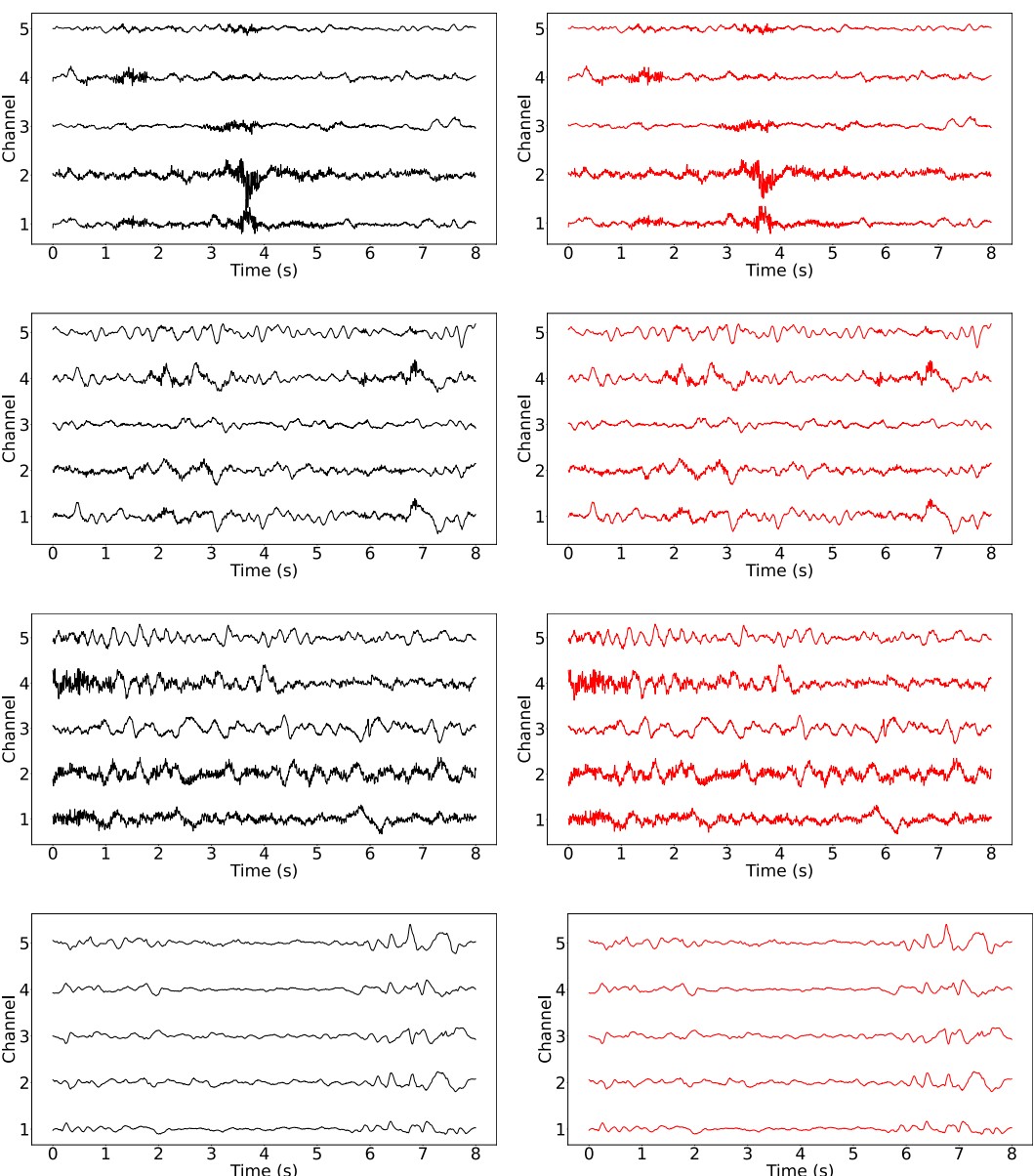

Figure A11: Reconstruction from samples of subject 5 of the CHB-MIT dataset by BrainCodec Base with $8\times$ compression ratio.

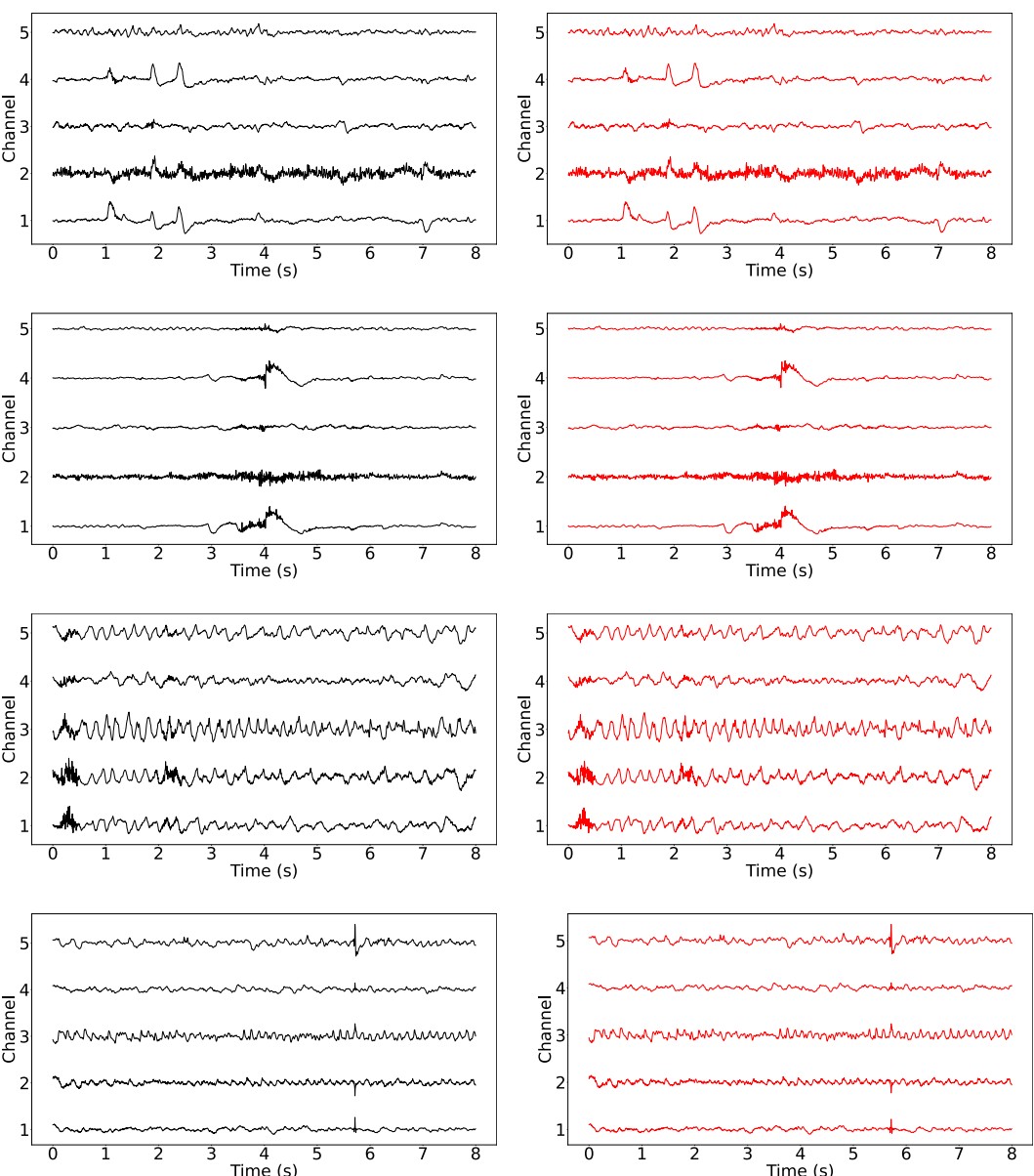

Figure A12: Reconstruction from samples of subject 2 of the CHB-MIT dataset by BrainCodec GAN with 8× compression ratio.

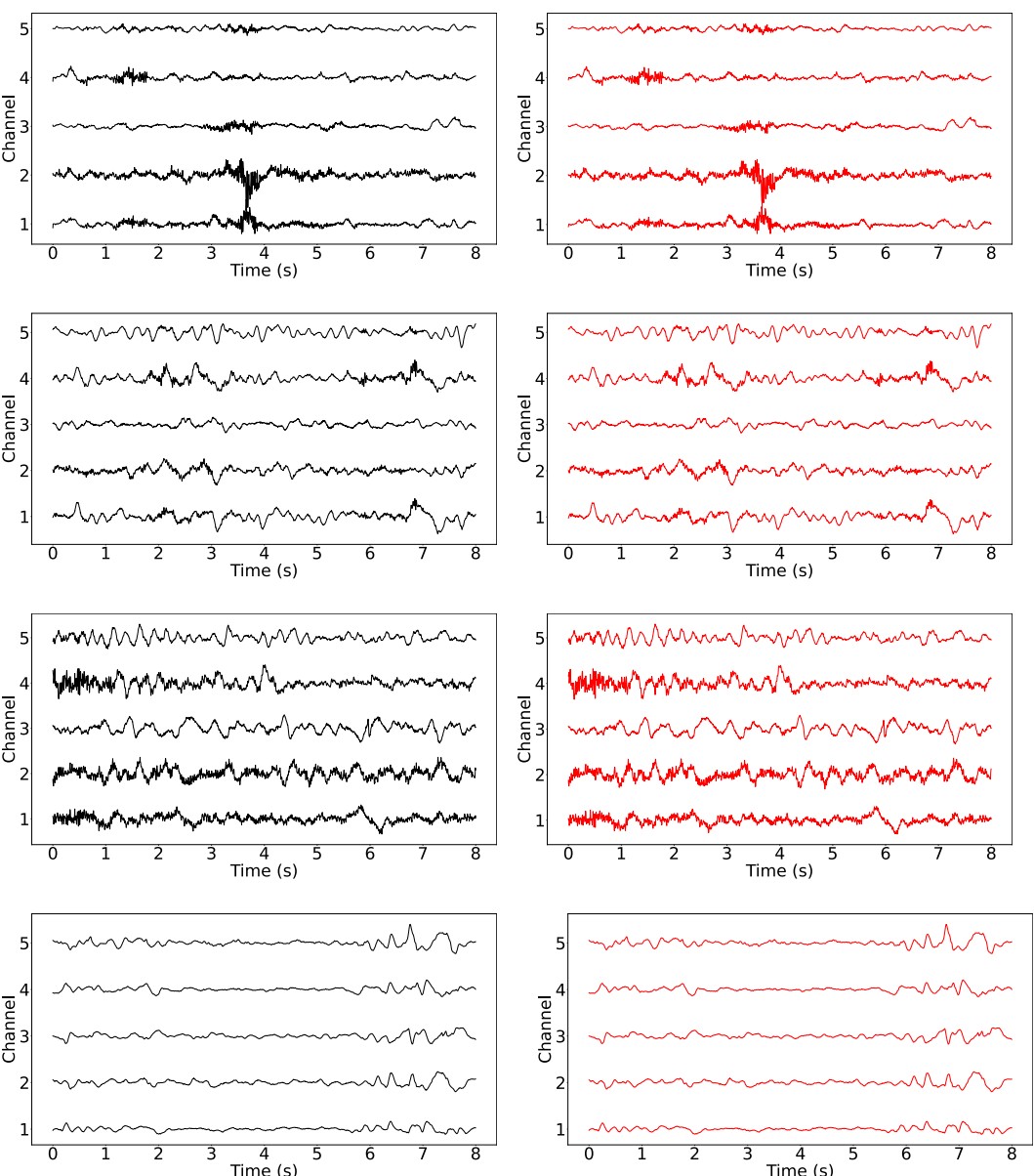

Figure A13: Reconstruction from samples of subject 5 of the CHB-MIT dataset by BrainCodec GAN with 8× compression ratio.

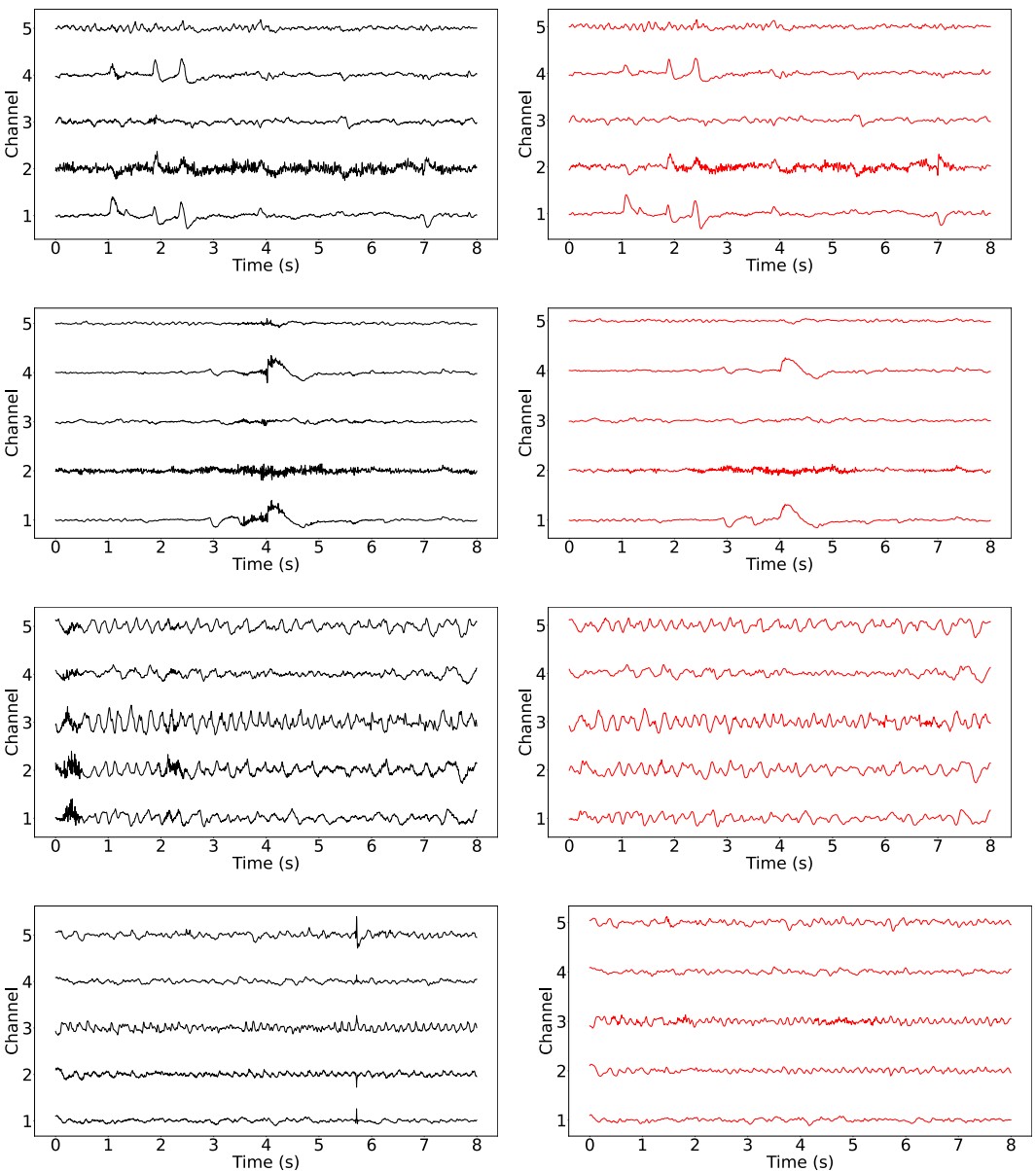

Figure A14: Reconstruction from samples of subject 2 of the CHB-MIT dataset by BrainCodec Base with 64× compression ratio.

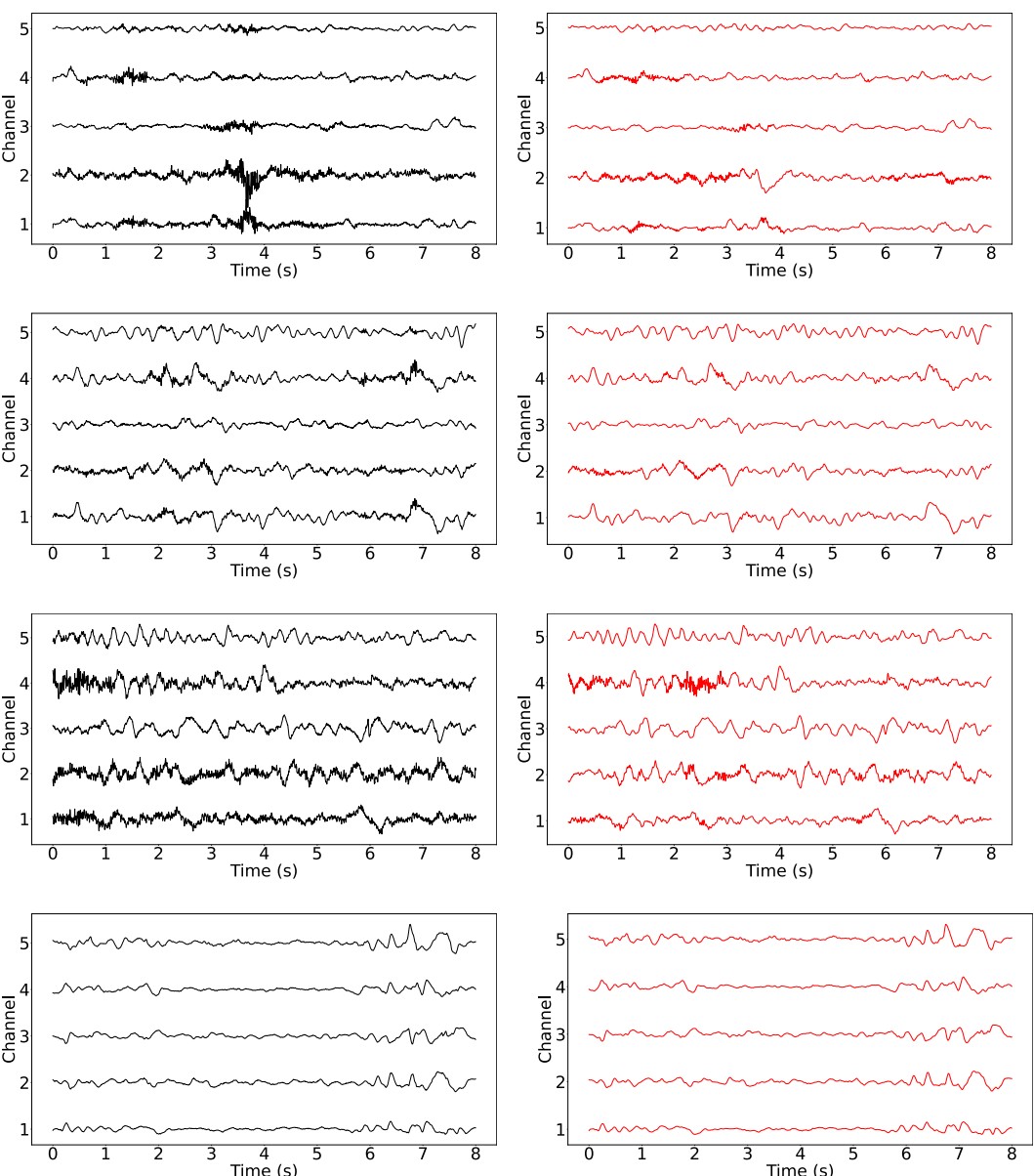

Figure A15: Reconstruction from samples of subject 5 of the CHB-MIT dataset by BrainCodec Base with 64× compression ratio.

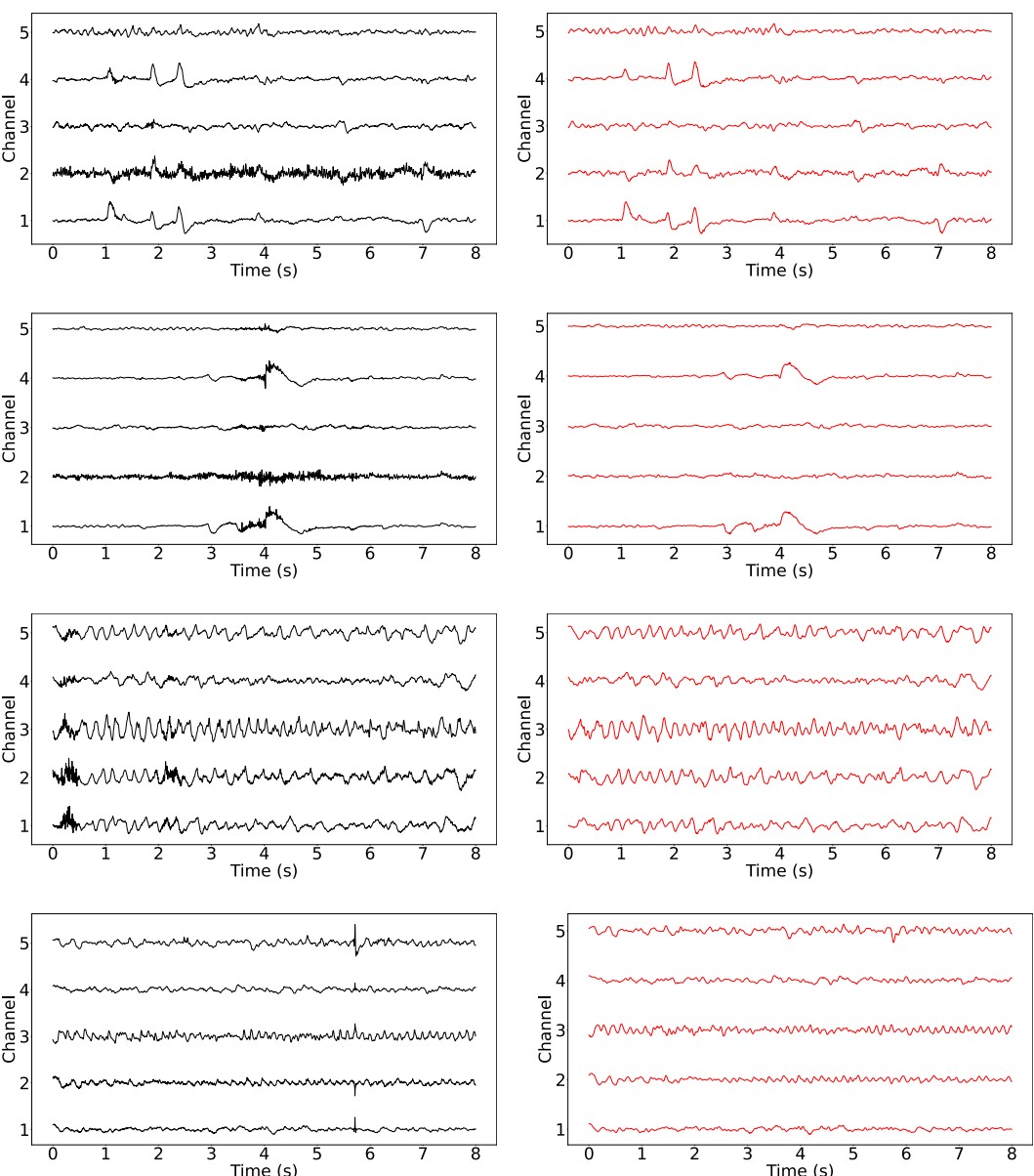

Figure A16: Reconstruction from samples of subject 2 of the CHB-MIT dataset by BrainCodec GAN with 64× compression ratio.

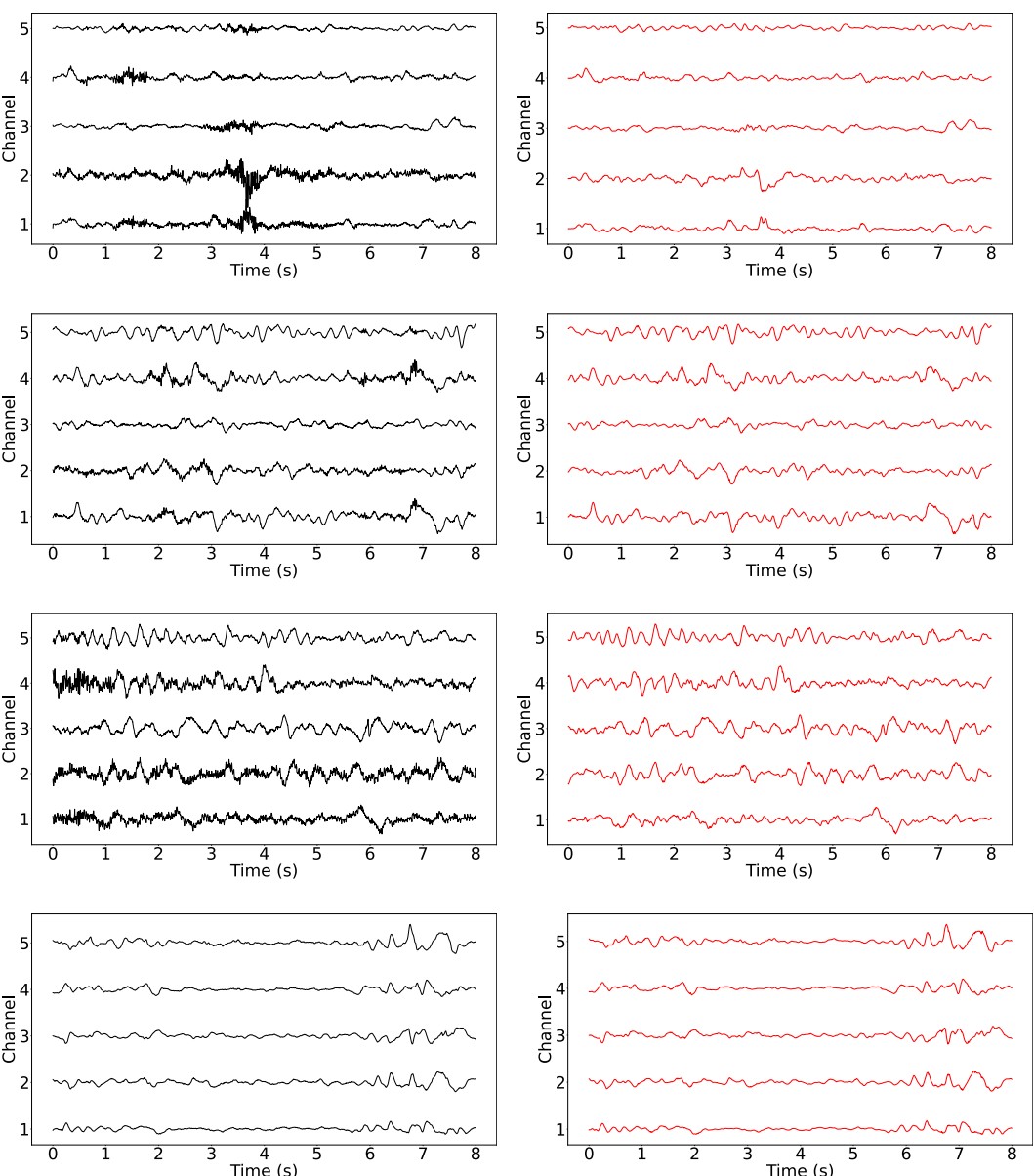

Figure A17: Reconstruction from samples of subject 5 of the CHB-MIT dataset by BrainCodec GAN with 64× compression ratio.

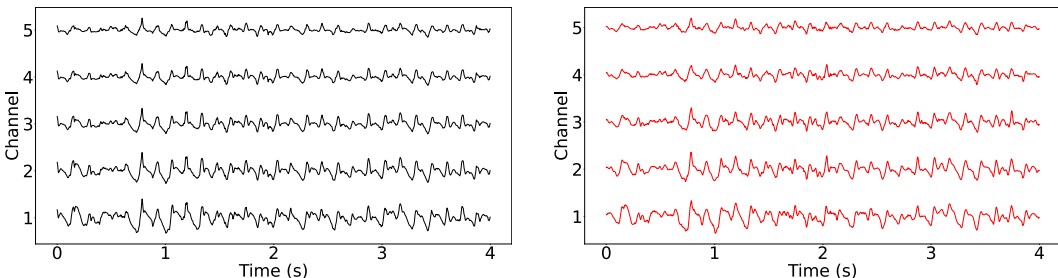

Figure A18: Reconstruction from samples of subject 2 of the SWEC dataset by BrainCodec GAN with 64× compression ratio.

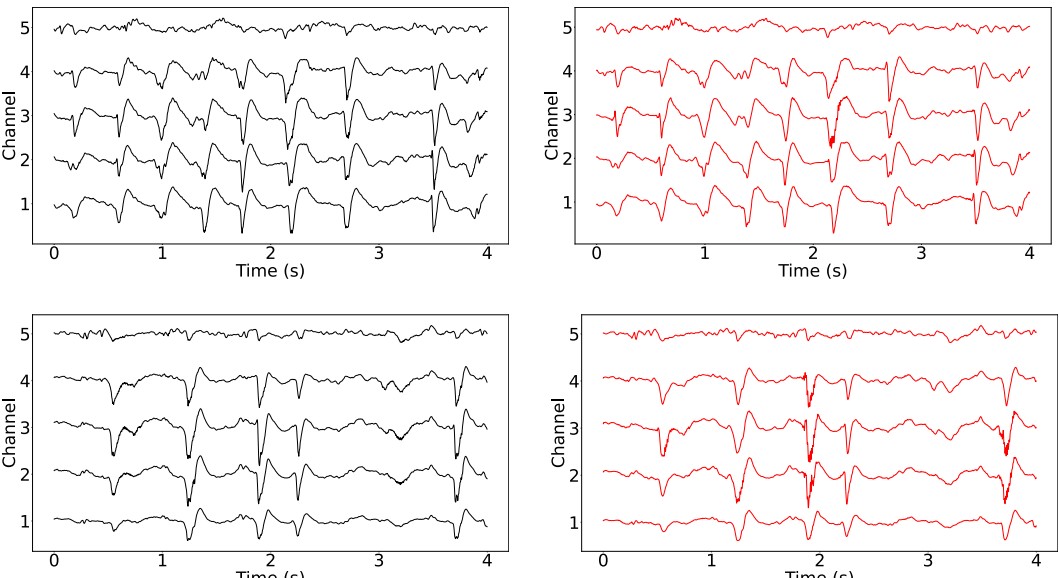

Figure A19: Reconstruction from samples of subject 4 of the SWEC dataset by BrainCodec GAN with 64× compression ratio.

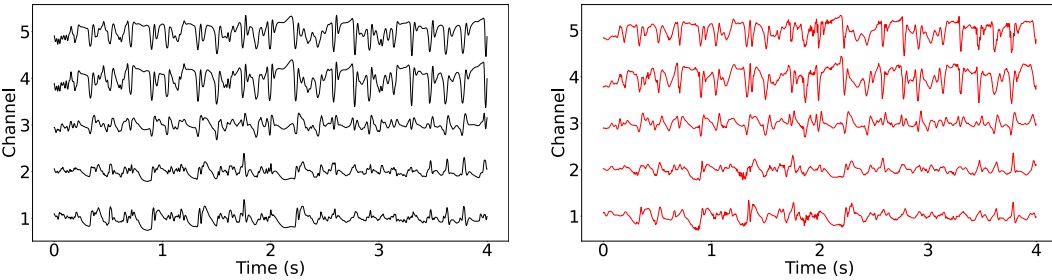

Figure A20: Reconstruction from samples of subject 6 of the SWEC dataset by BrainCodec GAN with 64× compression ratio.

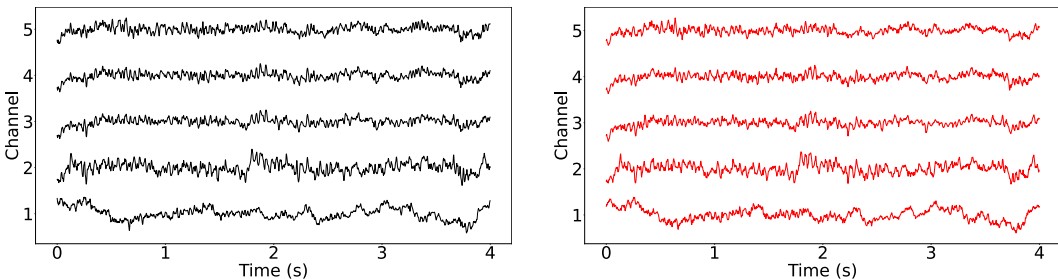

Figure A21: Reconstruction from samples of subject 7 of the SWEC dataset by BrainCodec GAN with 64× compression ratio.

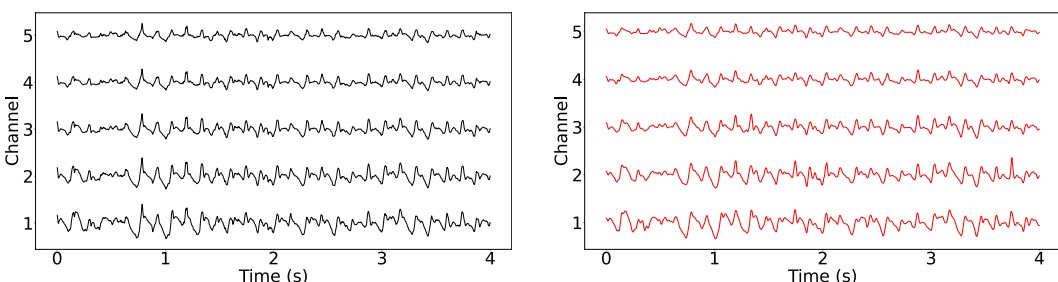

Figure A22: Reconstruction from samples of subject 2 of the SWEC dataset by BrainCodec Base with 64× compression ratio.

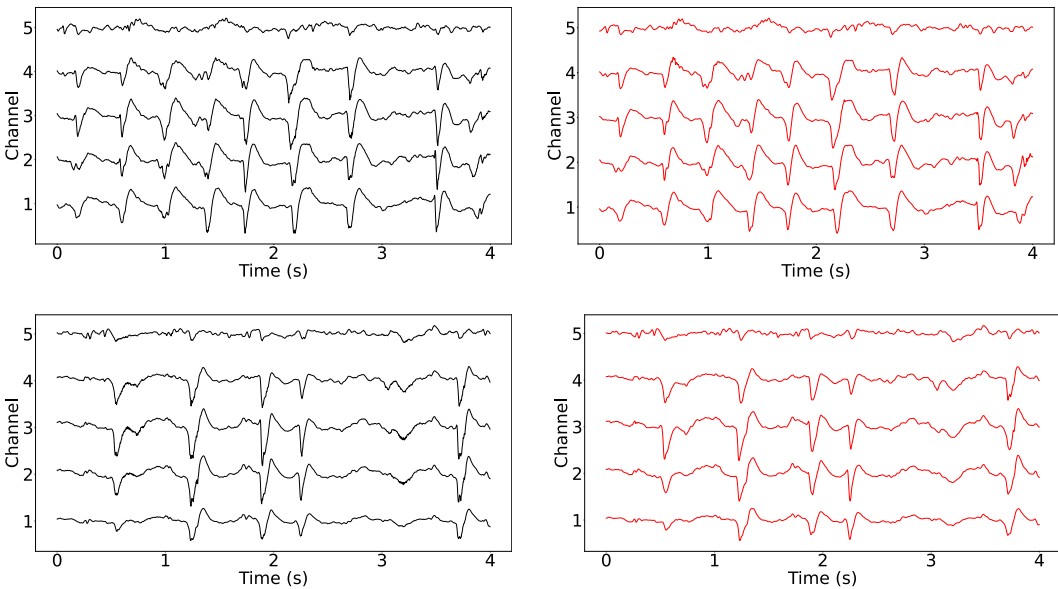

Figure A23: Reconstruction from samples of subject 4 of the SWEC dataset by BrainCodec Base with 64× compression ratio.

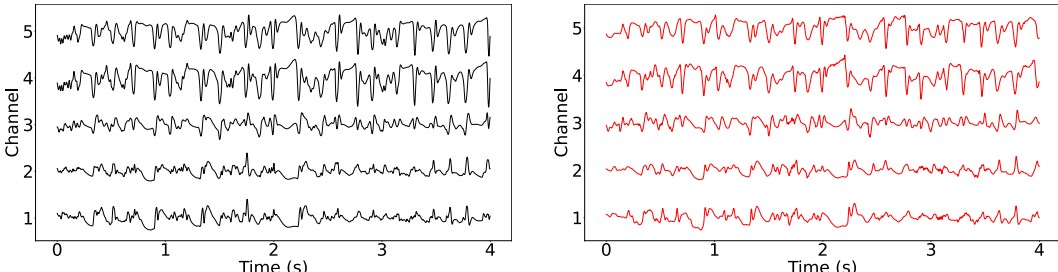

Figure A24: Reconstruction from samples of subject 6 of the SWEC dataset by BrainCodec Base with 64× compression ratio.

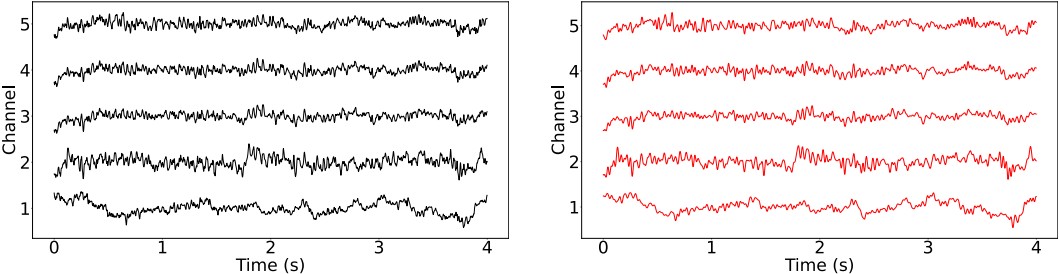

Figure A25: Reconstruction from samples of subject 7 of the SWEC dataset by BrainCodec Base with 64× compression ratio.

