# OpenReview forum: "The Case for Cleaner Biosignals: High-fidelity Neural Compressor Enables Transfer from Cleaner iEEG to Noisier EEG"
_ICLR.cc/2025/Conference — ICLR 2025 Poster_

### Official Review · Reviewer_cGqC · 2024-10-28

**Soundness:** 4
**Presentation:** 4
**Contribution:** 3
**Rating:** 6
**Confidence:** 5

**Summary:**

This paper introduces BrainCodec, a novel neural network-based compressor for EEG and iEEG signals. BrainCodec demonstrates promising compression ratios of up to 64x while achieving relatively high reconstruction fidelity, particularly when transferring learning from iEEG to EEG.

**Strengths:**

The core contribution of training a compressor on high-SNR iEEG data and applying it to low-SNR EEG data is novel and directly addresses a critical challenge in neuroelectrophysiological signal processing. The demonstrated improvement in reconstruction fidelity through this transfer learning is a significant finding.

**Weaknesses:**

1) The reliance on PRD as the primary metric for evaluating reconstruction fidelity is inadequate. PRD, being a global measure, can mask localized distortions that may significantly affect downstream task performance. Subtle yet clinically relevant distortions in the reconstructed signals could easily go undetected by PRD, potentially impairing the accuracy of subsequent analyses.

2) The ablation study is incomplete. While the authors mention exploring different architectural choices, they do not present the results of such explorations. A systematic investigation varying key architectural parameters (e.g., quantizer types, encoder/decoder depths and kernel sizes, activation functions) is necessary to understand their impact on compression and reconstruction quality, providing a stronger justification for the chosen architecture.

**Questions:**

1) What plans do the authors have to conduct a more in-depth analysis comparing the baseline BrainCodec with its GAN-augmented version, particularly in terms of frequency-specific characteristics of the reconstructed signals and their impact on downstream tasks?

2) How does the authors’ approach to evaluation change if they incorporate comparisons with established decoders from the literature, such as those cited in Zhang et al. (2023)? Would this increase the robustness and generalizability of the proposed compression technique?

3) In light of the recommendation to replace PRD with a more objective measure of signal quality derived from downstream motor imagery classification performance, how do the authors intend to implement this evaluation?

4) Given the suggestion to shift the emphasis from PRD to more direct assessments of classification accuracy, how might this impact the interpretation of BrainCodec's effectiveness in real-world applications? What additional data or analyses might be needed to support this shift in evaluation focus?

Reference:
Zhang, Y., Qiu, S., & He, H. (2023). Multimodal motor imagery decoding method based on temporal spatial feature alignment and fusion. Journal of Neural Engineering, 20(2), 026009.

---

> ### Author Response · Authors · 2024-11-19
>
> We thank the Reviewer for highlighting the compression performance of BrainCodec and acknowledging the concrete challenge represented by neurophysiological signals.
>
> We have updated the manuscript to include more evaluation metrics and additional ablations that better characterize the model's performance.
>
> -------------
>
> ## W1. Evaluation metrics
>
> We choose three separate evaluation metrics to better characterize the performance of BrainCodec:
>
> 1. **Percent-root-mean-square-difference (PRD).** PRD is the most used metric to determine the reconstruction fidelity of compressors on EEG and iEEG, as seen in the literature [1,2,3].
> 2. **Downstream classification performance.** Classification tasks are a concrete benchmark for the performance of BrainCodec, as they provide an immediately parsable metric of the distortion caused by the compression.
> 3. **Subjective expert evaluation.** The specific characteristics of EEG and iEEG are not as easily parsed as music or speech. Therefore, we assess the reconstruction fidelity through the expert evaluation of a neurologist who works with this data every day and is best poised to capture the fine difference between the original and the compressed signal.
>
> Following the Reviewers' suggestion, we now include four additional metrics borrowed from the signal processing domain, namely:
> 1. **PRD of the spectrogram (PRD-spec)**
> 2. **Root-mean-square error (RMSE)**
> 3. **Signal-to-noise ratio (SNR)**
> 4. **Peak signal-to-noise ratio (PSNR)**
>
> All tables in Appendix D have been updated accordingly, showing that BrainCodec's high reconstruction fidelity translates to these four additional evaluation metrics as well.
>
> Briefly here as an example, we report all the metrics collected for the Base model trained and tested on the iEEG SWEC dataset.
>
> |CR ↑  |PRD ↓ |PRD-spec ↓|RMSE ↓|SNR ↑  |PSNR ↑ |
> |----|----|--------|----|-----|-----|
> |2   |0.66|0.81    |1.67|57.60|70.14|
> |4   |0.94|1.05    |2.12|50.95|64.89|
> |8   |1.88|1.46    |3.11|39.86|55.12|
> |16  |4.60|3.39    |6.33|31.08|46.00|
> |64  |15.76|12.35   |16.69|18.01|33.84|
>
> [1] Higgins, Garry, et al. "EEG compression using JPEG2000: How much loss is too much?." 2010 Annual International Conference of the IEEE Engineering in Medicine and Biology. IEEE, 2010.
>
> [2] Du, XiuLi, et al. "Fast reconstruction of EEG signal compression sensing based on deep learning." Scientific Reports 14.1 (2024): 5087.
>
> [3] Nguyen, Binh, et al. "Wavelet transform and adaptive arithmetic coding techniques for EEG lossy compression." 2017 international joint conference on neural networks (IJCNN). IEEE, 2017.
>
> ------------
>
> ## W2. Additional ablations
>
> We strive to validate BrainCodec in many scenarios to improve its real-world applicability. As suggested by the Reviewer we have now included ablations on the following:
> 1. **Number of residuals.** We assess the performance at varying compression ratios by varying RVQ's number of residuals from 1 to 16 (i.e., 16 to 256 compression ratio). For our main results, we vary the Encoder framerate instead, because we find it yields superior results. For example, at 16 compression ratio we report a PRD of 4.60 against 8.74, a notable improvement.
> 2. **Codeword size.** We vary the size of each codeword from 64 to 1024 (i.e., a compression ratio of ~51 to ~85). This gives diminishing returns, with our choice of 256 a consistent middle-ground.
> 3. **Line length loss.** We train a model without the line length loss. We find that the performance is decreased, with a PRD of 15.95 against 15.76. However, the difference is especially noticeable in the reconstruction of higher frequencies in the spectrogram, as reported in Appendix B.4.
> 4. **Kernel size.** We vary the initial kernel size from 3 to 7, and show that our choice (3) yields the best results.
> 5. **Number of patients.** We train additional models using 2 and 4 training patients. The results once again point to diminishing returns, indicating that our single-patient training already produces high-fidelity reconstructions, as highlighted by the Reviewer.
>
> We report the most relevant ablations here. First, varying the number of residuals yields worse performance with respect to varying the Encoder framerate.
>
> | Residuals | CR ↑  |PRD ↓ | SNR ↑  |
> |-----------|-----|-------|-------|
> | 1         | 256 | 38.36 | 9.67  |
> | 4         | 64  | 15.76 | 18.01 |
> | 8         | 32  | 10.26 | 22.04 |
> | 16        | 16  | 8.74  | 23.76 |
>
> Second, the line length loss improves PRD, but most importantly improves the reconstruction of higher frequencies, as seen in Appendix B.4.
>
> | Line length loss | CR ↑  |PRD ↓ | SNR ↑  |
> |------------------|----|-------|-------|
> | Yes              | 64 | 15.76 | 18.01 |
> | No               | 64 | 15.95 | 17.33 |
>
> Overall, our results show that our choices of modules and objectives --- consistent with the characteristics of EEG and iEEG signals --- improve the reconstruction fidelity of BrainCodec and strengthen its effectiveness.

---

> > ### Author Response · Authors · 2024-11-19
> >
> > ## Q1. In-depth evaluation of GAN vs Base
> >
> > We present three main findings that we believe are interesting in the evaluation of the adversarial training.
> >
> > First, the GAN model has a superior performance in the downstream classification tasks, especially at high compression ratios (see Appendix C.3). This, coupled with the fact that the GAN model reconstructs high frequencies with greater fidelity, indicates that the downstream ML models rely heavily on these frequencies.
> >
> > Second, and directly related, the above is not the case for human evaluators. While debate on the importance of high frequencies in ictal presentations is ongoing, it appears that epileptologists rely on lower frequencies to make their assessments. Consistent with this message, our human expert evaluator found Base model reconstructions to be more useful, as they did not contain high frequency artifacts that might muddle the signal (e.g., by masquerading as HFOs).
> >
> > These two findings already provide some evidence as to how humans and machines differ in the way they make decisions.
> >
> > Third, we find that a single parameter (lambda_t) mediates between the GAN and Base models. This suggests the existence of a budget trade-off between temporal and frequency reconstruction. We found some initial evidence, but we plan to further study the phenomenon to fully characterize it.
> >
> > -------------
> >
> > ## Q2. Additional decoders
> >
> > Neural compression models have been gathering a significant amount of interest and effort, with a particular focus on the architecture of Encoders and Decoders[4]. Therefore, we have good reason to believe that employing Decoder models specifically developed for the data modalities at hand --- such as the one the Reviewer suggested --- is going to increase both performance and overall generalization capabilities across subjects and tasks.
> >
> > [4] Siuzdak, Hubert. "Vocos: Closing the gap between time-domain and Fourier-based neural vocoders for high-quality audio synthesis." arXiv preprint arXiv:2306.00814 (2023).
> >
> > -------------
> >
> > ## Q3. Motor imagery inspired metric
> >
> > In this work, we propose a composite metric that incorporates both the reconstruction fidelity (PRD) and the classification (task performance) aspects of the compression problem. Moreover, we also include a subjective fidelity evaluation performed by an expert neurologist, who is most able to distinguish the fine details which characterize the electrophysiological signal.
> >
> > Following the suggestion of the Reviewer and as mentioned previously in **W2**, we now also evaluate BrainCodec using the PRD of the spectrogram, the RMSE, the SNR, and the PSNR. This extensive suite of metrics improves the overall characterization of our model's behavior. Nonetheless, we are available to implement other metrics if they are better suited to the task.
> >
> > ------------
> >
> > ## Q4. Real-world applications
> >
> > We place great emphasis on the evaluation of BrainCodec's performance in real-world scenarios. In fact, we believe our model might be of immediate benefit in clinical and research environments to reduce storage and communication pressures, especially for ultra-long-term recordings. For this reason, we evaluate the loss of performance caused by our compressor on relevant downstream tasks. Moreover, we also provide the subjective evaluation of an expert neurologist, who would be the real-world end-user of our model. BrainCodec succeeds in both benchmarks with performance that surpasses the state-of-the-art and strengthens our suggestions on the applicability of BrainCodec.
> >
> > At the same time, the best avenue to show real-world applicability is to test our model in concrete scenarios. In particular, we are working to expand our assessment with more tasks, more datasets, and more heterogeneous environments. Already for this rebuttal, we include a new iEEG dataset, the Brain Treebank iEEG dataset, to our testing suite following the suggestion of a reviewer. This brings our total to 3 EEG datasets and 3 iEEG datasets. We find that the performance on this dataset is comparable to the SWEC iEEG dataset.
> >
> > | Treebank | CR ↑  |PRD ↓ | SNR ↑ |
> > |---------------|----|--------|-------|
> > | Base [SWEC]            | 2 | 3.32  | 34.58 |
> > |            | 4 | 3.54  | 33.70 |
> > |            | 8 | 3.89  | 32.56 |
> > |            | 16 | 5.26  | 28.92 |
> > |           | 64 | 14.36  | 18.89 |

---

### Official Review · Reviewer_HbiL · 2024-11-01

**Soundness:** 3
**Presentation:** 3
**Contribution:** 3
**Rating:** 8
**Confidence:** 4

**Summary:**

The authors introduce RVQ-based neural signal compressor, BrainCodec, which achieves a $64\times$ compression on (i)EEG signals without a notable decrease in quality. The authors also indicate that clean data sources appear to have an outsized effect on the performance of DL model -- training BrainCodec on iEEG and then transferring to EEG yields higher reconstruction quality than training on EEG directly.

**Dataset**: Two iEEG datasets (i.e., SWEC iEEG, MC iEEG), and three EEG datasets (i.e., CHB-MIT, BONN, BCI Competition IV-2a) are used for this study. The downstream classification tasks include 2-way seizure detection (on iEEG datasets) and 4-way motor imagery (on BCI Competition IV-2a).

**Model**: The authors propose BrainCodec, which treats different channels independently. BrainCodec utilizes a stack of convolution layers to compress the raw (i)EEG signals into embeddings, and further quantizes them through Residual Vector Quantization.

**Experiement**: The authors utilize the reconstructed (i)EEG signals for downstream classification tasks.

**Strengths:**

**Significance**: This study highlights the importance of clean data for pre-training. The authors find that training BrainCodec on iEEG and then transferring to EEG yields higher reconstruction quality than training on EEG directly. This finding may help the development of some VQ-based EEG foundation models (e.g., LaBraM[1]), as iEEG signals can further improve the representation quality compared to EEG signals.

**Clarity:** The text has a good structure and is well-written. The figures also help in understanding the method.

**Reference**

[1] Jiang W B, Zhao L M, Lu B L. Large brain model for learning generic representations with tremendous EEG data in BCI[J]. arXiv preprint arXiv:2405.18765, 2024.

**Weaknesses:**

**Major**

1. It seems the compression ratio only depends on the depth of the Encoder (Line 201). How about the effect of different configurations of RVQ? Could the authors provide some ablation studies over the configurations of RVQ (i.e., the codex size, the dimensions of each code, etc.)?

2. As AE+RVQ has a certain ability for representation learning, how about the classification performance from the quantized embeddings (after quantizer)?

**Minor**

1. I’m not sure whether the amount of iEEG datasets is enough, as the SWEC iEEG dataset only contains 14 hours of recording (across 15 subjects). Additional publicly available iEEG datasets the authors should be aware of:
   - Brain TreeBank (https://neurips.cc/virtual/2024/poster/97751): Their dataset (https://braintreebank.dev/) contains 43 hours of iEEG recording (not preprocessed).
   - Du-IN (https://arxiv.org/abs/2405.11459): Their dataset contains 36 hours of iEEG recording (after bipolar reference).

    To further prove the advantage of pre-training on clean data source (i.e., iEEG) -- **training BrainCodec on iEEG and then transferring to EEG yields higher reconstruction quality than training on EEG directly**, could the authors provide results of the model pre-trained on both SWEC iEEG dataset and Brain TreeBank? Besides, could the authors provide the effect of different reference methods (e.g., laplacian/bipolar reference instead of median reference, Line 274)?

2. More downstream classification can be included, e.g., 4 classification tasks in BrainBERT[1] based on the Brain TreeBank dataset.

**Reference**:

[1] Wang C, Subramaniam V, Yaari A U, et al. BrainBERT: Self-supervised representation learning for intracranial recordings[J]. arXiv preprint arXiv:2302.14367, 2023.

**Questions:**

None

---

> ### Author Response · Authors · 2024-11-19
>
> # Reviewer HbiL
>
> We thank the Reviewer for acknowledging the impact of our findings regarding the importance of data quality in the development of EEG foundational models.
>
> We have updated the manuscript to include new ablations on the RVQ component, a new iEEG dataset (Brain Treebank), and an ablation with respect to the reference schema.
>
> -------------
>
> ## W1. RVQ ablations
>
> We have chosen to vary the compression ratio based on the framerate of the Encoder because it yielded better performance and flexibility during our initial evaluation.
>
> We now perform a systematic assessment (included in Appendix B.3), in which we vary both the number of residuals from 1 to 16 (i.e., a compression ratio of 16 to 256), and the size of each code-word from 64 to 1024 (i.e., a compression ratio of ~51 to ~85).
>
> | Residuals | CR ↑  |PRD ↓ | SNR ↑ |
> |-----------|-----|-------|-------|
> | 1         | 256 | 38.36 | 9.67  |
> | 4         | 64  | 15.76 | 18.01 |
> | 8         | 32  | 10.26 | 22.04 |
> | 16        | 16  | 8.74  | 23.76 |
>
> | Codebook size | CR ↑  |PRD ↓ | SNR ↑ |
> |---------------|----|--------|-------|
> | 64            | 85 | 21.09  | 15.47 |
> | 128           | 73 | 18.16  | 16.79 |
> | 256           | 64 | 15.76  | 18.01 |
> | 512           | 57 | 13.95  | 19.02 |
> | 1024          | 51 | 12.74  | 19.77 |
>
> The performance at equivalent compression is notably inferior to our choice of varying the framerate.
>
> -------------
>
> ## W2. Quantized embeddings
>
> Inspired by the promising results of VQ-based models we also tried to use our embeddings for the seizure classification task. Unfortunately, an initial evaluation yielded unsatisfactory results.
>
> However, we are currently working towards fully integrating BrainCodec into this new and exciting area of research, but unfortunately, such immediate efforts are outside the scope of this manuscript.
>
> --------------
>
> ## W3. Additional iEEG datasets
>
> We wish to remark that, as the Reviewer correctly pointed out in their Summary, we evaluate two iEEG datasets, the SWEC and the MC iEEG dataset, for a total of 106 subjects.
>
> We are also always excited to find new high-quality datasets to include in our benchmarks. Following the Reviewer's suggestion, we have found the Brain Treebank dataset to be highly well-curated and now include it in our results, bringing us to a total of 3 iEEG datasets and 3 EEG datasets. Unfortunately, timing and resource constraints prevent us from also evaluating the Du-IN dataset.
>
> We present the performance of our BrainCodec trained on the SWEC dataset in Appendices C and D. The results are comparable to testing on the SWEC dataset itself and show once more that our model generalizes well to new datasets.
>
> | Treebank | CR ↑  |PRD ↓ | SNR ↑    |
> |---------------|----|--------|-------|
> | Base [SWEC]            | 2 | 3.32  | 34.58 |
> |            | 4 | 3.54  | 33.70 |
> |            | 8 | 3.89  | 32.56 |
> |            | 16 | 5.26  | 28.92 |
> |           | 64 | 14.36  | 18.89 |
>
> Moreover, as requested by the Reviewer we also train a model on both SWEC and TreeBank datasets jointly and evaluate it on both the SWEC and TreeBank datasets separately to compare it with other models. The evaluation of this model has not been completed yet, but we commit to including it as soon as possible.
>
> --------------
>
> ## W4. Reference schema ablation
>
> We thank the Reviewer for pointing us towards the un-processed Brain Treebank dataset, which allows us to also evaluate the effects of post-processing on our compressor.
>
> We now include in Appendix B.1 the reconstruction results of the Brain Treebank dataset for a median, bipolar, and Laplacian (referenced to the mean of the electrode group) reference, and also no reference schema.
>
> | Reference | CR ↑  |PRD ↓ | SNR ↑    |
> |-----------|----|-------|-------|
> | None      | 64 | 15.21 | 18.49 |
> | Median    | 64 | 14.67 | 18.36 |
> | Bipolar   | 64 | 14.36 | 18.89 |
> | Laplacian | 64 | 16.56 | 18.96 |
>
> All results are rather similar, with median and bipolar referencing performing the best. However, the effect of the reference schema is mostly influenced by the collection of the dataset itself, rather than the compressor architecture. Overall, the reference schema does not notably influence the reconstruction fidelity of BrainCodec.
>
> -------------
>
> ## W5. BrainBert tasks
>
> We prioritized running additional experiments such as ablations, adding the new iEEG dataset, and creating a combined EEG + iEEG model (see **W1** of Reviewer p69F), which consumed 16x Nvidia A100 for a week. Due to the limited amount of time to respond, we were not able to evaluate the BrainBert task.

---

> > ### Comment · Reviewer_HbiL · 2024-11-19
> > **Official Comment by Reviewer HbiL**
> >
> > Thank you for the responses. These results help. The results of **W1&W2** are consistent with my previous experiments. I have raised my score to 8, conditional on all ablation results (in W1 & W3 & W4) being added to the paper.

---

> > > ### Author Response · Authors · 2024-11-20
> > >
> > > We thank the Reviewer for increasing our scores and for providing valuable corroboration of our results. We confirm that the ablation results are already present in the revised and uploaded version of the Manuscript, pending the latest scores for the Brain Treebank dataset. We will update both the Manuscript and our response as soon as the evaluation is complete.

---

> ### Author Response · Authors · 2024-11-26
>
> We have now revised the manuscript to include the full results on the Treebank dataset as well, including models trained on both the SWEC and the Brain Treebank dataset. Below are the results in this scenario for the Base model:
> | Treebank      | CR ↑   | PRD ↓  |  SNR ↑  |
> | :------------------- | :---------- | :--------- | :---------- |
> | Base [SWEC+Treebank] | 2           | 4\.99      | 27\.39      |
> |                      | 4           | 8\.67      | 22\.52      |
> |                      | 8           | 11\.40     | 19\.93      |
> |                      | 16          | 14\.34     | 18\.16      |
> |                      | 64          | 15\.04     | 18\.37      |

---

> > ### Comment · Reviewer_HbiL · 2024-11-26
> > **Official Comment by Reviewer HbiL**
> >
> > Thanks for these additional results. I have no additional concerns. Good luck :)

---

### Official Review · Reviewer_Wj41 · 2024-11-01

**Soundness:** 3
**Presentation:** 3
**Contribution:** 3
**Rating:** 8
**Confidence:** 3

**Summary:**

The authors propose a method to map brain signals --- invasive (iEEG) and non-invasive (EEG) --- to representations that live in a lower-dimensional, discrete space. They do adapting a model designed for audio (Zeghidour et al., 2022 and Defossez et al., 2023): that model is a convolutional auto-encoder where the representation is quantized. The authors test the "quality" of the representations using two metrics: first, how well they can reconstruct the original signal and second, and how well the reconstructed signals predict downstream information. In both cases, compared to other methods, the authors are able to obtain higher metrics using lower-dimensional representations, which means effectively better compression.

**Strengths:**

The paper is very clearly written and is impactful. The authors convincingly make the point the representations learnt using the cleaner iEEG signals are useful to reconstruct the noisier EEG signals, in fact, more so than if the representations were learnt using EEG signals in the first place. This is an instance of favorable generalization from one data modality (iEEG) to another (EEG). Their comparison of different cost functions for training, adversarial and non-adversarial, is interesting, especially as it does not conclude that adversarial training is always beneficial.

**Weaknesses:**

- The authors "train the model on one single subject, and test the reconstruction on the remaining subjects". I am surprised that the authors obtain near-zero relative error in Figures 3 and 4. In EEG, the differences between subjects can be quite high, due to different physiologies, electrode positions, and reponse delays. So one would expect training on one subject and testing on another to lead to low-quality reconstruction. Can the authors comment on this surprising point?

- It would be relevant to include [1] in related works, as it uses a very similar model with the same name but applied to fMRI data.

[1] BrainCodec: Neural fMRI codec for the decoding of cognitive brain states. Yuto Nishimura, Masataka Sawayama, Ayumu Yamashita, Hideki Nakayama, Kaoru Amano. 2024.

**Questions:**

Could the authors address the points in the "weaknesses" section?

---

> ### Author Response · Authors · 2024-11-19
>
> We thank the Reviewer for highlighting the strength of our cross-modality results and for acknowledging the potential impact of our findings about the advantages and disadvantages of adversarial training.
>
> We have updated the manuscript to clarify our hypotheses about the performance of BrainCodec at low compression ratios.
>
> --------------
>
> ## W1. Near-perfect performance at low ratios
>
> The generalisation capabilities of BrainCodec are indeed remarkable, as the Reviewer aptly highlighted.
>
> We hypothesise that the *vocabulary* of brain states itself is consistent across patients, even if the recordings appear very heterogeneous. We now include the results obtained by training on 1, 2, and 4 patients in Appendix B.5, which show that performance does not increase markedly as the number of training patients increases. This indicates that the *vocabulary* is rather stable across patients and does not get stronger in representation power.
>
> | Training patients | PRD ↓ |PRD-spec ↓|RMSE ↓|SNR ↑  |PSNR ↑    |
> | :---------------- | :----- | :------- | :----- | :----- | :----- |
> | 1                 | 15\.56 | 10\.73   | 16\.83 | 18\.26 | 34\.41 |
> | 2                 | 14\.31 | 7\.36    | 13\.56 | 18\.58 | 34\.97 |
> | 4                 | 13\.96 | 7\.37    | 13\.45 | 18\.79 | 35\.2  |
>
> Moreover, we hypothesise that cross-patient consistency also increases as the framerate of the compression increases. When the length of each frame decreases also its heterogeneity decreases, lending itself better to being quantized. Since we decrease the compression ratio by increasing the framerate --- this empirically yields better performance rather than, e.g., decreasing the number of residuals (see Appendix B.3) ---, our results at low ratios validate this hypothesis.
>
> Finally, as a last point of interest, we notice that BrainCodec's performance at lower compression ratios is better in set S of the BONN dataset with respect to all other sets. Following our above hypotheses, this would indicate that the brain states in set S are more consistent than other sets. Indeed set S only contains ictal activity, and the literature[1] has shown that seizures often manifest in highly regular patterns.
>
> [1] Schindler, Kaspar, et al. "Forbidden ordinal patterns of periictal intracranial EEG indicate deterministic dynamics in human epileptic seizures." Epilepsia 52.10 (2011): 1771-1780.
>
> -----------
>
> ## W2. fMRI BrainCodec
>
> We thank the Reviewer for pointing to this model with similar naming that compresses fMRI data. While this related work has been published after our submission deadline, we still would like to point out some differences, apart from the different modality:
>
> 1. The fMRI BrainCodec compresses sequences of 1024-dimensional samples with one model pass. In contrast, our (i)EEG BrainCodec compresses each channel separately, i.e., we feed sequences of 1-dimensional data into the compressor. This way, we respect the heterogeneous nature of (i)EEG, being able to compress data with an arbitrary number of channels.
> 2. The fMRI BrainCodec achieves a compression ration of 2.6, while our (i)EEG BrainCodec guarantees high-fidelity reconstruction at 64x construction. Such a comparison between different modalities may be difficult. However, we are confident that our methods that leverage high-quality training data and domain-specific loss functions could further improve fMRI BrainCodec, too.
> 3. fMRI BrainCodec could not make use of the discriminator due to the noisy training data. Our (i)EEG BrainCodec, however, leverages the discriminator to improve the reconstruction of high-frequency components.
>
> Despite these differences, we are encouraged by the potential interpretability analysis, which fMRI BrainCodec could perform on the compressed representation. We plan to conduct similar experiments with our (i)EEG in future work, and we added a citation to the Discussion section of this submission.

---

> > ### Comment · Reviewer_Wj41 · 2024-11-21
> > **Answer to authors**
> >
> > Thank you for the response which answers my concerns and questions: I am raising my score.

---

> > > ### Author Response · Authors · 2024-11-21
> > >
> > > We thank the Reviewer for acknowledging our response and raising our scores.

---

### Official Review · Reviewer_p69F · 2024-11-04

**Soundness:** 3
**Presentation:** 2
**Contribution:** 4
**Rating:** 8
**Confidence:** 4

**Summary:**

BrainCodec leverages the higher signal-to-noise ratio (SNR) of iEEG data by training on iEEG and then transferring this model to compress noisier EEG data. The model achieves up to 64x compression with minimal loss of reconstruction fidelity, outperforming existing methods. Experimental results demonstrate that training on high-SNR iEEG data consistently provides better performance on EEG compression tasks than models trained directly on EEG. BrainCodec’s performance remains strong across various EEG and iEEG datasets, showing promise for reducing storage and transmission costs in clinical and research settings without compromising downstream task performance, such as seizure detection and motor imagery classification.

**Strengths:**

The methodology is rigorously implemented, leveraging recent advancements in neural audio compression (e.g., quantized autoencoders) and adapted for EEG and iEEG data.BrainCodec’s performance is thoroughly validated across multiple datasets, including both iEEG (SWEC, MC) and EEG (CHB-MIT, BONN, BCI IV-2a), with comprehensive baseline comparisons. The cross-modality improvement, where a model trained on iEEG performs better on EEG than a model trained solely on EEG, adds to the literature, reinforcing the importance of high-SNR data for model training. The high compression ratios achieved (up to 64x without notable degradation) have immediate applications in reducing storage and transmission costs, relevant for clinical environments.

**Weaknesses:**

Figures could use refinement for clearer interpretation, especially when comparing compression ratios and PRD values. While the model’s effectiveness across modalities is demonstrated, the paper doesn’t explore the impact of training on a combined EEG and iEEG dataset, which could potentially yield a more robust generalizable model. In the future more baseline comparisons would be useful — perhaps across the Mother of all BCI Benchmarks (MOABB), or braindecode datasets would further validate this approach. Also, further clinical validation would be valuable to ascertain the practical application of this work in the clinical setting. The cross-modality improvement, where a model trained on iEEG performs better on EEG than a model trained solely on EEG, adds to the literature, reinforcing the importance of high-SNR data for model training.

**Questions:**

Could you clarify the rationale behind not training BrainCodec on a combined dataset of both EEG and iEEG? Do you anticipate any limitations or challenges with such an approach?

Did you consider other performance metrics beyond PRD and classification accuracy for assessing reconstruction fidelity?

---

> ### Author Response · Authors · 2024-11-19
>
> We thank the Reviewer for recognizing the strengths and potential impact of our Manuscript.
>
> Following the Reviewer's suggestions, we have updated the Manuscript to include the results of our combined EEG+iEEG BrainCodec.
>
> -------------
>
> ## W1. EEG+iEEG model
>
> We agree with the Reviewer that an EEG+iEEG model might reveal interesting insights on the behavior of BrainCodec. For this reason, we now add the results of our combined EEG+iEEG BrainCodec in the Results section and in Appendix D. We train this model using subject ID01 of the SWEC iEEG dataset and subject ID01 of the CHB EEG dataset.
>
> First, we show the results of the mixed-modality model on the SWEC iEEG dataset.
>
> |  Training dataset  |  CR ↑  |PRD ↓ | SNR ↑     |
> | ---------------------: | --------: | -----------: | -----------: |
> | SWEC                   | 64        | 15\.76       | 18\.01       |
> | CHB                    |           | 22\.17       | 16\.23       |
> | SWEC+CHB               |           | 16\.21       | 17\.63       |
> | SWEC                   | 16        | 4\.60        | 31\.08       |
> | CHB                    |           | 7\.79        | 28\.63       |
> | SWEC+CHB               |           | 4\.78        | 30\.35       |
> | SWEC                   | 8         | 1\.88        | 39\.86       |
> | CHB                    |           | 3\.56        | 36\.60       |
> | SWEC+CHB               |           | 2\.16        | 39\.29       |
> | SWEC                   | 4         | 0\.94        | 50\.95       |
> | CHB                    |           | 1\.35        | 51\.11       |
> | SWEC+CHB               |           | 0\.95        | 47\.78       |
> | SWEC                   | 2         | 0\.66        | 57\.60       |
> | CHB                    |           | 0\.65        | 61\.76       |
> | SWEC+CHB               |           | 0\.89        | 54\.63       |
>
> The reconstruction fidelity of mixed-modal BrainCodec is overall inferior compared to the model trained exclusively on the SWEC iEEG dataset. This is consistent with earlier results, as iEEG-trained models always perform better on iEEG than EEG-trained models.
>
> This is not the case when testing the mixed-modality model on the CHB EEG dataset. Here, we surpass the CHB-trained model already from a compression ratio of 4. Moreover, we always perform better than a model trained exclusively on iEEG data. This indicates that mixed-modality training is beneficial to compressing EEG data.
>
> |  Training dataset  |  CR ↑  |PRD ↓ | SNR ↑     |
> | ---------------------: | --------: | -----------: | -----------: |
> | CHB                    | 64        | 33\.66       | 10\.96       |
> | SWEC                   |           | 32\.00       | 11\.70       |
> | SWEC+CHB               |           | 28\.73       | 11\.71       |
> | CHB                    | 16        | 13\.97       | 21\.01       |
> | SWEC                   |           | 14\.88       | 19\.99       |
> | SWEC+CHB               |           | 10\.98       | 21\.54       |
> | CHB                    | 8         | 7\.10        | 28\.32       |
> | SWEC                   |           | 11\.42       | 23\.70       |
> | SWEC+CHB               |           | 5\.51        | 28\.13       |
> | CHB                    | 4         | 3\.10        | 42\.22       |
> | SWEC                   |           | 8\.82        | 27\.36       |
> | SWEC+CHB               |           | 1\.78        | 37\.49       |
> | CHB                    | 2         | 0\.61        | 55\.30       |
> | SWEC                   |           | 6\.65        | 30\.66       |
> | SWEC+CHB               |           | 1\.04        | 46\.81       |
>
> Overall, w e find that training BrainCodec on both EEG and iEEG improves fidelity when reconstructing EEG, without compromising performance when reconstructing iEEG. The results of the EEG+iEEG model reinforce our conclusion that transfering from high-SNR signals to low-SNR signals is beneficial.
>
> ----------------
>
> ## W2. Mother of all benchmarks
>
> We thank the Reviewer for indicating this benchmark. While we wish to provide as much evaluation of BrainCodec as possible, computational constraints limit us from being able to provide results on this benchmark in the given small time-frame.
>
> Nonetheless, we believe our wide range of tasks covering both EEG and iEEG effectively characterize the behavior of BrainCodec on downstream classification.
>
> Moreover, to further increase the size and quality of our evaluation, we now add the Brain Treebank iEEG dataset to our set of testing datasets. We find that the performance on this dataset is comparable to the SWEC iEEG dataset, pointing towards the universality of our model.
>
> | Treebank | CR ↑  |PRD ↓ | SNR ↑ |
> |---------------|----|--------|-------|
> | Base [SWEC]            | 2 | 3.32  | 34.58 |
> |            | 4 | 3.54  | 33.70 |
> |            | 8 | 3.89  | 32.56 |
> |            | 16 | 5.26  | 28.92 |
> |           | 64 | 14.36  | 18.89 |

---

> ### Author Response · Authors · 2024-11-19
>
> ## Q1. EEG+iEEG model
>
> As detailed in **W1**, we now provide the results of our combined EEG and iEEG model trained on both the CHB and the SWEC datasets. The full results can be found in the Results and in Appendices C and D.
>
> -----------
>
> ## Q2. Additional performance metrics
>
> Following the Reviewers' suggestion, we now include four additional metrics borrowed from the signal processing domain, namely:
> 1. **PRD of the spectrogram (PRD-spec)**
> 2. **Root-mean-square error (RMSE)**
> 3. **Signal-to-noise ratio (SNR)**
> 4. **Peak signal-to-noise ratio (PSNR)**
>
> All tables in Appendix D have been updated accordingly, showing that BrainCodec's high reconstruction fidelity translates to these four additional evaluation metrics as well.
>
> Briefly here as an example, we report all the metrics collected for the Base model trained and tested on the iEEG SWEC dataset.
>
> |CR ↑  |PRD ↓ |PRD-spec ↓|RMSE ↓|SNR ↑  |PSNR ↑ |
> |----|----|--------|----|-----|-----|
> |2   |0.66|0.81    |1.67|57.60|70.14|
> |4   |0.94|1.05    |2.12|50.95|64.89|
> |8   |1.88|1.46    |3.11|39.86|55.12|
> |16  |4.60|3.39    |6.33|31.08|46.00|
> |64  |15.76|12.35   |16.69|18.01|33.84|

---

### Official Review · Reviewer_5Spf · 2024-11-09

**Soundness:** 2
**Presentation:** 3
**Contribution:** 2
**Rating:** 3
**Confidence:** 4

**Summary:**

This paper focuses on the neural signal compression. The authors propose the BrainCodec model for both EEG and iEEG data compression. The proposed method has been tested on several data sets, and  achieves superior reconstruction fidelity compared with other competing methods. The experiments further validate its effectiveness for downstream tasks.

**Strengths:**

1.	It is a universal compression model for both iEEG and EEG.
2.	It achieves high-fidelity compression of EEG signals up to a compression ratio of 64.
3.	It has been validated both experimentally and being rated by the neurologists.

**Weaknesses:**

1.	BraiCodec is developed based on the quantized autoencoder design. The methodology novelty is not clear.
2.	The experimental validation is insufficient. The effectiveness of the loss functions and different modules are not examined.
3.	The evaluation metrics of the compression performance are limited, and the rational of choice of some parameters for evaluation are not stated.

**Questions:**

1.	Neural signal compression has less been investigated. Real EEG/iEEG monitoring often requires real-time analysis. What are the application scenarios for their compression?
2.	Please pay attention to the table format.

---

> ### Author Response · Authors · 2024-11-19
>
> We wish to thank the Reviewer for the constructive criticism and for recognizing the universality, the high fidelity, and the comprehensive evaluation of our proposed compression method.
>
> We have uploaded a revised version of the Manuscript incorporating the suggested ablations and have further clarified the immediate practical applications of BrainCodec.
>
> We briefly highlight the main contributions of this work:
> 1. BrainCodec is a quantized neural compressor based on EnCodec and Soundstream with specific improvements for compressing neurophysiological signal, for example the line length loss;
> 2. Our training methodology fully embraces the specific characteristics of scalp EEG and intracranial EEG, indicating the signficant advantages of high-SNR signals such as iEEG even when compressing low-SNR EEG;
> 3. Our testing methodology is exhaustive, with multiple classical fidelity metrics and expert manual evaluation by a trained neurologist. At the same time, we also evaluate on downstream classification tasks, to showcase the practicality of BrainCodec.
>
> Moreover, during the writing of this rebuttal, we have also added the following results and contributions:
> 1. We have significantly improved the robustness of our ablations, including RVQ parameters, Encoder parameters, line length loss, and more;
> 2. We have increased the variety of the reported fidelity metrics, now including PRD, PRD of the spectrogram, RMSE, SNR, and peak SNR;
> 3. We have assessed the performance of BrainCodec in a mixed-modal scenario when trained on both EEG and iEEG data, and found that the performance transfer we had found from iEEG to EEG also holds in this case;
> 4. We have added a new iEEG dataset, the Brain Treebank datasets, bringing us to a total of 3 iEEG datasets and 3 iEEG datasets.
>
> We now address each raised point separately.
>
> ---------------
>
> ## W1. Methodological novelty
>
> We base our BrainCodec on well-known and battle-tested neural compression models, namely EnCodec and SoundStream, because these RVQ-VAE architectures have demonstrated excellent results on audio data. However, audio and electrophysiological signals have little in common beyond both being time-series. Indeed, naïvely applying EnCodec to iEEG data does not yield a useful compressor.
>
> Moreover, there are specific features of EEG and iEEG that need to be preserved after compression to maintain both signal fidelity and downstream classification performance. For example, the effects of epilepsy on the iEEG drove us to adopt the line length loss, which improves the reconstruction of both ictal and inter-ictal segments (see the next point).

---

> > ### Author Response · Authors · 2024-11-19
> >
> > ## W2. Experimental validation
> >
> > We strive to validate BrainCodec in many scenarios to improve its real-world applicability. As suggested by the Reviewer we have now included ablations on the following:
> > 1. **Number of residuals.** We assess the performance at varying compression ratios by varying RVQ's number of residuals from 1 to 16 (i.e., 16 to 256 compression ratio). For our main results, we vary the Encoder framerate instead, because we find it yields superior results. For example, at 16 compression ratio we report a PRD of 4.60 against 8.74, a notable improvement.
> > 2. **Codeword size.** We vary the size of each codeword from 64 to 1024 (i.e., a compression ratio of ~51 to ~85). This gives diminishing returns, with our choice of 256 a consistent middle-ground.
> > 3. **Line length loss.** We train a model without the line length loss. We find that the performance is decreased, with a PRD of 15.95 against 15.76. However, the difference is especially noticeable in the reconstruction of higher frequencies in the spectrogram, as reported in Appendix B.4.
> > 4. **Kernel size.** We vary the initial kernel size from 3 to 7, and show that our choice (3) yields the best results.
> > 5. **Number of patients.** We train additional models using 2 and 4 training patients. The results once again point to diminishing returns, indicating that our single-patient training already produces high-fidelity reconstructions, as highlighted by the Reviewer.
> >
> > We report the most relevant ablations here. First, varying the number of residuals yields worse performance with respect to varying the Encoder framerate.
> >
> > | Residuals | CR ↑  |PRD ↓ | SNR ↑  |
> > |-----------|-----|-------|-------|
> > | 1         | 256 | 38.36 | 9.67  |
> > | 4         | 64  | 15.76 | 18.01 |
> > | 8         | 32  | 10.26 | 22.04 |
> > | 16        | 16  | 8.74  | 23.76 |
> >
> > Second, the line length loss improves PRD, but most importantly improves the reconstruction of higher frequencies, as seen in Appendix B.4.
> >
> > | Line length loss | CR ↑  |PRD ↓ | SNR ↑  |
> > |------------------|----|-------|-------|
> > | Yes              | 64 | 15.76 | 18.01 |
> > | No               | 64 | 15.95 | 17.33 |
> >
> > Overall, our results show that our choices of modules and objectives --- consistent with the characteristics of EEG and iEEG signals --- improve the reconstruction fidelity of BrainCodec and strengthen its effectiveness.
> >
> > Moreover, to further increase the size and quality of our evaluation, we now add the Brain Treebank iEEG dataset to our set of testing datasets. We find that the performance on this dataset is comparable to the SWEC iEEG dataset, pointing towards the universality of our model.
> >
> > | Treebank | CR ↑  |PRD ↓ | SNR ↑ |
> > |---------------|----|--------|-------|
> > | Base [SWEC]            | 2 | 3.32  | 34.58 |
> > |            | 4 | 3.54  | 33.70 |
> > |            | 8 | 3.89  | 32.56 |
> > |            | 16 | 5.26  | 28.92 |
> > |           | 64 | 14.36  | 18.89 |

---

> ### Author Response · Authors · 2024-11-19
>
> ## W3. Evaluation metrics
>
> We choose three separate evaluation metrics to better characterize the performance of BrainCodec:
>
> 1. **Percent-root-mean-square-difference (PRD).** PRD is the most used metric to determine the reconstruction fidelity of compressors on EEG and iEEG, as seen in the literature [1,2,3].
> 2. **Downstream classification performance.** Classification tasks are a concrete benchmark for the performance of BrainCodec, as they provide an immediately parsable metric of the distortion caused by the compression.
> 3. **Subjective expert evaluation.** The specific characteristics of EEG and iEEG are not as easily parsed as music or speech. Therefore, we assess the reconstruction fidelity through the expert evaluation of a neurologist who works with this data every day and is best poised to capture the fine difference between the original and the compressed signal.
>
> Following the Reviewers' suggestion, we now include four additional metrics borrowed from the signal processing domain, namely:
> 1. **PRD of the spectrogram (PRD-spec)**
> 2. **Root-mean-square error (RMSE)**
> 3. **Signal-to-noise ratio (SNR)**
> 4. **Peak signal-to-noise ratio (PSNR)**
>
> All tables in Appendix D have been updated accordingly, showing that BrainCodec's high reconstruction fidelity translates to these four additional evaluation metrics as well.
>
> Briefly here as an example, we report all the metrics collected for the Base model trained and tested on the iEEG SWEC dataset.
>
> |CR ↑  |PRD ↓ |PRD-spec ↓|RMSE ↓|SNR ↑  |PSNR ↑ |
> |----|----|--------|----|-----|-----|
> |2   |0.66|0.81    |1.67|57.60|70.14|
> |4   |0.94|1.05    |2.12|50.95|64.89|
> |8   |1.88|1.46    |3.11|39.86|55.12|
> |16  |4.60|3.39    |6.33|31.08|46.00|
> |64  |15.76|12.35   |16.69|18.01|33.84|
>
> [1] Higgins, Garry, et al. "EEG compression using JPEG2000: How much loss is too much?." 2010 Annual International Conference of the IEEE Engineering in Medicine and Biology. IEEE, 2010.
>
> [2] Du, XiuLi, et al. "Fast reconstruction of EEG signal compression sensing based on deep learning." Scientific Reports 14.1 (2024): 5087.
>
> [3] Nguyen, Binh, et al. "Wavelet transform and adaptive arithmetic coding techniques for EEG lossy compression." 2017 international joint conference on neural networks (IJCNN). IEEE, 2017.
>
> --------------
>
> ## W4. Choice of parameters
>
> We base our choice of parameters on extensive evaluations which we have now included in Appendices B.1 to B.4. As mentioned in **W2** above, our choices of modules and parameters improve the reconstruction fidelity of BrainCodec.
>
> -----------------
>
> ## Q1. Application scenarios
>
> We believe BrainCodec is immediately applicable in any EEG and iEEG processing and classification pipeline. BrainCodec drastically improves data density via high-fidelity compression, paving the way for straightforward adoption in at least two scenarios.
>
> First, for wearable, or ultimately implantable recording devices: BrainCodec reduces the number of bits that must be transferred in the air to a receiving device, with wireless communication being the most energy-consuming operation of the device. In particular, we show that even a 64x compression, BrainCodec has no negative effect on downstream tasks. Hence, BrainCodec can effectively save significant energy in communications, leading to longer battery life for long-term monitoring devices. As the encoder of BrainCodec has ~1M parameters, its implementation on such edge devices would be feasible, and its weights could be quantized for more efficient inference.
>
> Second, in static setups, BrainCodec will lower resource requirements to store large quantities of EEG and iEEG data. Storage costs are also relevant concerns in the clinical domain. Indeed, the sharp rise in long-term recordings and healthcare-related wearable devices has also increased the rate at which electrophysiological data is collected in the clinic. BrainCodec will help mitigate these concerns in a real-time scenario as well, as it can compress and decompress up to 20 minutes of data per second on one Nvidia A100 GPU.
>
> -----------------
>
> ## Q2. Tables
>
> We have revised all our tables to add the new reconstruction metrics. We have also added a caption to all our tables.
>
> By clarifying and increasing the thoroughness of our evaluation in many scenarios involved both in the methodology and the concrete application of our BrainCodec, we hope to have improved on all areas which were not satisfactory to the Reviewer.

---

> > ### Author Response · Authors · 2024-11-24
> >
> > Thank you once again for your valuable feedback and dedicated service as a reviewer. We have carefully considered your input and made significant updates to the manuscript in response. We genuinely value your perspective and encourage you to review our rebuttal and the revised manuscript. Given the positive feedback from other reviewers, we hope our revisions address your concerns and provide the clarity needed to reconsider your assessment. Please don not hesitate to reach out if further clarification or discussion would be helpful.

---

### Meta-Review · Area_Chair_9sAq · 2024-12-16

**Metareview:**

This work proposes a neural compressor for EEG and iEEG data that is inspired by the Encodec and Soundstream approaches proposed for audio signals in 2023 and 2022. Three reviewers acknowledge the originality of the approach and are enthusiastic about the contribution that is judged convincing and well presented.

The paper is considered relevant and timely for the ICLR 2024 community, in particular the researchers interested in deep learning on biosignals.

As the AC, I encourage the authors to improve their supplementary material that contains the code. Please add a readme and the tooling necessary to more easily replicate the model training. You may also consider adding a notebook and a checkpoint showing how to compress a public data with your model.

**Additional Comments On Reviewer Discussion:**

5Spf remains unconvinced about the validation and use case of the method while all other 3 reviewers acknowledge the novelty and the convincing results provided by the paper. **3 reviewers do champion the paper with a rating of 8** and the review of 5Spf remains shallow after the discussions.

---

> ### Public Comment · ~Francesco_S._Carzaniga1 · 2025-02-10
>
> We thank the Area Chair for the positive response and all the Reviewers for their constructive feedback, which helped improve our work.
>
> Following acceptance and the suggestion of the AC, we have now revised our code and open sourced it at [BrainCodec](https://github.com/IBM/eeg-ieeg-brain-compressor).

---

### Decision · Program_Chairs · 2025-01-22

Accept (Poster)